The phylogenetic nomenclature of ornithischian dinosaurs

http://orcid.org/0000-0003-1228-3573 Madzia Daniel 1 daniel.madzia@gmail.com
http://orcid.org/0000-0003-4788-0546 Arbour Victoria M. 2 3
Boyd Clint A. 4
http://orcid.org/0000-0002-6930-2002 Farke Andrew A. 5
http://orcid.org/0000-0002-5819-8254 Cruzado-Caballero Penélope 6 7 8 9
http://orcid.org/0000-0001-9608-6635 Evans David C. 10
1 Department of Evolutionary Paleobiology, Institute of Paleobiology, Polish Academy of Sciences , Warsaw , Poland
2 Department of Knowledge, Royal BC Museum , Victoria, BC , Canada
3 School of Earth and Ocean Sciences, University of Victoria , Victoria, BC , Canada
4 North Dakota Geological Survey , Bismarck, ND , USA
5 Raymond M. Alf Museum of Paleontology at The Webb Schools , Claremont, CA , USA
6 Área de Paleontología, Departamento de Biología Animal, Edafología y Geología, Universidad de La Laguna , Santa Cruz de Tenerife , Spain
7 Instituto de Investigación en Paleobiología y Geología (IIPG), Universidad Nacional de Río Negro (UNRN) , Río Negro , Argentina
8 Instituto de Investigación en Paleobiología y Geología (IIPG), Consejo Nacional de Investigaciones Científicas y Tecnológicas (CONICET) , Río Negro , Argentina
9 Grupo Aragosaurus-IUCA, Área de Paleontología, Departamento de Ciencias de la Tierra, Universidad de Zaragoza , Zaragoza , Spain
10 Department of Natural History, Royal Ontario Museum , Toronto, ON , Canada
Knoll Fabien
Electronic publication date: 2021 Dec 9
Publication date: 2021
Volume: 9
Electronic Location ID: e12362
Received 2021 May 4; Accepted 2021 Sep 30
Copyright: © 2021 Madzia et al.
Copyright year: 2021
Copyright holder: Madzia et al.
License: This is an open access article distributed under the terms of the Creative Commons Attribution License, which permits unrestricted use, distribution, reproduction and adaptation in any medium and for any purpose provided that it is properly attributed. For attribution, the original author(s), title, publication source (PeerJ) and either DOI or URL of the article must be cited.
License URL: https://creativecommons.org/licenses/by/4.0/

Keywords: Phylogenetic nomenclature, Phylogenetic definition, PhyloCode, International Code of Phylogenetic Nomenclature, Ornithischia, Dinosauria

Funding: National Science Centre (Poland) 2020/37/B/NZ8/01321 Natural Sciences and Engineering Research Council Discovery Grant RGPIN-2020-04012 The study was funded by the National Science Centre (Poland) grant no. 2020/37/B/NZ8/01321 to Daniel Madzia. Victoria M. Arbour is funded by a Natural Sciences and Engineering Research Council Discovery Grant (RGPIN-2020-04012). The funders had no role in study design, data collection and analysis, decision to publish, or preparation of the manuscript.

==============================
Ornithischians form a large clade of globally distributed Mesozoic dinosaurs, and represent one of their three major radiations. Throughout their evolutionary history, exceeding 134 million years, ornithischians evolved considerable morphological disparity, expressed especially through the cranial and osteodermal features of their most distinguishable representatives. The nearly two-century-long research history on ornithischians has resulted in the recognition of numerous diverse lineages, many of which have been named. Following the formative publications establishing the theoretical foundation of phylogenetic nomenclature throughout the 1980s and 1990s, many of the proposed names of ornithischian clades were provided with phylogenetic definitions. Some of these definitions have proven useful and have not been changed, beyond the way they were formulated, since their introduction. Some names, however, have multiple definitions, making their application ambiguous. Recent implementation of the International Code of Phylogenetic Nomenclature (ICPN, or PhyloCode) offers the opportunity to explore the utility of previously proposed definitions of established taxon names. Since the Articles of the ICPN are not to be applied retroactively, all phylogenetic definitions published prior to its implementation remain informal (and ineffective) in the light of the Code. Here, we revise the nomenclature of ornithischian dinosaur clades; we revisit 76 preexisting ornithischian clade names, review their recent and historical use, and formally establish their phylogenetic definitions. Additionally, we introduce five new clade names: two for robustly supported clades of later-diverging hadrosaurids and ceratopsians, one uniting heterodontosaurids and genasaurs, and two for clades of nodosaurids. Our study marks a key step towards a formal phylogenetic nomenclature of ornithischian dinosaurs.

Introduction

The ornithischian, or ‘bird-hipped’, dinosaurs were a species-rich clade of Mesozoic archosaurs that first appeared in the Triassic (e.g., Langer & Ferigolo, 2013; Cabreira et al., 2016; Pacheco et al., 2019; Desojo et al., 2020; Müller & Garcia, 2020) or the earliest Jurassic (Agnolín & Rozadilla, 2018; Baron, 2019) and died out during the Cretaceous/Paleogene extinction event (e.g., Brusatte et al., 2015). Throughout their >134 million-year-long evolutionary history, ornithischians achieved global distribution (Weishampel et al., 2004; Boyd, 2015), evolved considerable taxic diversity (Tennant, Chiarenza & Baron, 2018), and an apparent morphological disparity, expressed through their markedly different body sizes (Benson et al., 2018) and especially the ‘exaggerated’ structures of the crania and osteodermal armor of some of their most distinctive members (e.g., Brown, 2017; Stubbs et al., 2019).

Here, we provide a nomenclatural revision of ornithischian dinosaur clades. Following the pivotal, early formative publications establishing the theoretical foundation of the phylogenetic nomenclature in the 1980s and early 1990s (e.g., Ghiselin, 1984; Gauthier, 1986; Rowe, 1987; de Queiroz, 1988; Estes, de Queiroz & Gauthier, 1988; Gauthier, de Queiroz & Estes, 1988; de Queiroz & Gauthier, 1990, 1992, 1994), many names of the ornithischian clades were provided phylogenetic definitions (e.g., Padian & May, 1993; Currie & Padian, 1997; Sereno, 1998; Sereno, 1999), some of which have proven useful and have not been changed, beyond the way they were formulated, since their introduction.

The implementation of the International Code of Phylogenetic Nomenclature, or the PhyloCode (de Queiroz & Cantino, 2020), an evolution-based system for naming organisms, hereafter abbreviated and referred to as ICPN (accessible at http://phylonames.org/code/), and parallel publication of Phylonyms: A Companion to the PhyloCode (de Queiroz, Cantino & Gauthier, 2020), offers the opportunity to consider the utility of previously proposed phylogenetic definitions of established taxon names and, in appropriate cases, formalize their use, as specified by the Articles of the ICPN.

Recent studies have thoroughly assessed the use of clade names applied to some ornithischian lineages, mostly early-diverging neornithischians and ornithopods (Boyd, 2015; Madzia, Boyd & Mazuch, 2018; Herne et al., 2019; Madzia, Jagt & Mulder, 2020). However, the Articles of the ICPN are not to be applied retroactively (ICPN: Preamble 6, see also Art. 7.1). As such, all these efforts remain informal and ineffective in the light of the Code.

We formalize some of the nomenclatural acts of previous studies and introduce phylogenetic definitions for 81 names of ornithischian dinosaur clades. Specifically, we provide formal phylogenetic definitions for the following 76 preexisting taxon names: Ankylopollexia, Ankylosauria, Ankylosauridae, Ankylosaurinae, Ankylosaurini, Aralosaurini, Brachylophosaurini, Camptosauridae, Centrosaurinae, Centrosaurini, Cerapoda, Ceratopsia, Ceratopsidae, Ceratopsoidea, Chaoyangsauridae, Chasmosaurinae, Clypeodonta, Coronosauria, Dryomorpha, Dryosauridae, Edmontosaurini, Elasmaria, Eucentrosaura, Euhadrosauria, Euiguanodontia, Euornithopoda, Eurypoda, Genasauria, Hadrosauridae, Hadrosauriformes, Hadrosaurinae, Hadrosauroidea, Hadrosauromorpha, Heterodontosauridae, Huayangosauridae, Hypsilophodontia, Hypsilophodontidae, Iguanodontia, Iguanodontidae, Jeholosauridae, Kritosaurini, Lambeosaurinae, Lambeosaurini, Leptoceratopsidae, Marginocephalia, Nasutoceratopsini, Neoceratopsia, Neoiguanodontia, Neornithischia, Nodosauridae, Nodosaurinae, Ornithischia, Ornithopoda, Orodrominae, Pachycephalosauria, Pachycephalosauridae, Pachycephalosaurinae, Pachycephalosaurini, Pachyrhinosaurini, Pachyrostra, Parasaurolophini, Polacanthinae, Protoceratopsidae, Rhabdodontidae, Rhabdodontomorpha, Saurolophinae, Saurolophini, Shamosaurinae, Stegosauria, Stegosauridae, Styracosterna, Thescelosauridae, Thescelosaurinae, Thyreophora, Triceratopsini, and Tsintaosaurini. These names cover all major ornithischian clades and the vast majority of their subclades for which taxon names were used and defined in the past. Additionally, we introduce five new clade names: Corythosauria, for the node uniting lambeosaurin and parasaurolophin lambeosaurines, Euceratopsia, for the node uniting leptoceratopsid and coronosaur ceratopsians, Saphornithischia, for the node uniting heterodontosaurids and genasaurs, and Panoplosaurini and Struthiosaurini for clades of later-diverging nodosaurids.

Methods

Protocol

In order to be formally established under the ICPN, clade names must comply especially with the provisions of Articles 7 and 9–11 of the Code (ICPN: Art. 7.2d). These Articles are fully followed here. The entries, provided in ‘Phylogenetic nomenclature of ornithischian clades’ below, partly follow the scheme used in Phylonyms (de Queiroz, Cantino & Gauthier, 2020); they include the following sub-sections: ‘Definition’, ‘Reference phylogeny’, ‘Composition’, ‘Synonyms’, and ‘Comments’. The sub-sections ‘Diagnostic apomorphies’ and ‘Etymology’, as used in Phylonyms, have been omitted. Note that detailed discussion of apomorphies is not strictly required by the Code, and inclusion of a reference phylogeny alone is sufficient (ICPN: Art. 9.13). Recent assessments of the phylogenetic relationships of numerous taxa, particularly those nested near the basal neornithischian-ornithopod transition, but also within some major clades, such as ornithopods, currently provide conflicting results (e.g., Norman, 2015; Han et al., 2018; Madzia, Boyd & Mazuch, 2018; Andrzejewski, Winkler & Jacobs, 2019; Herne et al., 2019; Párraga & Prieto-Márquez, 2019; Dieudonné et al., 2020; Yang et al., 2020; Barta & Norell, 2021; Černý, Madzia & Slater, 2021). It is extremely difficult, and perhaps impossible at the moment, to list unambiguous diagnostic apomorphies for many clades that have long been associated with widely-used names, and detailed discussion would be far beyond the scope of the paper. Instead, emphasis was placed on using definitions that are reflective of all currently inferred phylogenies. In turn, ‘Etymology’ was omitted because all but five of the clade names that are established in the present study are preexisting (Art. 6.2 of the ICPN). The only reason for discussing the etymological origin of taxon names would be to provide arguments for the inclusion of certain internal specifiers (e.g., within the context of Art. 11.10 of the ICPN). With that respect, relevant comments are provided in the ‘Comments’ sub-section of the name entries. The five new clade names introduced in the present study are provided with their etymologies. Additionally, owing to the fact that the phylogenetic relationships of ornithischian dinosaurs are intensively researched, each clade name entry could be supplemented with numerous reference phylogenies. Rather than list all of the relevant phylogeny reconstructions available, we decided to refer to a subset of the more recent tree topologies that justify the ‘conversion’ of the taxon name in accordance with the ICPN.

We have not followed any strict approach while selecting primary reference phylogenies. Instead of providing references to studies that represent, for example, the most recent iterations of some datasets, we preferred to refer to studies that we have been either directly involved in, and are therefore familiar with the original data used for phylogeny inference, or consider to cover relevant data sampling.

With respect to the clade ‘Composition’, we list only those subtaxa that are included in the primary reference phylogeny. It is essential to realize that some of the clades for which names are provided have insufficiently explored origins and their basal branching is expressed through polytomies (this applies especially, but not exclusively, to non-hadrosaurid ornithopod subclades). In such cases, the actual extent may not be certain and some of the taxa listed in the ‘Composition’ subsection may in fact fall outside the clades. Note also that some of the selected primary reference phylogenies do not show the placements of all taxa used as specifiers in the definitions of the names to be defined. In such cases, the phylogenetic positions of these specifiers are discussed in the ‘Comments’.

We also realize that the list of taxon names provided in ‘Synonyms’ is not exhaustive and does not list all historically used approximate synonyms. When discussing names that may be considered synonymous with those whose application is preferred here, we have focused especially on those names that have been used for the same or very similar contents in recent years, or those that have been used interchangeably with those that we define (e.g., Iguanodontidae and Iguanodontoidea, Thescelosauridae and Parksosauridae). Therefore, the names that have not been in use for a long time were mostly omitted.

Further, Article 8.1 of the ICPN states that, “(i)n order for a name to be established under [the ICPN], the name and other required information must be submitted to the registration database for phylogenetically defined names (see Art. 22.2). A name may be submitted to the database prior to acceptance for publication, but it is given only a temporary registration number at that time. The registration number will become permanent after the author notifies the database that the paper or book in which the name will appear has been published, provides a full reference to the publication, and confirms that the definition in the database is identical to that in the publication”. We have therefore registered all names, whose phylogenetic definitions are established in the present study, to the database of phylogenetically defined names, the RegNum (ICPN: Art. 22; Appendix A), and obtained registration numbers that are included in the clade name entries.

Finally, we follow the ICPN in that all scientific names are italicized (ICPN: Recommendation 6.1A.) and that names are attributed to the earliest author(s) to spell them rather than according to the Principle of Coordination (ICPN: Note 9.15A.3).

Phylogenetic definitions

The names of ornithischian clades are defined using the following two types of definitions: (a) minimum-clade definition, known previously as ‘node-based’ definition (ICPN: Art. 9.5) and (b) maximum-clade definition, known previously as ‘branch-based’ or ‘stem-based’ definition (ICPN: Art. 9.6). We refer to the aforementioned Articles of the ICPN for details.

Adopted conventions for abbreviated definitions. We abbreviate the definitions using the following conventions (as per Notes 9.4.1 and 11.12.1 of the ICPN): max = the largest; min = the smallest; ∇ = clade; () = containing; & = and; ∨ = or; ~ = but not (in trivial maximum-clade definitions) or it does not (while using a qualifying clause); | = on the condition that. See also Note 9.6.2 of the ICPN for explanation of differences between the use of ‘&’ and ‘∨’ in the definitions. Additionally, we apply the set theory symbols ∈, that means “belongs to”, and ∉, meaning “not element of”, to indicate that a name is applied within or outside another clade, respectively (see Euornithopoda, Jeholosauridae, Orodrominae, and Polacanthinae for some examples).

Selection of specifiers. Specifiers are selected following Art. 11 of the ICPN. Numerous names pertaining to ornithischian clades have been informally defined in the past and these definitions can still be considered applicable. We have attempted to formalize most of these definitions, providing only the changes that were necessary to reflect all currently inferred phylogenies and to comply with the Articles of the ICPN. However, in some cases we have decided to replace certain specifiers with taxa that we consider to be more appropriate candidates. For example, we have replaced Parasaurolophus walkeri Parks, 1922 in some definitions with Iguanodon bernissartensis Boulenger in Beneden, 1881 (designated as the type species of Iguanodon Mantell, 1825 by the International Commission on Zoological Nomenclature (2000)), provided that this taxon has always been considered part of the clade (when selected as an internal specifier) or outside the clade (when selected as an external specifier) whose name is being defined. I. bernissartensis is known based on multiple complete or near-complete individuals of different ontogenetic stages and has been extensively researched (e.g., Norman, 1980; Verdú et al., 2017). It has also been frequently used as the specifier in previous, informal phylogenetic definitions, and was recently included as the internal specifier of Dinosauria (Langer et al., 2020). It is further essential to note that some taxa had to be used as internal specifiers despite their suggested dubious taxonomic status. For example, Ceratops montanus Marsh, 1888 is the name-bearer of Ceratopsia, Ceratopsoidea, Ceratopsidae, and Ceratopsinae (the last name is not converted to a clade name in the present study). At the same time, however, the taxon is generally considered to lack diagnostic features and is commonly treated as a nomen dubium (e.g., Dodson, Forster & Sampson, 2004; Mallon et al., 2016). Following Article 11.10 of the ICPN (which specifies that “(w)hen a clade name is converted from a preexisting name that is typified under a rank-based code or is a new or converted name derived from the stem of a typified name, the definition of the clade name must use the type species of that preexisting typified name or of the genus name from which it is derived (or the type specimen of that species) as an internal specifier.”), Ceratops montanus must be the internal specifier (or among the internal specifiers) in the definitions of the names in question.

Phylogenetic Nomenclature of Ornithischian Clades

For the sake of clarity, all clade names are provided in alphabetical order. The definitions are summarized in Table 1. The extent of all clade names is further depicted on Fig. 1 that shows the relationships of taxa included in the present study as specifiers (both, internal as well as external) and additionally on Figs. 2–4 that represent selected ornithischian-wide phylogenies published within recent years: Madzia, Boyd & Mazuch (2018: Fig. 4B), Dieudonné et al. (2020: Figs. 1 and 2), and Yang et al. (2020: Fig. 12).

Figure 1 Specifier-based phylogeny of Ornithischia.

Subclade topologies reflect those of the primary reference phylogenies: Ankylosauria (Figure 11 of Arbour & Currie, 2016; Figure 5 of Rivera-Sylva et al., 2018a), Hadrosauridae (Figure 25 of Prieto-Márquez et al., 2013; Figure 18 of Prieto-Márquez, Wagner & Lehman, 2020), Marginocephalia (Figure 27 of Schott & Evans, 2017; Figure 9 of Chiba et al., 2018; Figure 9a of Fowler & Freedman Fowler, 2020; Figure 10 of Morschhauser et al., 2019; Figure 4 of Yu et al., 2020), non-ankylosaur Thyreophora (Figure 16 of Han et al., 2018), non-cerapod Neornithischia (Figure 4 of Madzia, Boyd & Mazuch, 2018), non-genasaur Ornithischia (Figure 4 of Madzia, Boyd & Mazuch, 2018), non-hadrosaurid Ornithopoda (Figure 2.26 of Norman, 2014; Figure 4 of Madzia, Boyd & Mazuch, 2018; Figure 12 of Madzia, Jagt & Mulder, 2020). Abbreviations: Ch. – Chasmosaurinae; Ni. – Neoiguanodontia; Pd. – Pachycephalosauridae; Pn. – Pachycephalosaurinae; Pr. – Pachyrostra; Rh. – Rhabdodontomorpha; Rd. – Rhabdodontidae; and Sh. – Shamosaurinae. Majority of the silhouettes were obtained from phylopic.org: Ankylosaurinae (Andrew A. Farke, CC BY 3.0), Camptosauridae (Tasman Dixon, public domain), Centrosaurinae (Andrew A. Farke, CC BY 3.0), Chaoyangsauridae (Andrew A. Farke, CC BY 3.0), Chasmosaurinae (Jagged Fang Designs, public domain), Dryosauridae (Gereth Monger, CC BY 3.0), Heterodontosauridae (Scott Hartman, CC BY 3.0), Iguanodontidae (Tasman Dixon, public domain), Lambeosaurinae (Dmitry Bogdanov, CC BY 3.0), Nodosaurinae (Scott Hartman, public domain), Polacanthinae (FunkMonk, public domain), Protoceratopsidae (Andrew A. Farke, CC BY 3.0), Rhabdodontidae (Scott Hartman, CC BY 3.0), Stegosauria (Scott Hartman, CC BY 3.0). We have further added silhouettes for Elasmaria (Victoria M. Arbour, CC BY 4.0), Pachycephalosauria (Victoria M. Arbour, CC BY 4.0), Saurolophinae (Victoria M. Arbour, CC BY 4.0), and Thescelosauridae (Victoria M. Arbour, CC BY 4.0).

Figure 2 The phylogenetic nomenclature of ornithischian dinosaurs using the topology of Madzia, Boyd & Mazuch (2018: Fig. 4B).

Note that Nanosaurus agilis has been analyzed by Madzia, Boyd & Mazuch (2018) as ‘Othnielosaurus’. The name was changed here following Carpenter & Galton (2018). Additionally, the name Marasuchus lilloensis was placed in quotation marks to highlight that the taxon may not be distinct from Lagosuchus talampayensis (Agnolin & Ezcurra, 2019).

Figure 3 The phylogenetic nomenclature of ornithischian dinosaurs using the topology of Dieudonné et al. (2020: Figs. 1 and 2).

Note that Dieudonné et al. (2020) followed Carpenter & Lamanna (2015) in placing aphanoecetes within Camptosaurus. Owing to the results of recent phylogenetic analyses (e.g., Madzia, Jagt & Mulder, 2020; Verdú et al., 2020), aphanoecetes is placed here within Uteodon McDonald, 2011. Additionally, the name Psittacosaurus major was changed to Psittacosaurus lujiatunensis (following Hedrick & Dodson, 2013), and Ankylosauria and Stegosauria of Dieudonné et al. (2020) were placed in quotation marks to highlight that these names have not been necessarily used by the authors as defined in the present study. Note also that the extent of Ornithischia is difficult to indicate on the tree because Chilesaurus diegosuarezi may represent a theropod (see ‘Discussion’). Abbreviation: An. – Ankylosauria.

Figure 4 The phylogenetic nomenclature of ornithischian dinosaurs using the topology of Yang et al. (2020: Fig. 12).

Ankylosauria and Stegosauria of Yang et al. (2020) were placed in quotation marks to highlight that these names have not been necessarily used by the authors as defined in the present study. In turn, Psittacosauridae of Yang et al. (2020) was placed in quotation marks because the name has not been formally defined yet. Abbreviation: St. – Stegosauria.

Table 1 The phylogenetic nomenclature of ornithischian dinosaurs.

Clade name	Authorship	Definition type	Abbreviated definition	Primary reference phylogeny	
Ankylopollexia	Sereno, 1986	minimum-clade	min ∇ (Camptosaurus dispar (Marsh, 1879) & Iguanodon bernissartensis Boulenger in Beneden, 1881)	Figure 12 of Madzia, Jagt & Mulder (2020)	
Ankylosauria	Osborn, 1923	maximum-clade	max ∇ (Ankylosaurus magniventris Brown, 1908 ~ Stegosaurus stenops Marsh, 1887)	Figure 11 of Arbour & Currie (2016)	
Ankylosauridae	Brown, 1908	maximum-clade	max ∇ (Ankylosaurus magniventris Brown, 1908 ~ Nodosaurus textilis Marsh, 1889)	Figure 11 of Arbour & Currie (2016)	
Ankylosaurinae	Nopcsa, 1918	maximum-clade	max ∇ (Ankylosaurus magniventris Brown, 1908 ~ Shamosaurus scutatus Tumanova, 1983)	Figure 11 of Arbour & Currie (2016)	
Ankylosaurini	Arbour & Currie, 2016	maximum-clade	max ∇ (Ankylosaurus magniventris Brown, 1908 ~ Pinacosaurus grangeri Gilmore, 1933 & Saichania chulsanensis Maryańska, 1977)	Figure 11 of Arbour & Currie (2016)	
Aralosaurini	Prieto-Márquez et al., 2013	maximum-clade	max ∇ (Aralosaurus tuberiferus Rozhdestvensky, 1968 & Canardia garonnensis Prieto-Márquez et al., 2013 ~ Lambeosaurus lambei Parks, 1923 & Parasaurolophus walkeri Parks, 1922 & Tsintaosaurus spinorhinus Young, 1958)	Figure 25 of Prieto-Márquez et al. (2013)	
Brachylophosaurini	Gates et al., 2011	maximum-clade	max ∇ (Brachylophosaurus canadensis Sternberg, 1953 ~ Edmontosaurus regalis Lambe, 1917 & Hadrosaurus foulkii Leidy, 1858 & Kritosaurus navajovius Brown, 1910 & Saurolophus osborni Brown, 1912)	Figure 18 of Prieto-Márquez, Wagner & Lehman (2020)	
Camptosauridae	Marsh, 1885	maximum-clade	max ∇ (Camptosaurus dispar (Marsh, 1879) ~ Iguanodon bernissartensis Boulenger in Beneden, 1881)	Figure 13 of Madzia, Jagt & Mulder (2020)	
Centrosaurinae	Lambe, 1915	maximum-clade	max ∇ (Centrosaurus apertus Lambe, 1905 ~ Chasmosaurus belli (Lambe, 1902) & Triceratops horridus Marsh, 1889)	Figure 9 of Chiba et al. (2018)	
Centrosaurini	Ryan et al., 2017	maximum-clade	max ∇ (Centrosaurus apertus Lambe, 1905 ~ Pachyrhinosaurus canadensis Sternberg, 1950)	Figure 9 of Chiba et al. (2018)	
Cerapoda	Sereno, 1986	minimum-clade	min ∇ (Iguanodon bernissartensis Boulenger in Beneden, 1881 & Pachycephalosaurus wyomingensis (Gilmore, 1931) & Triceratops horridus Marsh, 1889)	Figure 4 of Madzia, Boyd & Mazuch (2018)	
Ceratopsia	Marsh, 1890	maximum-clade	max ∇ (Ceratops montanus Marsh, 1888 & Triceratops horridus Marsh, 1889 ~ Pachycephalosaurus wyomingensis (Gilmore, 1931))	Figure 10 of Morschhauser et al. (2019)	
Ceratopsidae	Marsh, 1888	minimum-clade	min ∇ (Centrosaurus apertus Lambe, 1905 & Ceratops montanus Marsh, 1888 & Chasmosaurus belli (Lambe, 1902) & Triceratops horridus Marsh, 1889)	Figure 4 of Yu et al. (2020)	
Ceratopsoidea	Hay, 1902	maximum-clade	max ∇ (Ceratops montanus Marsh, 1888 & Triceratops horridus Marsh, 1889 ~ Protoceratops andrewsi Granger & Gregory, 1923)	Figure 4 of Yu et al. (2020)	
Chaoyangsauridae	Zhao, Cheng & Xu, 1999	maximum-clade	max ∇ (Chaoyangsaurus youngi Zhao, Cheng & Xu, 1999 ~ Psittacosaurus mongoliensis Osborn, 1923 & Triceratops horridus Marsh, 1889)	Figure 10 of Morschhauser et al. (2019)	
Chasmosaurinae	Lambe, 1915	maximum-clade	max ∇ (Chasmosaurus belli (Lambe, 1902) & Triceratops horridus Marsh, 1889 ~ Centrosaurus apertus Lambe, 1905)	Figure 9a of Fowler & Freedman Fowler (2020)	
Clypeodonta	Norman, 2014	minimum-clade	min ∇ ∈ Ornithopoda (Edmontosaurus regalis Lambe, 1917 & Hypsilophodon foxii Huxley, 1869)	Figure 50 of Norman (2015)	
Coronosauria	Sereno, 1986	minimum-clade	min ∇ (Protoceratops andrewsi Granger & Gregory, 1923 & Triceratops horridus Marsh, 1889)	Figure 10 of Morschhauser et al. (2019)	
Corythosauria	New	minimum-clade	min ∇ (Corythosaurus casuarius Brown, 1914a & Lambeosaurus lambei Parks, 1923 & Parasaurolophus walkeri Parks, 1922)	Figure 18 of Prieto-Márquez, Wagner & Lehman (2020)	
Dryomorpha	Sereno, 1986	minimum-clade	min ∇ (Dryosaurus altus (Marsh, 1878) & Iguanodon bernissartensis Boulenger in Beneden, 1881)	Figure 12 of Madzia, Jagt & Mulder (2020)	
Dryosauridae	Milner & Norman, 1984	maximum-clade	max ∇ (Dryosaurus altus (Marsh, 1878) ~ Iguanodon bernissartensis Boulenger in Beneden, 1881)	Figure 12 of Madzia, Jagt & Mulder (2020)	
Edmontosaurini	Glut, 1997	maximum-clade	max ∇ (Edmontosaurus regalis Lambe, 1917 ~ Brachylophosaurus canadensis Sternberg, 1953 & Hadrosaurus foulkii Leidy, 1858 & Kritosaurus navajovius Brown, 1910 & Saurolophus osborni Brown, 1912)	Figure 18 of Prieto-Márquez, Wagner & Lehman (2020)	
Elasmaria	Calvo, Porfiri & Novas, 2007	minimum-clade	min ∇ (Macrogryphosaurus gondwanicus Calvo, Porfiri & Novas, 2007 & Talenkauen santacrucensis Novas, Cambiaso & Ambrosio, 2004 | ~ Hypsilophodon foxii Huxley, 1869 ∨ Iguanodon bernissartensis Boulenger in Beneden, 1881 ∨ Thescelosaurus neglectus Gilmore, 1913)	Figure 31 of Rozadilla, Agnolín & Novas (2019)	
Eucentrosaura	Chiba et al., 2018	minimum-clade	min ∇ (Centrosaurus apertus Lambe, 1905 & Pachyrhinosaurus canadensis Sternberg, 1950)	Figure 9 of Chiba et al. (2018)	
Euceratopsia	New	minimum-clade	min ∇ (Leptoceratops gracilis Brown, 1914b & Protoceratops andrewsi Granger & Gregory, 1923 & Triceratops horridus Marsh, 1889)	Figure 4 of Yu et al. (2020)	
Euhadrosauria	Weishampel, Norman & Grigorescu, 1993	minimum-clade	min ∇ (Lambeosaurus lambei Parks, 1923 & Saurolophus osborni Brown, 1912 | ~ Hadrosaurus foulkii Leidy, 1858)	Figure 18 of Prieto-Márquez, Wagner & Lehman (2020)	
Euiguanodontia	Coria & Salgado, 1996	minimum-clade	min ∇ (Camptosaurus dispar (Marsh, 1879) & Dryosaurus altus (Marsh, 1878) & Gasparinisaura cincosaltensis Coria & Salgado, 1996 | ~ Tenontosaurus tilletti Ostrom, 1970)	Figure 13 of Coria & Salgado (1996)	
Euornithopoda	Sereno, 1986	maximum-clade	max ∇ ∈ Ornithopoda (Iguanodon bernissartensis Boulenger in Beneden, 1881 ~ Heterodontosaurus tucki Crompton & Charig, 1962)	Figure 1 of Sereno (1999)	
Eurypoda	Sereno, 1986	minimum-clade	min ∇ (Ankylosaurus magniventris Brown, 1908 & Stegosaurus stenops Marsh, 1887)	Figure 3 of Thompson et al. (2012)	
Genasauria	Sereno, 1986	minimum-clade	min ∇ (Ankylosaurus magniventris Brown, 1908 & Iguanodon bernissartensis Boulenger in Beneden, 1881 & Stegosaurus stenops Marsh, 1887 & Triceratops horridus Marsh, 1889)	Figure 16 of Han et al. (2018)	
Hadrosauridae	Cope, 1869	minimum-clade	min ∇ (Hadrosaurus foulkii Leidy, 1858 & Lambeosaurus lambei Parks, 1923 & Saurolophus osborni Brown, 1912)	Figure 18 of Prieto-Márquez, Wagner & Lehman (2020)	
Hadrosauriformes	Sereno, 1997	minimum-clade	min ∇ (Hadrosaurus foulkii Leidy, 1858 & Iguanodon bernissartensis Boulenger in Beneden, 1881)	Figure 12 of Madzia, Jagt & Mulder (2020)	
Hadrosaurinae	Lambe, 1918	maximum-clade	max ∇ (Hadrosaurus foulkii Leidy, 1858 ~ Lambeosaurus lambei Parks, 1923)	Figure 5 of Kobayashi et al. (2019)	
Hadrosauroidea	von Huene, 1952	maximum-clade	max ∇ (Hadrosaurus foulkii Leidy, 1858 ~ Iguanodon bernissartensis Boulenger in Beneden, 1881)	Figure 12 of Madzia, Jagt & Mulder (2020)	
Hadrosauromorpha	Norman, 2014	maximum-clade	max ∇ (Hadrosaurus foulkii Leidy, 1858 ~ Probactrosaurus gobiensis Rozhdestvensky, 1966)	Figure 12 of Madzia, Jagt & Mulder (2020)	
Heterodontosauridae	Kuhn, 1966	maximum-clade	max ∇ (Heterodontosaurus tucki Crompton & Charig, 1962 ~ Iguanodon bernissartensis Boulenger in Beneden, 1881 & Pachycephalosaurus wyomingensis (Gilmore, 1931) & Stegosaurus stenops Marsh, 1887 & Triceratops horridus Marsh, 1889)	Figure 4 of Madzia, Boyd & Mazuch (2018)	
Huayangosauridae	Dong, Tang & Zhou, 1982	maximum-clade	max ∇ (Huayangosaurus taibaii Dong, Tang & Zhou, 1982 ~ Stegosaurus stenops Marsh, 1887)	Figure 12 of Maidment et al. (2020)	
Hypsilophodontia	Cooper, 1985	minimum-clade	min ∇ ∈ Ornithopoda (Hypsilophodon foxii Huxley, 1869 & Tenontosaurus tilletti Ostrom, 1970 | ~ Iguanodon bernissartensis Boulenger in Beneden, 1881)	Figure 50 of Norman (2015)	
Hypsilophodontidae	Dollo, 1882	maximum-clade	max ∇ (Hypsilophodon foxii Huxley, 1869 ~ Iguanodon bernissartensis Boulenger in Beneden, 1881 & Rhabdodon priscus Matheron, 1869)	Figure 2 of Dieudonné et al. (2020)	
Iguanodontia	Baur, 1891	minimum-clade	min ∇ (Dryosaurus altus (Marsh, 1878) & Iguanodon bernissartensis Boulenger in Beneden, 1881 & Rhabdodon priscus Matheron, 1869 & Tenontosaurus tilletti Ostrom, 1970 | ~ Hypsilophodon foxii Huxley, 1869)	Figure 12 of Madzia, Jagt & Mulder (2020)	
Iguanodontidae	Bonaparte, 1850	maximum-clade	max ∇ (Iguanodon bernissartensis Boulenger in Beneden, 1881 ~ Hadrosaurus foulkii Leidy, 1858)	Figure 13 of Madzia, Jagt & Mulder (2020)	
Jeholosauridae	Han et al., 2012	maximum-clade	max ∇ ∉ Hypsilophodontidae ∨ Thescelosauridae (Jeholosaurus shangyuanensis Xu, Wang & You, 2000 ~ Hypsilophodon foxii Huxley, 1869 & Iguanodon bernissartensis Boulenger in Beneden, 1881 & Pachycephalosaurus wyomingensis (Gilmore, 1931) & Thescelosaurus neglectus Gilmore, 1913 & Triceratops horridus Marsh, 1889)	Figure 25 of Herne et al. (2019)	
Kritosaurini	Glut, 1997	maximum-clade	max ∇ (Kritosaurus navajovius Brown, 1910 ~ Brachylophosaurus canadensis Sternberg, 1953 & Edmontosaurus regalis Lambe, 1917 & Hadrosaurus foulkii Leidy, 1858 & Saurolophus osborni Brown, 1912)	Figure 18 of Prieto-Márquez, Wagner & Lehman (2020)	
Lambeosaurinae	Parks, 1923	maximum-clade	max ∇ (Lambeosaurus lambei Parks, 1923 ~ Hadrosaurus foulkii Leidy, 1858 & Saurolophus osborni Brown, 1912)	Figure 18 of Prieto-Márquez, Wagner & Lehman (2020)	
Lambeosaurini	Sullivan et al., 2011	maximum-clade	max ∇ (Lambeosaurus lambei Parks, 1923 ~ Aralosaurus tuberiferus Rozhdestvensky, 1968 & Parasaurolophus walkeri Parks, 1922 & Tsintaosaurus spinorhinus Young, 1958)	Figure 18 of Prieto-Márquez, Wagner & Lehman (2020)	
Leptoceratopsidae	Nopcsa, 1923	maximum-clade	max ∇ (Leptoceratops gracilis Brown, 1914b ~ Protoceratops andrewsi Granger & Gregory, 1923 & Triceratops horridus Marsh, 1889)	Figure 10 of Morschhauser et al. (2019)	
Marginocephalia	Sereno, 1986	minimum-clade	min ∇ (Ceratops montanus Marsh, 1888 & Pachycephalosaurus wyomingensis (Gilmore, 1931) & Triceratops horridus Marsh, 1889)	Figure 16 of Han et al. (2018)	
Nasutoceratopsini	Ryan et al., 2017	maximum-clade	max ∇ (Nasutoceratops titusi Sampson et al., 2013 ~ Centrosaurus apertus Lambe, 1905)	Figure 9 of Chiba et al. (2018)	
Neoceratopsia	Sereno, 1986	maximum-clade	max ∇ (Triceratops horridus Marsh, 1889 ~ Chaoyangsaurus youngi Zhao, Cheng & Xu, 1999 & Psittacosaurus mongoliensis Osborn, 1923)	Figure 10 of Morschhauser et al. (2019)	
Neoiguanodontia	Norman, 2014	minimum-clade	min ∇ (Hypselospinus fittoni (Lydekker, 1889) & Iguanodon bernissartensis Boulenger in Beneden, 1881 & Parasaurolophus walkeri Parks, 1922)	Figure 2.26 of Norman (2014)	
Neornithischia	Cooper, 1985	maximum-clade	max ∇ (Iguanodon bernissartensis Boulenger in Beneden, 1881 & Triceratops horridus Marsh, 1889 ~ Ankylosaurus magniventris Brown, 1908 & Stegosaurus stenops Marsh, 1887)	Figure 4 of Madzia, Boyd & Mazuch (2018)	
Nodosauridae	Marsh, 1890	maximum-clade	max ∇ (Nodosaurus textilis Marsh, 1889 ~ Ankylosaurus magniventris Brown, 1908)	Figure 5 of Rivera-Sylva et al. (2018a)	
Nodosaurinae	Abel, 1919	maximum-clade	max ∇ (Nodosaurus textilis Marsh, 1889 ~ Hylaeosaurus armatus Mantell, 1833 & Mymoorapelta maysi Kirkland & Carpenter, 1994 & Polacanthus foxii Owen in Anonymous, 1865)	Figure 5 of Rivera-Sylva et al. (2018a)	
Ornithischia	Seeley, 1888	maximum-clade	max ∇ (Iguanodon bernissartensis Boulenger in Beneden, 1881 ~ Allosaurus fragilis Marsh, 1877a & Camarasaurus supremus Cope, 1877)	Figure 4 of Madzia, Boyd & Mazuch (2018)	
Ornithopoda	Marsh, 1881	maximum-clade	max ∇ (Iguanodon bernissartensis Boulenger in Beneden, 1881 ~ Pachycephalosaurus wyomingensis (Gilmore, 1931) & Triceratops horridus Marsh, 1889)	Figure 4 of Madzia, Boyd & Mazuch (2018)	
Orodrominae	Brown et al., 2013	maximum-clade	max ∇ ∈ Hypsilophodontidae ∨ Thescelosauridae (Orodromeus makelai Horner & Weishampel, 1988 ~ Hypsilophodon foxii Huxley, 1869 & Thescelosaurus neglectus Gilmore, 1913)	Figure 4 of Madzia, Boyd & Mazuch (2018)	
Pachycephalosauria	Maryańska & Osmólska, 1974	maximum-clade	max ∇ (Pachycephalosaurus wyomingensis (Gilmore, 1931) ~ Ceratops montanus Marsh, 1888 & Triceratops horridus Marsh, 1889)	Figure 27 of Schott & Evans (2017)	
Pachycephalosauridae	Sternberg, 1945	minimum-clade	min ∇ (Pachycephalosaurus wyomingensis (Gilmore, 1931) & Stegoceras validum Lambe, 1902 | ~ Heterodontosaurus tucki Crompton & Charig, 1962)	Figure 27 of Schott & Evans (2017)	
Pachycephalosaurinae	Sereno, 1997	maximum-clade	max ∇ (Pachycephalosaurus wyomingensis (Gilmore, 1931) ~ Stegoceras validum Lambe, 1902)	Figure 27 of Schott & Evans (2017)	
Pachycephalosaurini	Sullivan, 2003	maximum-clade	max ∇ (Pachycephalosaurus wyomingensis (Gilmore, 1931) ~ Prenocephale prenes Maryańska & Osmólska, 1974 & Sphaerotholus goodwini Williamson & Carr, 2003)	Figure 27 of Schott & Evans (2017)	
Pachyrhinosaurini	Fiorillo & Tykoski, 2012	maximum-clade	max ∇ (Pachyrhinosaurus canadensis Sternberg, 1950 ~ Centrosaurus apertus Lambe, 1905)	Figure 9 of Chiba et al. (2018)	
Pachyrostra	Fiorillo & Tykoski, 2012	minimum-clade	min ∇ (Achelousaurus horneri Sampson, 1995 & Pachyrhinosaurus canadensis Sternberg, 1950)	Figure 9 of Chiba et al. (2018)	
Panoplosaurini	New	maximum-clade	max ∇ (Panoplosaurus mirus Lambe, 1919 ~ Nodosaurus textilis Marsh, 1889 & Struthiosaurus austriacus Bunzel, 1871)	Figure 5 of Rivera-Sylva et al. (2018a)	
Parasaurolophini	Glut, 1997	maximum-clade	max ∇ (Parasaurolophus walkeri Parks, 1922 ~ Aralosaurus tuberiferus Rozhdestvensky, 1968 & Lambeosaurus lambei Parks, 1923 & Tsintaosaurus spinorhinus Young, 1958)	Figure 18 of Prieto-Márquez, Wagner & Lehman (2020)	
Polacanthinae	Lapparent & Lavocat, 1955	maximum-clade	max ∇ ∈ Ankylosauridae ∨ Nodosauridae (Polacanthus foxii Owen in Anonymous, 1865 ~ Ankylosaurus magniventris Brown, 1908 & Nodosaurus textilis Marsh, 1889)	Figure 9 of Yang et al. (2013)	
Protoceratopsidae	Granger & Gregory, 1923	maximum-clade	max ∇ (Protoceratops andrewsi Granger & Gregory, 1923 ~ Ceratops montanus Marsh, 1888 & Leptoceratops gracilis Brown, 1914b & Triceratops horridus Marsh, 1889)	Figure 10 of Morschhauser et al. (2019)	
Rhabdodontidae	Weishampel et al., 2003	minimum-clade	min ∇ (Rhabdodon priscus Matheron, 1869 & Zalmoxes robustus (Nopcsa, 1900))	Figure 4 of Madzia, Boyd & Mazuch (2018)	
Rhabdodontomorpha	Dieudonné et al., 2016	maximum-clade	max ∇ (Rhabdodon priscus Matheron, 1869 ~ Hypsilophodon foxii Huxley, 1869 & Iguanodon bernissartensis Boulenger in Beneden, 1881)	Figure 2 of Dieudonné et al. (2020)	
Saphornithischia	New	minimum-clade	min ∇ (Heterodontosaurus tucki Crompton & Charig, 1962 & Iguanodon bernissartensis Boulenger in Beneden, 1881 & Stegosaurus stenops Marsh, 1887 & Triceratops horridus Marsh, 1889)	Figure 4 of Madzia, Boyd & Mazuch (2018)	
Saurolophinae	Brown, 1914a	maximum-clade	max ∇ (Saurolophus osborni Brown, 1912 ~ Lambeosaurus lambei Parks, 1923 | ~ Hadrosaurus foulkii Leidy, 1858)	Figure 18 of Prieto-Márquez, Wagner & Lehman (2020)	
Saurolophini	Glut, 1997	maximum-clade	max ∇ (Saurolophus osborni Brown, 1912 ~ Brachylophosaurus canadensis Sternberg, 1953 & Edmontosaurus regalis Lambe, 1917 & Hadrosaurus foulkii Leidy, 1858 & Kritosaurus navajovius Brown, 1910)	Figure 18 of Prieto-Márquez, Wagner & Lehman (2020)	
Shamosaurinae	Tumanova, 1983	maximum-clade	max ∇ (Gobisaurus domoculus Vickaryous et al., 2001 & Shamosaurus scutatus Tumanova, 1983 ~ Ankylosaurus magniventris Brown, 1908)	Figure 11 of Arbour & Currie (2016)	
Stegosauria	Marsh, 1877b	maximum-clade	max ∇ (Stegosaurus stenops Marsh, 1887 ~ Ankylosaurus magniventris Brown, 1908)	Figure 12 of Maidment et al. (2020)	
Stegosauridae	Marsh, 1880	maximum-clade	max ∇ (Stegosaurus stenops Marsh, 1887 ~ Huayangosaurus taibaii Dong, Tang & Zhou, 1982)	Figure 12 of Maidment et al. (2020)	
Struthiosaurini	New	maximum-clade	max ∇ (Struthiosaurus austriacus Bunzel, 1871 ~ Nodosaurus textilis Marsh, 1889 & Panoplosaurus mirus Lambe, 1919)	Figure 5 of Rivera-Sylva et al. (2018a)	
Styracosterna	Sereno, 1986	maximum-clade	max ∇ (Iguanodon bernissartensis Boulenger in Beneden, 1881 ~ Camptosaurus dispar (Marsh, 1879))	Figure 12 of Madzia, Jagt & Mulder (2020)	
Thescelosauridae	Sternberg, 1937	maximum-clade	max ∇ (Thescelosaurus neglectus Gilmore, 1913 ~ Iguanodon bernissartensis Boulenger in Beneden, 1881 | ~ Hypsilophodon foxii Huxley, 1869)	Figure 4 of Madzia, Boyd & Mazuch (2018)	
Thescelosaurinae	Sternberg, 1940	maximum-clade	max ∇ ∈ Hypsilophodontidae ∨ Thescelosauridae (Thescelosaurus neglectus Gilmore, 1913 ~ Hypsilophodon foxii Huxley, 1869 & Orodromeus makelai Horner & Weishampel, 1988)	Figure 4 of Madzia, Boyd & Mazuch (2018)	
Thyreophora	Nopcsa, 1915	maximum-clade	max ∇ (Ankylosaurus magniventris Brown, 1908 & Stegosaurus stenops Marsh, 1887 ~ Iguanodon bernissartensis Boulenger in Beneden, 1881 & Triceratops horridus Marsh, 1889)	Figure 16 of Han et al. (2018)	
Triceratopsini	Longrich, 2011	maximum-clade	max ∇ (Triceratops horridus Marsh, 1889 ~ Anchiceratops ornatus Brown, 1914c & Arrhinoceratops brachyops Parks, 1925)	Figure 9a of Fowler & Freedman Fowler (2020)	
Tsintaosaurini	Prieto-Márquez et al., 2013	maximum-clade	max ∇ (Pararhabdodon isonensis Casanovas-Cladellas, Santafé-Llopis & Isidro-Llorens, 1993 & Tsintaosaurus spinorhinus Young, 1958 ~ Aralosaurus tuberiferus Rozhdestvensky, 1968 & Lambeosaurus lambei Parks, 1923 & Parasaurolophus walkeri Parks, 1922)	Figure 18 of Prieto-Márquez, Wagner & Lehman (2020)	

Ankylopollexia Sereno, 1986 (converted clade name)

Registration number: 585

Definition. The smallest clade containing Camptosaurus dispar (Marsh, 1879) and Iguanodon bernissartensis Boulenger in Beneden, 1881. This is a minimum-clade definition. Abbreviated definition: min ∇ (Camptosaurus dispar (Marsh, 1879) & Iguanodon bernissartensis Boulenger in Beneden, 1881).

Reference phylogeny. Figure 12 of Madzia, Jagt & Mulder (2020) is treated here as the primary reference phylogeny. Additional reference phylogenies include Figure 3 of Madzia, Boyd & Mazuch (2018), Figure 20 of Verdú et al. (2018), Figure 9 of Verdú et al. (2020), Figure 11 of McDonald et al. (2021), and Figure 11 of Santos-Cubedo et al. (2021).

Composition. The clade Ankylopollexia comprises Camptosaurus dispar and members of the clade Styracosterna.

Synonyms. No other taxon names are currently in use for the same or approximate clade.

Comments. The name Ankylopollexia has been (informally) defined before by Sereno (1998: 62) who applied the minimum-clade definition and used Camptosaurus and Parasaurolophus as the internal specifiers. Since the name has traditionally been used in the exact sense, we apply it to the same clade, but prefer to use Iguanodon bernissartensis as the second internal specifier rather than P. walkeri because the name Ankylopollexia was formed after the stiff cone-shaped thumb that characterizes Iguanodon-grade ornithopods. The inclusion of a different internal specifier does not change the extent of Ankylopollexia under any of the published phylogeny inferences. Also, even though the name derives from an apomorphy, it was never used for an apomorphy-based clade.

Ankylosauria Osborn, 1923 (converted clade name)

Registration number: 588

Definition. The largest clade containing Ankylosaurus magniventris Brown, 1908 but not Stegosaurus stenops Marsh, 1887. This is a maximum-clade definition. Abbreviated definition: max ∇ (Ankylosaurus magniventris Brown, 1908 ~ Stegosaurus stenops Marsh, 1887).

Reference phylogeny. Figure 11 of Arbour & Currie (2016) is treated here as the primary reference phylogeny. Additional reference phylogenies include Figure 3 of Thompson et al. (2012), Figure 1 of Arbour, Zanno & Gates (2016), Figure 3 of Brown et al. (2017), and Figure 26 of Wiersma & Irmis (2018).

Composition. Under the primary reference phylogeny, Ankylosauria comprises Minmi sp. (= Kunbarrasaurus ieversi), Mymoorapelta maysi, and members of the clades Ankylosauridae and Nodosauridae.

Synonyms. The name Ankylosauromorpha Carpenter, 2001 has been recently used under an alternative systematic scheme for the same branch as Ankylosauria, as defined herein (Norman, 2021; see ‘Discussion’). No other taxon names are currently in use for the same or approximate clade.

Comments. The name Ankylosauria has been (informally) defined before (Carpenter, 1997; Sereno, 1998; Sereno, 2005). These definitions were maximum-clade and used Ankylosaurus (Carpenter, 1997; Sereno, 1998) or Ankylosaurus magniventris (Sereno, 2005) as the internal specifier and Stegosaurus (Carpenter, 1997; Sereno, 1998) or Stegosaurus stenops (Sereno, 2005) as the external specifier. Since Ankylosauria has been ‘traditionally’ used in this sense (though, see also ‘Discussion’), we formalize this definition. Note that Norman (2021) recently provided two phylogenetic definitions for Ankylosauria, a maximum-clade and a minimum-clade. In the maximum-clade definition Norman (2021) used Euoplocephalus and Edmontonia as the internal specifiers and Scelidosaurus as the external specifier, while in the minimum-clade definition the use of the name was anchored on Euoplocephalus and Edmontonia. See ‘Discussion’ for additional comments. Note that the external specifier Stegosaurus stenops is not included in the primary reference phylogeny. From the taxa analyzed by Arbour & Currie (2016), S. stenops is most closely related to Huayangosaurus taibaii (see, e.g., Maidment et al., 2020).

Ankylosauridae Brown, 1908 (converted clade name)

Registration number: 589

Definition. The largest clade containing Ankylosaurus magniventris Brown, 1908 but not Nodosaurus textilis Marsh, 1889. This is a maximum-clade definition. Abbreviated definition: max ∇ (Ankylosaurus magniventris Brown, 1908 ~ Nodosaurus textilis Marsh, 1889).

Reference phylogeny. Figure 11 of Arbour & Currie (2016) is treated here as the primary reference phylogeny. Additional reference phylogenies include Figure 3 of Thompson et al. (2012), Figure 1 of Arbour, Zanno & Gates (2016), Figure 3 of Brown et al. (2017), Figure 26 of Wiersma & Irmis (2018), and Figure 9 of Zheng et al. (2018).

Composition. Under the primary reference phylogeny, Ankylosauridae comprises Ahshislepelta minor, Aletopelta coombsi, Cedarpelta bilbeyhallorum, Chuanqilong chaoyangensis, Gastonia burgei, Liaoningosaurus paradoxus, and members of the clades Shamosaurinae and Ankylosaurinae.

Synonyms. No other taxon names are currently in use for the same or approximate clade.

Comments. The name Ankylosauridae has been (informally) defined before by Sereno (1998, 2005) who applied a maximum-clade definition and used Ankylosaurus magniventris as the internal specifier and Panoplosaurus mirus as the external specifier. Considering that Ankylosauridae has been traditionally used as a sister taxon to Nodosauridae (see, e.g., Thompson et al., 2012 for details), we use a definition that incorporates Nodosaurus textilis as the external specifier. Note that N. textilis is not included in the primary reference phylogeny. Both, A. magniventris and N. textilis were analyzed by, and their relationship is indicated in, Rivera-Sylva et al. (2018a).

Ankylosaurinae Nopcsa, 1918 (converted clade name)

Registration number: 590

Definition. The largest clade containing Ankylosaurus magniventris Brown, 1908 but not Shamosaurus scutatus Tumanova, 1983. This is a maximum-clade definition. Abbreviated definition: max ∇ (Ankylosaurus magniventris Brown, 1908 ~ Shamosaurus scutatus Tumanova, 1983).

Reference phylogeny. Figure 11 of Arbour & Currie (2016) is treated here as the primary reference phylogeny. Additional reference phylogenies include Figure 3 of Thompson et al. (2012), Figure 1 of Arbour, Zanno & Gates (2016), Figure 8 of Arbour & Evans (2017), Figure 26 of Wiersma & Irmis (2018), and Figure 9 of Zheng et al. (2018).

Composition. Under the primary reference phylogeny, Ankylosaurinae comprises Crichtonpelta benxiensis, Pinacosaurus spp., Saichania chulsanensis, Tarchia kielanae, Tsagantegia longicranialis, Zaraapelta nomadis, ‘Zhejiangosaurus luoyangensis’, and members of the clade Ankylosaurini.

Synonyms. No other taxon names are currently in use for the same or approximate clade.

Comments. The name Ankylosaurinae was (informally) defined before (Sereno, 1998; Sereno, 2005; Vickaryous, Maryanska & Weishampel, 2004). All these definitions were maximum-clade and used Ankylosaurus (Sereno, 1998) or Ankylosaurus magniventris (Sereno, 2005; Vickaryous, Maryanska & Weishampel, 2004) as the internal specifiers and Minmi paravertebra and Shamosaurus scutatus (Sereno, 1998), Gargoyleosaurus parkpinorum, Minmi paravertebra, and Shamosaurus scutatus (Sereno, 2005) or only Shamosaurus scutatus (Vickaryous, Maryanska & Weishampel, 2004) as the external specifiers. Owing to the dubious taxonomic status of ‘M. paravertebra’ (Arbour & Currie, 2016) and non-ankylosaurid affinities of G. parkpinorum (e.g., Arbour & Currie, 2016; Rivera-Sylva et al., 2018a; Wiersma & Irmis, 2018; Zheng et al., 2018), we formalize the definition of Vickaryous, Maryanska & Weishampel (2004) in that we use a single external specifier (Shamosaurus scutatus).

Ankylosaurini Arbour & Currie, 2016 (converted clade name)

Registration number: 592

Definition. The largest clade containing Ankylosaurus magniventris Brown, 1908 but not Pinacosaurus grangeri Gilmore, 1933 and Saichania chulsanensis Maryańska, 1977. This is a maximum-clade definition. Abbreviated definition: max ∇ (Ankylosaurus magniventris Brown, 1908 ~ Pinacosaurus grangeri Gilmore, 1933 & Saichania chulsanensis Maryańska, 1977).

Reference phylogeny. Figure 11 of Arbour & Currie (2016) is treated here as the primary reference phylogeny. Additional reference phylogenies include Figure 1 of Arbour, Zanno & Gates (2016), Figure 8 of Arbour & Evans (2017), Figure 26 of Wiersma & Irmis (2018), and Figure 9 of Zheng et al. (2018).

Composition. Under the primary reference phylogeny, Ankylosaurini comprises Ankylosaurus magniventris, Anodontosaurus lambei, Dyoplosaurus acutosquameus, Euoplocephalus tutus, Nodocephalosaurus kirtlandensis, Scolosaurus cutleri, Talarurus plicatospineus, and Ziapelta sanjuanensis.

Synonyms. No other taxon names are currently in use for the same or approximate clade.

Comments. The name Ankylosaurini was first (informally) defined by Arbour & Currie (2016) who applied the maximum-clade definition and used Ankylosaurus magniventris as the internal specifier and Pinacosaurus grangeri and Saichania chulsanensis as the external specifiers. The name was used for a clade that largely includes later-diverging North American ankylosaurines, many of which were previously synonymized with Euoplocephalus tutus (Arbour & Currie, 2013), although under some topologies the name may be more restricted in its use (Thompson et al., 2012).

Aralosaurini Prieto-Márquez et al., 2013 (converted clade name)

Registration number: 593

Definition. The largest clade containing Aralosaurus tuberiferus Rozhdestvensky, 1968 and Canardia garonnensis Prieto-Márquez et al., 2013 but not Lambeosaurus lambei Parks, 1923, Parasaurolophus walkeri Parks, 1922, and Tsintaosaurus spinorhinus Young, 1958. This is a maximum-clade definition. Abbreviated definition: max ∇ (Aralosaurus tuberiferus Rozhdestvensky, 1968 & Canardia garonnensis Prieto-Márquez et al., 2013 ~ Lambeosaurus lambei Parks, 1923 & Parasaurolophus walkeri Parks, 1922 & Tsintaosaurus spinorhinus Young, 1958).

Reference phylogeny. Figure 25 of Prieto-Márquez et al. (2013) is treated here as the primary reference phylogeny. Additional reference phylogeny includes Figure 11 of McDonald et al. (2021).

Composition. Under the primary reference phylogeny, Aralosaurini comprises Aralosaurus tuberiferus and Canardia garonnensis.

Synonyms. No other taxon names are currently in use for the same or approximate clade.

Comments. The name was first (informally) defined by Prieto-Márquez et al. (2013) who applied the minimum-clade definition and used Aralosaurus tuberiferus and Canardia garonnensis as the internal specifiers. Following such definition, however, Aralosaurini would cover the entire lambeosaurine branch under some topologies that include both of the internal specifiers (Kobayashi et al., 2019; Prieto-Márquez et al., 2019; Zhang et al., 2019; Gates, Evans & Sertich, 2021; Kobayashi et al., 2021; Longrich et al., 2021), or would even comprise the same contents as Euhadrosauria (Ramírez-Velasco et al., 2021). Recently, however, McDonald et al. (2021) inferred Aralosaurini as delimited by Prieto-Márquez et al. (2013). Therefore, we define the name but make it inapplicable under a subset of recent phylogenies.

Brachylophosaurini Gates et al., 2011 (converted clade name)

Registration number: 594

Definition. The largest clade containing Brachylophosaurus canadensis Sternberg, 1953 but not Edmontosaurus regalis Lambe, 1917, Hadrosaurus foulkii Leidy, 1858, Kritosaurus navajovius Brown, 1910, and Saurolophus osborni Brown, 1912. This is a maximum-clade definition. Abbreviated definition: max ∇ (Brachylophosaurus canadensis Sternberg, 1953 ~ Edmontosaurus regalis Lambe, 1917 & Hadrosaurus foulkii Leidy, 1858 & Kritosaurus navajovius Brown, 1910 & Saurolophus osborni Brown, 1912).

Reference phylogeny. Figure 18 of Prieto-Márquez, Wagner & Lehman (2020) is treated here as the primary reference phylogeny. Additional reference phylogenies include Figure 5 of Kobayashi et al. (2019), Figure 11 of Prieto-Márquez et al. (2019), Figure 9 of Zhang et al. (2019), Figure 5 of Zhang et al. (2020), Figure 7 of Kobayashi et al. (2021), and Figure 10 of Longrich et al. (2021).

Composition. Under the primary reference phylogeny, Brachylophosaurini comprises Acristavus gagslarsoni, Brachylophosaurus canadensis, Maiasaura peeblesorum, and Probrachylophosaurus bergei (erroneously named ‘Probrachylophosaurus canadensis’ in the primary reference phylogeny).

Synonyms. The name Maiasaurini Sereno, 2005 is an approximate synonym of Brachylophosaurini. To our knowledge, the name was used only in two recent papers (McFeeters et al., 2021; McFeeters, Evans & Maddin, 2021) that attributed the name to Horner (1992). However, this attribution was due to the adherence of the authors to the Principle of Coordination, as Horner (1992) used the name Maiasaurinae. Nevertheless, all recent phylogenetic studies consistently use Brachylophosaurini (e.g., Freedman Fowler & Horner, 2015; Cruzado-Caballero & Powell, 2017; Xing, Mallon & Currie, 2017; Kobayashi et al., 2019; Zhang et al., 2019; Prieto-Márquez, Wagner & Lehman, 2020; Zhang et al., 2020; Kobayashi et al., 2021; McDonald et al., 2021). No other taxon names are currently in use for the same or approximate clade.

Comments. The name Brachylophosaurini has been (informally) defined before (Gates et al., 2011; Freedman Fowler & Horner, 2015). These definitions were maximum-clade and used Brachylophosaurus, Maiasaura, and Acristavus (Gates et al., 2011) or Brachylophosaurus, Probrachylophosaurus, Maiasaura, and Acristavus (Freedman Fowler & Horner, 2015) as the internal specifiers and Gryposaurus and Saurolophus as the external specifiers. The composition of Brachylophosaurini and the relationships of the clade to other hadrosaurids have been stable across studies since the introduction of the name. Therefore, using more than one internal specifier is unnecessary. We use a definition that ensures Brachylophosaurini does not cover taxa ‘traditionally’ comprised within Edmontosaurini, Kritosaurini, and Saurolophini.

Camptosauridae Marsh, 1885 (converted clade name)

Registration number: 595

Definition. The largest clade containing Camptosaurus dispar (Marsh, 1879) but not Iguanodon bernissartensis Boulenger in Beneden, 1881. This is a maximum-clade definition. Abbreviated definition: max ∇ (Camptosaurus dispar (Marsh, 1879) ~ Iguanodon bernissartensis Boulenger in Beneden, 1881).

Reference phylogeny. Figure 13 of Madzia, Jagt & Mulder (2020) is treated here as the primary reference phylogeny. Additional reference phylogenies include Figure 20 of Verdú et al. (2018), Figure 11 of Santos-Cubedo et al. (2021), and Figure 9 of Verdú et al. (2020).

Composition. Under the primary reference phylogeny, Camptosauridae comprises Camptosaurus dispar and Cumnoria prestwichii. Under alternative hypotheses, however, Camptosauridae includes only a single unequivocal member, Camptosaurus dispar (e.g., Madzia, Jagt & Mulder, 2020: Fig. 12).

Synonyms. No other taxon names are currently in use for the same or approximate clade.

Comments. The name Camptosauridae was first (informally) defined by Sereno (1998: 62) who used the maximum-clade definition and selected Camptosaurus as the internal specifier and Parasaurolophus as the external specifier. We prefer to use Iguanodon bernissartensis as the external specifier to maintain the ‘node-branch triplet’ (‘node-stem triplet’ of Sereno (1998: 52–54)) comprising Ankylopollexia, Camptosauridae, and Styracosterna (all formally defined in the present paper). The inclusion of a different external specifier does not change the extent of Camptosauridae under any of the published phylogeny inferences.

Centrosaurinae Lambe, 1915 (converted clade name)

Registration number: 596

Definition. The largest clade containing Centrosaurus apertus Lambe, 1905 but not Chasmosaurus belli (Lambe, 1902) and Triceratops horridus Marsh, 1889. This is a maximum-clade definition. Abbreviated definition: max ∇ (Centrosaurus apertus Lambe, 1905 ~ Chasmosaurus belli (Lambe, 1902) & Triceratops horridus Marsh, 1889).

Reference phylogeny. Figure 9 of Chiba et al. (2018) is treated here as the primary reference phylogeny. Additional reference phylogenies include Figure 10 of Ryan et al. (2017), Figure 13 of Dalman et al. (2018), Figure 10 of Wilson, Ryan & Evans (2020), Figure 4 of Yu et al. (2020), and Figure 23 of Dalman et al. (2021).

Composition. Under the primary reference phylogeny, Centrosaurinae comprises Albertaceratops nesmoi, Diabloceratops eatoni, Machairoceratops cronusi, Medusaceratops lokii, Sinoceratops zhuchengensis, Wendiceratops pinhornensis, Xenoceratops foremostensis, and members of the clades Eucentrosaura and Nasutoceratopsini.

Synonyms. No other taxon names are currently in use for the same or approximate clade. Although Ceratops montanus may fall within the largest clade containing Centrosaurus apertus but not Chasmosaurus belli and Triceratops horridus as well, the name Ceratopsinae Abel, 1919 has not been associated with the same contents as Centrosaurinae in the past. Therefore, Ceratopsinae is not considered to be an approximate synonym of Centrosaurinae. In any case, C. montanus does not seem to be diagnostic beyond Ceratopsidae at present (Dodson, Forster & Sampson, 2004; Mallon et al., 2016). Therefore, its position within the clade is uncertain. Lucas et al. (2016: 202) have argued that Pachyrhinosaurinae von Huene, 1950 has priority over Centrosaurinae under the Article 61 of the ICZN (International Commission on Zoological Nomenclature, 1999). However, the name Pachyrhinosaurinae has not been used in the literature recently and even Lucas et al. (2016) used Centrosaurinae for the clade in question.

Comments. The name Centrosaurinae has been (informally) defined before (Sereno, 1998; Dodson, Forster & Sampson, 2004; Sereno, 2005). These definitions were maximum-clade and used Pachyrhinosaurus (Sereno, 1998), Centrosaurus (Dodson, Forster & Sampson, 2004), or Centrosaurus apertus (Sereno, 2005) as the internal specifier and Triceratops (Sereno, 1998; Dodson, Forster & Sampson, 2004) or Triceratops horridus (Sereno, 2005) as the external specifier. We apply the name Centrosaurinae for the same known contents; adopting the mandatory Centrosaurus apertus as the internal specifier and Chasmosaurus belli and Triceratops horridus as the external specifiers.

Centrosaurini Ryan et al., 2017 (converted clade name)

Registration number: 687

Definition. The largest clade containing Centrosaurus apertus Lambe, 1905 but not Pachyrhinosaurus canadensis Sternberg, 1950. This is a maximum-clade definition. Abbreviated definition: max ∇ (Centrosaurus apertus Lambe, 1905 ~ Pachyrhinosaurus canadensis Sternberg, 1950).

Reference phylogeny. Figure 9 of Chiba et al. (2018) is treated here as the primary reference phylogeny. Additional reference phylogenies include Figure 7 of Fiorillo & Tykoski (2012), Figure 10 of Ryan et al. (2017), Figure 13 of Dalman et al. (2018), and Figure 23 of Dalman et al. (2021).

Composition. Under the primary reference phylogeny, Centrosaurini comprises Centrosaurus apertus, Coronosaurus brinkmani, Rubeosaurus ovatus (?= Styracosaurus albertensis; see Holmes et al., 2020), Spinops sternbergorum, and Styracosaurus albertensis. Under an alternative hypothesis, Centrosaurini includes only a single unequivocal member, Centrosaurus apertus (Wilson, Ryan & Evans, 2020: Fig. 10). However, a Bayesian analysis of the same matrix and published in the same study reconstructed Centrosaurini to comprise Centrosaurus apertus, Coronosaurus brinkmani, and Spinops sternbergorum (Wilson, Ryan & Evans, 2020: Fig. 9).

Synonyms. No other taxon names are currently in use for the same or approximate clade.

Comments. The name was first (informally) defined by Ryan et al. (2017) who applied the maximum-clade definition and used Centrosaurus apertus as the internal specifier and Pachyrhinosaurus canadensis as the external specifier. We formalize this definition.

Cerapoda Sereno, 1986 (converted clade name)

Registration number: 597

Definition. The smallest clade containing Iguanodon bernissartensis Boulenger in Beneden, 1881, Pachycephalosaurus wyomingensis (Gilmore, 1931), and Triceratops horridus Marsh, 1889. This is a minimum-clade definition. Abbreviated definition: min ∇ (Iguanodon bernissartensis Boulenger in Beneden, 1881 & Pachycephalosaurus wyomingensis (Gilmore, 1931) & Triceratops horridus Marsh, 1889).

Reference phylogeny. Figure 4 of Madzia, Boyd & Mazuch (2018) is treated here as the primary reference phylogeny. Additional reference phylogenies include Figure 16 of Han et al. (2018), Figure 25 of Herne et al. (2019), Figure 1 of Dieudonné et al. (2020), and Figure 57 of Barta & Norell (2021).

Composition. Under the primary reference phylogeny, Cerapoda comprises members of the clades Ornithopoda and Marginocephalia.

Synonyms. No other taxon names are currently in use for the same or approximate clade.

Comments. The name Cerapoda has been (informally) defined before (Weishampel, 2004; Butler, Upchurch & Norman, 2008). Both types of definitions, minimum-clade as well as maximum-clade, have been proposed for the name. Weishampel (2004) preferred a maximum-clade definition and used Triceratops as the internal specifier and Ankylosaurus as the external specifier, while Butler, Upchurch & Norman (2008) applied a minimum-clade definition, using Triceratops horridus and Parasaurolophus walkeri as the internal specifiers. Subsequent authors followed the latter definition (Boyd, 2015; Madzia, Boyd & Mazuch, 2018; Herne et al., 2019; Yang et al., 2020). We apply a minimum-clade definition as well and use Iguanodon bernissartensis, Pachycephalosaurus wyomingensis, and Triceratops horridus as the internal specifiers. Note that the internal specifiers Pachycephalosaurus wyomingensis and Triceratops horridus are not included in the primary reference phylogeny. The former belongs to Pachycephalosauria (see, e.g., Dieudonné et al., 2020), while the latter is part of Ceratopsia (e.g., Morschhauser et al., 2019), both within Marginocephalia that is indicated on Figure 4 of Madzia, Boyd & Mazuch (2018).

Ceratopsia Marsh, 1890 (converted clade name)

Registration number: 598

Definition. The largest clade containing Ceratops montanus Marsh, 1888 and Triceratops horridus Marsh, 1889 but not Pachycephalosaurus wyomingensis (Gilmore, 1931). This is a maximum-clade definition. Abbreviated definition: max ∇ (Ceratops montanus Marsh, 1888 & Triceratops horridus Marsh, 1889 ~ Pachycephalosaurus wyomingensis (Gilmore, 1931)).

Reference phylogeny. Figure 10 of Morschhauser et al. (2019) is treated here as the primary reference phylogeny. Additional reference phylogenies include Figure 16 of Han et al. (2018), Figure S1 of Knapp et al. (2018), Figure 1 of Dieudonné et al. (2020), Figure 3 of Yu et al. (2020), and Figure 4 of Yu et al. (2020).

Composition. Under the primary reference phylogeny, Ceratopsia comprises Psittacosaurus spp. and members of the clades Chaoyangsauridae and Neoceratopsia.

Synonyms. No other taxon names are currently in use for the same or approximate clade.

Comments. The name Ceratopsia has been (informally) defined before (Dodson, 1997; Sereno, 1998; Sereno, 2005). These definitions were maximum-clade and used Ceratopsidae (Dodson, 1997), Triceratops (Sereno, 1998), or Triceratops horridus (Sereno, 2005) as the internal specifiers and Pachycephalosauridae (Dodson, 1997), Pachycephalosaurus (Sereno, 1998), or Pachycephalosaurus wyomingensis, Heterodontosaurus tucki, Hypsilophodon foxii, and Ankylosaurus magniventris (Sereno, 2005) as the external specifiers. Even though the position of Hypsilophodon foxii and Heterodontosaurus tucki is indeed somewhat unstable across studies (see, e.g., Han et al., 2018; Madzia, Boyd & Mazuch, 2018; Herne et al., 2019; Dieudonné et al., 2020; Yang et al., 2020), inclusion of these taxa among the external specifiers is not necessary. We use a definition similar to that of Sereno (1998) but include the mandatory Ceratops montanus as a second internal specifier. Note that the internal specifier Ceratops montanus and the external specifier Pachycephalosaurus wyomingensis are not included in the primary reference phylogeny. The former belongs to Ceratopsidae (e.g., Mallon et al., 2016), while the latter is part of Pachycephalosauria (see, e.g., Dieudonné et al., 2020).

Ceratopsidae Marsh, 1888 (converted clade name)

Registration number: 599

Definition. The smallest clade containing Centrosaurus apertus Lambe, 1905, Ceratops montanus Marsh, 1888, Chasmosaurus belli (Lambe, 1902), and Triceratops horridus Marsh, 1889. This is a minimum-clade definition. Abbreviated definition: min ∇ (Centrosaurus apertus Lambe, 1905 & Ceratops montanus Marsh, 1888 & Chasmosaurus belli (Lambe, 1902) & Triceratops horridus Marsh, 1889).

Reference phylogeny. Figure 4 of Yu et al. (2020) is treated here as the primary reference phylogeny. Additional reference phylogenies include Figure 14 of Mallon et al. (2016), Figure S1 of Knapp et al. (2018), Figure 9a of Fowler & Freedman Fowler (2020), Figure 10 of Wilson, Ryan & Evans (2020), and Figure 3 of Yu et al. (2020).

Composition. Under the primary reference phylogeny, Ceratopsidae comprises members of the clades Centrosaurinae and Chasmosaurinae.

Synonyms. No other taxon names are currently in use for the same or approximate clade.

Comments. The name Ceratopsidae has been (informally) defined before (Sereno, 1998, Dodson, Forster & Sampson, 2004; Sereno, 2005). These definitions were minimum-clade and used Triceratops and Pachyrhinosaurus (Sereno, 1998), Triceratops and Centrosaurus (Dodson, Forster & Sampson, 2004), and Triceratops horridus and Pachyrhinosaurus canadensis (Sereno, 2005) as the internal specifiers. Considering that Ceratopsidae ‘traditionally’ contains two subclades, Centrosaurinae and Chasmosaurinae, we include the nomenclatural types of these clades, Centrosaurus apertus and Chasmosaurus belli, respectively, as the internal specifiers, and additionally add Triceratops horridus, a common specifier in the nomenclature of ceratopsian clades and the only taxon that has always been used as an internal specifier in the definition of Ceratopsidae. Finally, we also include a fourth internal specifier, the mandatory Ceratops montanus. Even though the taxon is considered a nomen dubium (e.g., Dodson, Forster & Sampson, 2004; Mallon et al., 2016), its placement within the smallest clade comprising centrosaurines and chasmosaurines does not appear to be questionable (see, e.g., Mallon et al., 2016).

Ceratopsoidea Hay, 1902 (converted clade name)

Registration number: 601

Definition. The largest clade containing Ceratops montanus Marsh, 1888 and Triceratops horridus Marsh, 1889 but not Protoceratops andrewsi Granger & Gregory, 1923. This is a maximum-clade definition. Abbreviated definition: max ∇ (Ceratops montanus Marsh, 1888 & Triceratops horridus Marsh, 1889 ~ Protoceratops andrewsi Granger & Gregory, 1923).

Reference phylogeny. Figure 4 of Yu et al. (2020) is treated here as the primary reference phylogeny. Additional reference phylogenies include Figure S1 of Knapp et al. (2018), Figure 10 of Morschhauser et al. (2019), and Figure 3 of Yu et al. (2020).

Composition. Under the primary reference phylogeny, Ceratopsoidea comprises Turanoceratops tardabilis, Zuniceratops christopheri, and members of the clade Ceratopsidae.

Synonyms. No other taxon names are currently in use for the same or approximate clade.

Comments. The name Ceratopsoidea has been (informally) defined before by Sereno (1998, 2005) who applied a maximum-clade definition and used Triceratops horridus as the internal specifier and Protoceratops andrewsi as the external specifier. We include an additional internal specifier, the mandatory Ceratops montanus.

Chaoyangsauridae Zhao, Cheng & Xu, 1999 (converted clade name)

Registration number: 602

Definition. The largest clade containing Chaoyangsaurus youngi Zhao, Cheng & Xu, 1999 but not Psittacosaurus mongoliensis Osborn, 1923 and Triceratops horridus Marsh, 1889. This is a maximum-clade definition. Abbreviated definition: max ∇ (Chaoyangsaurus youngi Zhao, Cheng & Xu, 1999 ~ Psittacosaurus mongoliensis Osborn, 1923 & Triceratops horridus Marsh, 1889).

Reference phylogeny. Figure 10 of Morschhauser et al. (2019) is treated here as the primary reference phylogeny. Additional reference phylogenies include Figure 10 of Han et al. (2015), Figure 15 of Han et al. (2018), and Figure 3 of Yu et al. (2020).

Composition. Under the primary reference phylogeny, Chaoyangsauridae comprises Chaoyangsaurus youngi, Hualianceratops wucaiwanensis, Xuanhuaceratops niei, and Yinlong downsi.

Synonyms. No other taxon names are currently in use for the same or approximate clade.

Comments. The name Chaoyangsauridae has been (informally) defined before by Han et al. (2015) who applied a maximum-clade definition and used Chaoyangsaurus youngi as the internal specifier and Triceratops horridus and Psittacosaurus mongoliensis as the external specifiers. We formalize this definition.

Chasmosaurinae Lambe, 1915 (converted clade name)

Registration number: 603

Definition. The largest clade containing Chasmosaurus belli (Lambe, 1902) and Triceratops horridus Marsh, 1889 but not Centrosaurus apertus Lambe, 1905. This is a maximum-clade definition. Abbreviated definition: max ∇ (Chasmosaurus belli (Lambe, 1902) & Triceratops horridus Marsh, 1889 ~ Centrosaurus apertus Lambe, 1905).

Reference phylogeny. Figure 9a of Fowler & Freedman Fowler (2020) is treated here as the primary reference phylogeny. Additional reference phylogenies include Figure 3 of Brown & Henderson (2015), Figure 14 of Mallon et al. (2016), Figure S1 of Knapp et al. (2018), Figure 3 of Campbell et al. (2019), and Figure 4 of Yu et al. (2020).

Composition. Under the primary reference phylogeny, Chasmosaurinae comprises Agujaceratops mariscalensis, Anchiceratops ornatus, Arrhinoceratops brachyops, Bravoceratops polyphemus, Chasmosaurus spp., Coahuilaceratops magnacuerna, Kosmoceratops richardsoni, Navajoceratops sullivani, Pentaceratops sternbergii, Terminocavus sealyi, Utahceratops gettyi, Vagaceratops irvinensis, and members of the clade Triceratopsini.

Synonyms. The taxon Ceratops montanus may also fall within the largest clade containing Chasmosaurus belli and Triceratops horridus but not Centrosaurus apertus (see, e.g., Mallon et al., 2016). In such case, Ceratopsinae Abel, 1919 would be an approximate synonym. Though the name has been advocated to be the proper name for the clade (it has been (informally) defined by Sereno, 1998 and Sereno, 2005), it was actually introduced 4 years later than Chasmosaurinae. Note that the Principle of Coordination, which would make Ceratopsinae attributable to Marsh (1888), rather than to Abel (1919), does not apply under the ICPN (see Note 9.15A.3). Therefore, Ceratopsinae would not have priority over Chasmosaurinae under the ICPN. Anyway, C. montanus does not seem to be diagnostic beyond Ceratopsidae at present (Mallon et al., 2016), and its position within the clade is thus uncertain.

Comments. The name Chasmosaurinae has been (informally) defined before by Dodson, Forster & Sampson (2004) who applied a maximum-clade definition and used Triceratops as the internal specifier and Centrosaurus as the external specifier. We apply the name Chasmosaurinae for the same known contents; adopting Triceratops horridus and the mandatory Chasmosaurus belli as the internal specifiers and Centrosaurus apertus as the external specifier.

Clypeodonta Norman, 2014 (converted clade name)

Registration number: 604

Definition. The smallest clade within Ornithopoda containing Edmontosaurus regalis Lambe, 1917 and Hypsilophodon foxii Huxley, 1869. This is a minimum-clade definition. Abbreviated definition: min ∇ ∈ Ornithopoda (Edmontosaurus regalis Lambe, 1917 & Hypsilophodon foxii Huxley, 1869).

Reference phylogeny. Figure 50 of Norman (2015) is treated here as the primary reference phylogeny. Additional reference phylogenies include Figure 25 of Herne et al. (2019) and Figure 2 of Dieudonné et al. (2020).

Composition. Under the primary reference phylogeny, Clypeodonta comprises a clade formed by Hypsilophodon foxii, Rhabdodontidae, and Tenontosaurus spp., and a clade uniting Dryosauridae and Ankylopollexia (termed Iguanodontia in Norman, 2015). However, see ‘Comments’ below for discussion of potential alternative composition of Clypeodonta.

Synonyms. No other taxon names are currently in use for the same or approximate clade. Iguanodontia, as reconstructed, for example, by Madzia, Jagt & Mulder (2020) covers a similar taxic composition; though the topology of Madzia, Jagt & Mulder (2020) differs from that of the primary reference phylogeny of Clypeodonta significantly.

Comments. The name Clypeodonta was claimed as being new in two different studies (Norman, 2014: 29; Norman, 2015: 102), although Norman (2015: 170) also cites Norman (2014) as the establishing reference. The use of the name Clypeodonta differed across studies. Originally, Norman (2014, 2015) intended to use it for a subclade of Ornithopoda that (approximately) comprises Hypsilophodon foxii and its relatives, and ornithopods later-diverging than H. foxii, and (informally) defined the name as pertaining to either, the branch of “Parasaurolophus walkeri and all taxa more closely related to P. walkeri than to Thescelosaurus neglectus” (Norman, 2014: 29) or the node of “Hypsilophodon foxii, Edmontosaurus regalis, their most recent common ancestor, and all of its descendants” (Norman, 2015: 170). In both these studies, Clypeodonta is said (Norman, 2014: 29) or figured (Norman, 2015: Fig. 50) to cover the same known contents although neither of the studies included taxa in their analyses that would fall outside the clade (except for Lesothosaurus diagnosticus). Madzia, Boyd & Mazuch (2018) followed the definition of Norman (2015). In their phylogenetic analysis, however, the name covers a much broader contents as one of the internal specifiers of Clypeodonta, Hypsilophodon foxii, is reconstructed outside Cerapoda in that study (Madzia, Boyd & Mazuch, 2018: Fig. 4). Still, Madzia, Boyd & Mazuch (2018: Appendix 1) stated that as Clypeodonta was a relatively new name with no ‘traditional’ meaning, they saw no reason for its redefinition. They also noted, though, that “given the unstable position of H. foxii among neornithischians, the name might have only limited utility” (Madzia, Boyd & Mazuch, 2018: Appendix 1).

Here we define the name Clypeodonta using the minimum-clade definition of Norman (2015). However, by including the part “within Ornithopoda” in the definition, we restrict the use of Clypeodonta only when H. foxii represents an ornithopod (see Article 11.14 of the ICPN), following the original intent of Norman (2014, 2015).

Coronosauria Sereno, 1986 (converted clade name)

Registration number: 605

Definition. The smallest clade containing Protoceratops andrewsi Granger & Gregory, 1923 and Triceratops horridus Marsh, 1889. This is a minimum-clade definition. Abbreviated definition: min ∇ (Protoceratops andrewsi Granger & Gregory, 1923 & Triceratops horridus Marsh, 1889).

Reference phylogeny. Figure 10 of Morschhauser et al. (2019) is treated here as the primary reference phylogeny. Additional reference phylogenies include Figure S1 of Knapp et al. (2018), Figure 8A of Arbour & Evans (2019), Figure 3 of Yu et al. (2020), and Figure 4 of Yu et al. (2020).

Composition. Under the primary reference phylogeny, Coronosauria comprises members of the clades Protoceratopsidae and Ceratopsoidea.

Synonyms. No other taxon names are currently in use for the same or approximate clade.

Comments. The name Coronosauria has been (informally) defined before by Sereno (1998, 2005) who applied the minimum-clade definition and used Triceratops horridus and Protoceratops andrewsi as the internal specifiers. We formalize this definition.

Corythosauria (new clade name)

Registration number: 746

Definition. The smallest clade containing Corythosaurus casuarius Brown, 1914a, Lambeosaurus lambei Parks, 1923, and Parasaurolophus walkeri Parks, 1922. This is a minimum-clade definition. Abbreviated definition: min ∇ (Corythosaurus casuarius Brown, 1914a & Lambeosaurus lambei Parks, 1923 & Parasaurolophus walkeri Parks, 1922).

Etymology. Derived from the stem of Corythosaurus Brown, 1914a, the name of an included taxon, which combines the Greek words korythos (helmet) and sauros (lizard, reptile).

Reference phylogeny. Figure 18 of Prieto-Márquez, Wagner & Lehman (2020) is treated here as the primary reference phylogeny. Additional reference phylogenies include Figure 5 of Kobayashi et al. (2019), Figure 11 of Prieto-Márquez et al. (2019), Figure 9 of Zhang et al. (2019), Figure 5 of Zhang et al. (2020), Figure 7 of Kobayashi et al. (2021), and Figure 10 of Longrich et al. (2021).

Composition. Under the primary reference phylogeny, Corythosauria comprises members of the clades Lambeosaurini and Parasaurolophini.

Synonyms. No other taxon names are currently in use for the same or approximate clade.

Comments. The name Corythosauria is established for the well-supported node uniting Lambeosaurini and Parasaurolophini, two lambeosaurine clades characterized by their distinctive, ‘crested’ crania.

Dryomorpha Sereno, 1986 (converted clade name)

Registration number: 606

Definition. The smallest clade containing Dryosaurus altus (Marsh, 1878) and Iguanodon bernissartensis Boulenger in Beneden, 1881. This is a minimum-clade definition. Abbreviated definition: min ∇ (Dryosaurus altus (Marsh, 1878) & Iguanodon bernissartensis Boulenger in Beneden, 1881).

Reference phylogeny. Figure 12 of Madzia, Jagt & Mulder (2020) is treated here as the primary reference phylogeny. Additional reference phylogenies include Figure 20 of Verdú et al. (2018), Figure 2 of Dieudonné et al. (2020), Figure 9 of Verdú et al. (2020), and Figure 11 of Santos-Cubedo et al. (2021).

Composition. Under the primary reference phylogeny, Dryomorpha comprises members of the clades Dryosauridae and Ankylopollexia.

Synonyms. No other taxon names are currently in use for the same or approximate clade.

Comments. The name Dryomorpha was first (informally) defined by Sereno (2005) who attributed the name to “(t)he most inclusive clade containing Dryosaurus altus (Marsh, 1878) and Parasaurolophus walkeri Parks, 1922”. However, due to the use of ‘most’, rather than ‘least’, such definition makes the name inapplicable within Ornithischia. Boyd (2015) later corrected the wording and proposed a minimum-clade definition using the same taxa as the internal specifiers. Here we use the same type of definition but replace P. walkeri with I. bernissartensis. This taxon has always been considered a part of Dryomorpha.

Dryosauridae Milner & Norman, 1984 (converted clade name)

Registration number: 607

Definition. The largest clade containing Dryosaurus altus (Marsh, 1878) but not Iguanodon bernissartensis Boulenger in Beneden, 1881. This is a maximum-clade definition. Abbreviated definition: max ∇ (Dryosaurus altus (Marsh, 1878) ~ Iguanodon bernissartensis Boulenger in Beneden, 1881).

Reference phylogeny. Figure 12 of Madzia, Jagt & Mulder (2020) is treated here as the primary reference phylogeny. Additional reference phylogenies include Figure 20 of Verdú et al. (2018), Figure 9 of Verdú et al. (2020), Figure 57 of Barta & Norell (2021), and Figure 11 of Santos-Cubedo et al. (2021).

Composition. Under the primary reference phylogeny, Dryosauridae comprises Callovosaurus leedsi, ‘Camptosaurus’ valdensis, Dryosaurus altus, Dysalotosaurus lettowvorbecki, Elrhazosaurus nigeriensis, Eousdryosaurus nanohallucis, and Valdosaurus canaliculatus.

Synonyms. No other taxon names are currently in use for the same or approximate clade.

Comments. Dryosauridae was first (informally) defined by Sereno (1998: 61) who used the maximum-clade definition and Dryosaurus altus as the internal specifier and Parasaurolophus walkeri as the external specifier. Here we use the same type of definition but replace P. walkeri with I. bernissartensis. This taxon has always been considered outside Dryosauridae.

Edmontosaurini Glut, 1997 (converted clade name)

Registration number: 608

Definition. The largest clade containing Edmontosaurus regalis Lambe, 1917 but not Brachylophosaurus canadensis Sternberg, 1953, Hadrosaurus foulkii Leidy, 1858, Kritosaurus navajovius Brown, 1910, and Saurolophus osborni Brown, 1912. This is a maximum-clade definition. Abbreviated definition: max ∇ (Edmontosaurus regalis Lambe, 1917 ~ Brachylophosaurus canadensis Sternberg, 1953 & Hadrosaurus foulkii Leidy, 1858 & Kritosaurus navajovius Brown, 1910 & Saurolophus osborni Brown, 1912).

Reference phylogeny. Figure 18 of Prieto-Márquez, Wagner & Lehman (2020) is treated here as the primary reference phylogeny. Additional reference phylogenies include Figure 5 of Kobayashi et al. (2019), Figure 11 of Prieto-Márquez et al. (2019), Figure 9 of Zhang et al. (2019), Figure 5 of Zhang et al. (2020), Figure 7 of Kobayashi et al. (2021), and Figure 10 of Longrich et al. (2021).

Composition. Under the primary reference phylogeny, Edmontosaurini comprises Edmontosaurus spp., Kerberosaurus manakini, Kundurosaurus nagornyi, and Shantungosaurus giganteus.

Synonyms. No other taxon names are currently in use for the same or approximate clade.

Comments. The name Edmontosaurini has been (informally) defined before (Sereno, 2005; Xing et al., 2014). Sereno (2005) applied the maximum-clade definition and used Edmontosaurus regalis as the internal specifier and Maiasaura peeblesorum and Saurolophus osborni as the external specifiers. In turn, Xing et al. (2014) applied a minimum-clade definition, with Edmontosaurus and Kerberosaurus as the internal specifiers. We formalize a maximum-clade definition similar to that of Sereno (2005) but replace M. peeblesorum with Brachylophosaurus canadensis, as the representative of Brachylophosaurini, and further add Kritosaurus navajovius and Hadrosaurus foulkii.

Elasmaria Calvo, Porfiri & Novas, 2007 (converted clade name)

Registration number: 609

Definition. The smallest clade containing Macrogryphosaurus gondwanicus Calvo, Porfiri & Novas, 2007 and Talenkauen santacrucensis Novas, Cambiaso & Ambrosio, 2004, provided that it does not include Hypsilophodon foxii Huxley, 1869, Iguanodon bernissartensis Boulenger in Beneden, 1881, or Thescelosaurus neglectus Gilmore, 1913. This is a minimum-clade definition. Abbreviated definition: min ∇ (Macrogryphosaurus gondwanicus Calvo, Porfiri & Novas, 2007 & Talenkauen santacrucensis Novas, Cambiaso & Ambrosio, 2004 | ~ Hypsilophodon foxii Huxley, 1869 ∨ Iguanodon bernissartensis Boulenger in Beneden, 1881 ∨ Thescelosaurus neglectus Gilmore, 1913).

Reference phylogeny. Figure 31 of Rozadilla, Agnolín & Novas, 2019 is treated here as the primary reference phylogeny. Additional reference phylogenies include Figure 4 of Madzia, Boyd & Mazuch (2018), Figure 26 of Herne et al. (2019), Figure 2 of Dieudonné et al. (2020), and Figure 57 of Barta & Norell (2021).

Composition. Under the primary reference phylogeny, Elasmaria comprises Anabisetia saldiviai, Atlascopcosaurus loadsi, Fulgurotherium austral, Gasparinisaura cincosaltensis, Kangnasaurus coetzeei, Macrogryphosaurus gondwanicus, Morrosaurus antarcticus, Notohypsilophodon comodorensis, Quantassaurus intrepidus, and Trinisaura santamartaensis.

Synonyms. No other taxon names are currently in use for the same or approximate clade.

Comments. The name Elasmaria has been (informally) defined before (Calvo, Porfiri & Novas, 2007; Herne et al., 2019). The definition proposed by Calvo, Porfiri & Novas (2007) was minimum-clade, while the definition of Herne et al. (2019) was maximum-clade. However, both studies used Talenkauen santacrucensis and Macrogryphosaurus gondwanicus as the internal specifiers. Herne et al. (2019) proposed to add Iguanodon bernissartensis and Hypsilophodon foxii as the external specifiers to maintain the use of the name Elasmaria to the ‘traditional’ contents under a hypothesis in which one of the internal specifiers was reconstructed, for example, closer to iguanodontians. We keep the use of a minimum-clade definition (as first proposed for the name). However, even though all phylogenetic analyses consistently reconstruct close relationships between T. santacrucensis and M. gondwanicus, we follow Herne et al. (2019) in that the unsettled placement of elasmarians on the neornithischian phylogenetic tree warrants addition of external specifiers. We include Iguanodon bernissartensis and Hypsilophodon foxii as the external specifiers (following Herne et al., 2019) and further add a third external specifier, Thescelosaurus neglectus, to reflect that elasmarians were already inferred as a clade within Thescelosaurinae, as the sister taxon to Thescelosaurus spp. (Boyd, 2015).

Eucentrosaura Chiba et al., 2018 (converted clade name)

Registration number: 688

Definition. The smallest clade containing Centrosaurus apertus Lambe, 1905 and Pachyrhinosaurus canadensis Sternberg, 1950. This is a minimum-clade definition. Abbreviated definition: min ∇ (Centrosaurus apertus Lambe, 1905 & Pachyrhinosaurus canadensis Sternberg, 1950).

Reference phylogeny. Figure 9 of Chiba et al. (2018) is treated here as the primary reference phylogeny. Additional reference phylogenies include Figure 7 of Fiorillo & Tykoski (2012), Figure 10 of Ryan et al. (2017), Figure 13 of Dalman et al. (2018), and Figure 23 of Dalman et al. (2021).

Composition. Under the primary reference phylogeny, Eucentrosaura comprises members of the clades Centrosaurini and Pachyrhinosaurini.

Synonyms. No other taxon names are currently in use for the same or approximate clade.

Comments. The name was first (informally) defined by Chiba et al. (2018) who applied the minimum-clade definition and used Centrosaurus apertus and Pachyrhinosaurus canadensis as the internal specifiers. We formalize this definition.

Euceratopsia (new clade name)

Registration number: 610

Definition. The smallest clade containing Leptoceratops gracilis Brown, 1914b, Protoceratops andrewsi Granger & Gregory, 1923, and Triceratops horridus Marsh, 1889. This is a minimum-clade definition. Abbreviated definition: min ∇ (Leptoceratops gracilis Brown, 1914b & Protoceratops andrewsi Granger & Gregory, 1923 & Triceratops horridus Marsh, 1889).

Etymology. Derived from the Greek eu- (true) and formed to show its association to members of Ceratopsia. Note that Euceratopsia does not derive from the name Ceratops Marsh, 1888, and, as such, the taxon does not have to be the internal specifier in the used definition.

Reference phylogeny. Figure 4 of Yu et al. (2020) is treated here as the primary reference phylogeny. Additional reference phylogenies include Figure 16 of Han et al. (2018), Figure S1 of Knapp et al. (2018), Figure 10 of Morschhauser et al. (2019), and Figure 3 of Yu et al. (2020).

Composition. Under the primary reference phylogeny, Euceratopsia comprises members of the clades Leptoceratopsidae and Coronosauria.

Synonyms. The name Coronosauria Sereno, 1986 covers the same contents under the topology of You & Dodson (2004). However, see ‘Comments’. No other taxon names are currently in use for the same or approximate clade.

Comments. The name Euceratopsia is established for the well-supported node uniting the three latest-diverging clades of ceratopsians – Leptoceratopsidae, Protoceratopsidae, and Ceratopsoidea. The monophyly of the grouping is supported by all recently published phylogenies that infer Euceratopsia to branch into two clades – leptoceratopsids and coronosaurs (protoceratopsids + ceratopsoids). Both these clades comprise representatives that are very close or survived to the Cretaceous/Paleogene mass extinction event (Fowler, 2017: Table S1). It is worth noting that You & Dodson (2004) reconstructed leptoceratopsids to be the sister taxon to Ceratopsoidea, and Protoceratopsidae to be the sister taxon to Leptoceratopsidae + Ceratopsoidea. Under such topology, Euceratopsia becomes a heterodefinitional synonym of Coronosauria, with the latter having priority.

Euhadrosauria Weishampel, Norman & Grigorescu, 1993 (converted clade name)

Registration number: 611

Definition. The smallest clade containing Lambeosaurus lambei Parks, 1923 and Saurolophus osborni Brown, 1912, provided that it does not include Hadrosaurus foulkii Leidy, 1858. This is a minimum-clade definition. Abbreviated definition: min ∇ (Lambeosaurus lambei Parks, 1923 & Saurolophus osborni Brown, 1912 | ~ Hadrosaurus foulkii Leidy, 1858).

Reference phylogeny. Figure 18 of Prieto-Márquez, Wagner & Lehman (2020) is treated here as the primary reference phylogeny. Additional reference phylogenies include Figure 11 of Prieto-Márquez et al. (2019), Figure 9 of Zhang et al. (2019), Figure 7 of Kobayashi et al. (2021), Figure 10 of Longrich et al. (2021), and Figure 11 of McDonald et al. (2021).

Composition. Under the primary reference phylogeny, Euhadrosauria comprises members of the clades Saurolophinae and Lambeosaurinae.

Synonyms. The name Hadrosauridae Cope, 1869 is an approximate synonym of Euhadrosauria. If Hadrosaurus foulkii nests within the smallest clade containing Saurolophus osborni and Lambeosaurus lambei, and within the ‘Saurolophus branch’ of the clade (see the entry for the name Saurolophinae), the name Hadrosauridae is used for the node instead, and Euhadrosauria becomes inapplicable. Additionally, the name Saurolophidae has been used for the same contents as well (see ‘Comments’).

Comments. The history and application of Euhadrosauria is complicated and has been thoroughly described and discussed by Madzia, Jagt & Mulder (2020: 14–16). We therefore refer to that study for details.

Euiguanodontia Coria & Salgado, 1996 (converted clade name)

Registration number: 612

Definition. The smallest clade containing Camptosaurus dispar (Marsh, 1879), Dryosaurus altus (Marsh, 1878), and Gasparinisaura cincosaltensis Coria & Salgado, 1996, provided that it does not include Tenontosaurus tilletti Ostrom, 1970. This is a minimum-clade definition. Abbreviated definition: min ∇ (Camptosaurus dispar (Marsh, 1879) & Dryosaurus altus (Marsh, 1878) & Gasparinisaura cincosaltensis Coria & Salgado, 1996 | ~ Tenontosaurus tilletti Ostrom, 1970).

Reference phylogeny. Figure 13 of Coria & Salgado (1996) is treated here as the primary reference phylogeny.

Composition. Under the primary reference phylogeny, Euiguanodontia comprises Gasparinisaura and members of the clades Dryosauridae and Ankylopollexia.

Synonyms. No other taxon names are currently in use for the same or approximate clade.

Comments. The name Euiguanodontia is applicable only on the condition that G. cincosaltensis, D. altus, and C. dispar form a clade exclusive of T. tilletti, as originally used by Coria & Salgado (1996). We follow the definition advocated by Madzia, Boyd & Mazuch (2018: Appendix 1) and refer to that study for additional comments. Note also that Euiguanodontia must be a subclade of Iguanodontia under the proposed definition because T. tilletti is an internal specifier in the definition of the name. Finally, note that the internal specifiers Dryosaurus altus and Camptosaurus dispar are not included in the primary reference phylogeny. The former belongs to Dryosauridae (e.g., Madzia, Boyd & Mazuch, 2018), while the latter is part of Ankylopollexia (see, e.g., Madzia, Jagt & Mulder, 2020). Both these clades are indicated on Figure 13 of Coria & Salgado (1996).

Euornithopoda Sereno, 1986 (converted clade name)

Registration number: 613

Definition. The largest clade within Ornithopoda containing Iguanodon bernissartensis Boulenger in Beneden, 1881 but not Heterodontosaurus tucki Crompton & Charig, 1962. This is a maximum-clade definition. Abbreviated definition: max ∇ ∈ Ornithopoda (Iguanodon bernissartensis Boulenger in Beneden, 1881 ~ Heterodontosaurus tucki Crompton & Charig, 1962).

Reference phylogeny. Figure 1 of Sereno (1999) is treated here as the primary reference phylogeny.

Composition. Under the primary reference phylogeny, Euornithopoda comprises Tenontosaurus spp. and members of the clades Ankylopollexia, Dryosauridae, and Hypsilophodontidae.

Synonyms. No other taxon names are currently in use for the same or approximate clade.

Comments. The name Euornithopoda has been (informally) defined before (Sereno, 1998; Sereno, 2005). These definitions were maximum-clade and used Parasaurolophus as the internal specifier and Heterodontosaurus tucki, Pachycephalosaurus wyomingensis, Triceratops horridus, and Ankylosaurus magniventris (Sereno, 2005) as the external specifiers. Here we define the name Euornithopoda using a similar maximum-clade definition as that of Sereno (1998) but replace Parasaurolophus with Iguanodon bernissartensis. Also, by including the part “within Ornithopoda” in the definition, we restrict the use of Euornithopoda to the branch only when Heterodontosaurus tucki represents an ornithopod (see Article 11.14 of the ICPN), thus maintaining the ‘traditional’ use (Sereno, 1998; Sereno, 2005).

Eurypoda Sereno, 1986 (converted clade name)

Registration number: 614

Definition. The smallest clade containing Ankylosaurus magniventris Brown, 1908 and Stegosaurus stenops Marsh, 1887. This is a minimum-clade definition. Abbreviated definition: min ∇ (Ankylosaurus magniventris Brown, 1908 & Stegosaurus stenops Marsh, 1887).

Reference phylogeny. Figure 3 of Thompson et al. (2012) is treated here as the primary reference phylogeny. Additional reference phylogenies include Figure 16 of Han et al. (2018) and Figure 1 of Dieudonné et al. (2020).

Composition. Under the primary reference phylogeny, Eurypoda comprises members of the clades Ankylosauria and Stegosauria.

Synonyms. No other taxon names are currently in use for the same or approximate clade.

Comments. The name Eurypoda has been (informally) defined before by Sereno (1998) who used Ankylosaurus and Stegosaurus as the internal specifiers. Since Eurypoda has never been proposed an alternative use, we formalize this definition. Note that the internal specifier Stegosaurus stenops is not included in the primary reference phylogeny. The taxon is most closely related to the clade comprising the operational taxonomic units (OTUs) Stegosaurus armatus (nomen dubium according to Galton, 2010; S. armatus has long been the type species of Stegosaurus but was replaced by S. stenops as the type through an ICZN ruling (International Commission on Zoological Nomenclature, 2013)) and Huayangosaurus taibaii (see, e.g., Maidment et al., 2020).

Genasauria Sereno, 1986 (converted clade name)

Registration number: 615

Definition. The smallest clade containing Ankylosaurus magniventris Brown, 1908, Iguanodon bernissartensis Boulenger in Beneden, 1881, Stegosaurus stenops Marsh, 1887, and Triceratops horridus Marsh, 1889. This is a minimum-clade definition. Abbreviated definition: min ∇ (Ankylosaurus magniventris Brown, 1908 & Iguanodon bernissartensis Boulenger in Beneden, 1881 & Stegosaurus stenops Marsh, 1887 & Triceratops horridus Marsh, 1889).

Reference phylogeny. Figure 16 of Han et al. (2018) is treated here as the primary reference phylogeny. Additional reference phylogenies include Figure 4 of Madzia, Boyd & Mazuch (2018), Figure 25 of Herne et al. (2019), Figure 1 of Dieudonné et al. (2020), Figure 12 of Yang et al. (2020), and Figure 57 of Barta & Norell (2021).

Composition. Under the primary reference phylogeny, Genasauria comprises members of the clades Neornithischia and Thyreophora.

Synonyms. No other taxon names are currently in use for the same or approximate clade.

Comments. The name Genasauria has been (informally) defined before (Currie & Padian, 1997; Sereno, 1998; Sereno, 2005; Butler, Upchurch & Norman, 2008). These definitions were minimum-clade and used Thyreophora and Cerapoda (Currie & Padian, 1997), Ankylosaurus and Triceratops (Sereno, 1998), Ankylosaurus magniventris, Triceratops horridus, and Parasaurolophus walkeri (Sereno, 2005), and Ankylosaurus magniventris, Stegosaurus stenops, Triceratops horridus, Parasaurolophus walkeri, and Pachycephalosaurus wyomingensis (Butler, Upchurch & Norman, 2008) as the internal specifiers. In order to maintain the ‘traditional’ concept of Genasauria as a clade comprising Neornithischia and Thyreophora, the internal specifiers in the definition of Genasauria are used from among the taxa representing the four major subclades – Ornithopoda (Iguanodon bernissartensis), Marginocephalia (Triceratops horridus), Ankylosauria (Ankylosaurus magniventris), and Stegosauria (Stegosaurus stenops). Addition of P. wyomingensis as another internal specifier (to include representatives of both marginocephalian clades – Ceratopsia and Pachycephalosauria) is considered unnecessary because pachycephalosaurs have always been inferred to be part of Genasauria as defined herein. Note that the internal specifiers Ankylosaurus magniventris and Triceratops horridus are not included in the primary reference phylogeny. The former belongs to Ankylosauria within Thyreophora (see, e.g., Thompson et al., 2012), while the latter is part of Ceratopsia (e.g., Morschhauser et al., 2019).

Hadrosauridae Cope, 1869 (converted clade name)

Registration number: 616

Definition. The smallest clade containing Hadrosaurus foulkii Leidy, 1858, Lambeosaurus lambei Parks, 1923, and Saurolophus osborni Brown, 1912. This is a minimum-clade definition. Abbreviated definition: min ∇ (Hadrosaurus foulkii Leidy, 1858 & Lambeosaurus lambei Parks, 1923 & Saurolophus osborni Brown, 1912).

Reference phylogeny. Figure 18 of Prieto-Márquez, Wagner & Lehman (2020) is treated here as the primary reference phylogeny. Additional reference phylogenies include Figure 5 of Kobayashi et al. (2019), Figure 11 of Prieto-Márquez et al. (2019), Figure 9 of Zhang et al. (2019), Figure 5 of Zhang et al. (2020), Figure 7 of Kobayashi et al. (2021), and Figure 10 of Longrich et al. (2021).

Composition. Under the primary reference phylogeny, Hadrosauridae comprises Hadrosaurus foulkii, Eotrachodon orientalis, Latirhinus uitstlani, Aquilarhinus palimentus, and members of the clade Euhadrosauria.

Synonyms. Several taxon names have been historically or recently used as approximate synonyms of Hadrosauridae. Of these, only the names Saurolophidae and Euhadrosauria have recently been attributed to a clade of the same or a similar composition (e.g., Prieto-Márquez, 2010; Verdú et al., 2018; Zhang et al., 2019; Madzia, Jagt & Mulder, 2020; Prieto-Márquez, Wagner & Lehman, 2020; Verdú et al., 2020; Zhang et al., 2020; Kobayashi et al., 2021; Ramírez-Velasco et al., 2021). See ‘Comments’ below.

Comments. The use of Hadrosauridae and other names applied to the same or similar clades (Saurolophidae and Euhadrosauria) have been thoroughly described and discussed by Madzia, Jagt & Mulder (2020: 14–16) who recommended to use Hadrosauridae for the smallest clade containing H. foulkii, S. osborni, and L. lambei; Euhadrosauria for the smallest clade containing S. osborni and L. lambei; and to abandon Saurolophidae. Note that under some phylogenies, in which H. foulkii is reconstructed within the smallest clade containing S. osborni and L. lambei, the names Hadrosauridae and Euhadrosauria, as (informally) defined by Madzia, Jagt & Mulder (2020), become heterodefinitional synonyms. Although such option may still be viewed acceptable, we decided to apply a minimum-clade definition for Euhadrosauria that makes the name inapplicable under such hypothesis.

Hadrosauriformes Sereno, 1997 (converted clade name)

Registration number: 617

Definition. The smallest clade containing Hadrosaurus foulkii Leidy, 1858 and Iguanodon bernissartensis Boulenger in Beneden, 1881. This is a minimum-clade definition. Abbreviated definition: min ∇ (Hadrosaurus foulkii Leidy, 1858 & Iguanodon bernissartensis Boulenger in Beneden, 1881).

Reference phylogeny. Figure 12 of Madzia, Jagt & Mulder (2020) is treated here as the primary reference phylogeny. Additional reference phylogenies include Figure 20 of Verdú et al. (2018), Figure 3 of Párraga & Prieto-Márquez (2019), Figure 8 of Słowiak et al. (2020), Figure 9 of Verdú et al. (2020) and Figure 11 of McDonald et al. (2021).

Composition. Under the primary reference phylogeny, Hadrosauriformes comprises members of the clades Iguanodontidae and Hadrosauroidea.

Synonyms. If Hypselospinus fittoni nests within the smallest clade containing Hadrosaurus foulkii and Iguanodon bernissartensis, the name Hadrosauriformes is a potential heterodefinitional synonym of Neoiguanodontia (see the name entry). In such case, the name Hadrosauriformes should have priority. The name Iguanodontoidea Hay, 1902 has been also used as an approximate synonym (Sereno, 1986; Norman, 2002). Note that Norman (2002) used Iguanodontoidea for a clade “(s)erially more derived than Camptosaurus” (Norman, 2002: 138) and defined it as “Iguanodon and all iguanodontians more closely related to Edmontosaurus than to Camptosaurus”. Such definition would make Iguanodontoidea applicable for the same clade as Styracosterna (see the name entry). However, Figure 35 of Norman (2002) shows that the name does not cover Lurdusaurus, which should be included within the clade under such maximum-clade definition. Since Norman (2002) considers Iguanodontoidea to be a synonym of Hadrosauriformes of Sereno (1997, 1998, 1999), it is apparent that Norman (2002) concept of Iguanodontoidea would be more similar to that of Hadrosauriformes rather than Styracosterna.

Comments. The name Hadrosauriformes has been (informally) defined before (Sereno, 1998; Norman, 2015; Madzia, Jagt & Mulder, 2020). However, only Madzia, Jagt & Mulder (2020: Table 1) included the mandatory H. foulkii as the internal specifier. We formalize the definition of Madzia, Jagt & Mulder (2020).

Hadrosaurinae Lambe, 1918 (converted clade name)

Registration number: 618

Definition. The largest clade containing Hadrosaurus foulkii Leidy, 1858 but not Lambeosaurus lambei Parks, 1923. This is a maximum-clade definition. Abbreviated definition: max ∇ (Hadrosaurus foulkii Leidy, 1858 ~ Lambeosaurus lambei Parks, 1923).

Reference phylogeny. Figure 5 of Kobayashi et al. (2019) is treated here as the primary reference phylogeny. Additional reference phylogenies include Figure 13 of Cruzado-Caballero & Powell (2017), Figure 20 of Xing, Mallon & Currie (2017), Figure 5 of Zhang et al. (2020), and Figure 10 of Longrich et al. (2021).

Composition. Under the primary reference phylogeny, Hadrosaurinae comprises Hadrosaurus foulkii and members of the clades Brachylophosaurini, Edmontosaurini, Kritosaurini, and Saurolophini.

Synonyms. The name Saurolophinae Brown, 1914a has been recently used for the same clade (under the hypothesis in which H. foulkii is nested outside the smallest clade containing Saurolophus osborni and Lambeosaurus lambei). See the entry for the name Saurolophinae.

Comments. The name Hadrosaurinae has been (informally) defined before by (Sereno, 1998; Sereno, 2005). Sereno (1998) applied the maximum-clade definition and used Saurolophus as the internal specifier and Parasaurolophus as the external specifier. In turn, Sereno (2005), apparently erroneously, defined Hadrosaurinae as pertaining to “(t)he most inclusive taxon containing Saurolophus osborni Brown, 1912 and Parasaurolophus walkeri Parks, 1922 and including Hadrosaurus foulkii Leidy, 1858”. Our formal maximum-clade definition was formed to make Hadrosaurinae applicable regardless of whether the taxon lies ouside or within the smallest clade containing Saurolophus osborni and Lambeosaurus lambei.

Hadrosauroidea von Huene, 1952 (converted clade name)

Registration number: 619

Definition. The largest clade containing Hadrosaurus foulkii Leidy, 1858 but not Iguanodon bernissartensis Boulenger in Beneden, 1881. This is a maximum-clade definition. Abbreviated definition: max ∇ (Hadrosaurus foulkii Leidy, 1858 ~ Iguanodon bernissartensis Boulenger in Beneden, 1881).

Reference phylogeny. Figure 12 of Madzia, Jagt & Mulder (2020) is treated here as the primary reference phylogeny. Additional reference phylogenies include Figure 20 of Verdú et al. (2018), Figure 8 of Słowiak et al. (2020), Figure 9 of Verdú et al. (2020), Figure 11 of McDonald et al. (2021), and Figure 11 of Santos-Cubedo et al. (2021).

Composition. Under the primary reference phylogeny, Hadrosauroidea comprises Altirhinus kurzanovi, Batyrosaurus rozhdestvenskyi, Bolong yixianensis, Equijubus normani, Gongpoquansaurus mazongshanensis, Jinzhousaurus yangi, Koshisaurus katsuyama, Mantellisaurus atherfieldensis, Morelladon beltrani, Ouranosaurus nigeriensis, Penelopognathus weishampeli, Proa valdearinnoensis, Probactrosaurus gobiensis, Ratchasimasaurus suranareae, Sirindhorna khoratensis, Xuwulong yueluni, Zuoyunlong huangi, and members of the clade Hadrosauromorpha.

Synonyms. No other taxon names are currently in use for the same or approximate clade.

Comments. The name Hadrosauroidea was first (informally) defined by Sereno (1998: 62) who used the maximum-clade definition and Parasaurolophus walkeri as the internal specifier and Iguanodon bernissartensis as the external specifier. We formalize the definition of Madzia, Jagt & Mulder (2020: Table 1) who replaced P. walkeri with H. foulkii.

Hadrosauromorpha Norman, 2014 (converted clade name)

Registration number: 620

Definition. The largest clade containing Hadrosaurus foulkii Leidy, 1858 but not Probactrosaurus gobiensis Rozhdestvensky, 1966. This is a maximum-clade definition. Abbreviated definition: max ∇ (Hadrosaurus foulkii Leidy, 1858 ~ Probactrosaurus gobiensis Rozhdestvensky, 1966).

Reference phylogeny. Figure 12 of Madzia, Jagt & Mulder (2020) is treated here as the primary reference phylogeny. Additional reference phylogenies include Figure 20 of Verdú et al. (2018), Figure 9 of Verdú et al. (2020), Figure 7 of Kobayashi et al. (2021), and Figure 11 of Santos-Cubedo et al. (2021).

Composition. Under the primary reference phylogeny, Hadrosauromorpha comprises Bactrosaurus johnsoni, Datonglong tianzhenensis, Eolambia caroljonesa, Gilmoreosaurus mongoliensis, Jeyawati rugoculus, Jintasaurus meniscus, Levnesovia transoxiana, Nanyangosaurus zhugeii, ‘Orthomerus dolloi’, Plesiohadros djadokhtaensis, Protohadros byrdi, Tanius sinensis, Tethyshadros insularis, Shuangmiaosaurus gilmorei, Zhanghenglong yangchengensis, and members of the clade Hadrosauridae.

Synonyms. No other taxon names are currently in use for the same or approximate clade.

Comments. Hadrosauromorpha was first (informally) defined by Norman (2014: 32) who used the maximum-clade definition and Parasaurolophus walkeri as the internal specifier and Probactrosaurus gobiensis as the external specifier. We formalize the definition of Madzia, Jagt & Mulder (2020: Table 1) who replaced P. walkeri with H. foulkii.

Heterodontosauridae Kuhn, 1966 (converted clade name)

Registration number: 622

Definition. The largest clade containing Heterodontosaurus tucki Crompton & Charig, 1962 but not Iguanodon bernissartensis Boulenger in Beneden, 1881, Pachycephalosaurus wyomingensis (Gilmore, 1931), Stegosaurus stenops Marsh, 1887, and Triceratops horridus Marsh, 1889. This is a maximum-clade definition. Abbreviated definition: max ∇ (Heterodontosaurus tucki Crompton & Charig, 1962 ~ Iguanodon bernissartensis Boulenger in Beneden, 1881 & Pachycephalosaurus wyomingensis (Gilmore, 1931) & Stegosaurus stenops Marsh, 1887 & Triceratops horridus Marsh, 1889).

Reference phylogeny. Figure 4 of Madzia, Boyd & Mazuch (2018) is treated here as the primary reference phylogeny. Additional reference phylogenies include Figure 25 of Herne et al. (2019), Figure 12 of Yang et al. (2020), and Figure 57 of Barta & Norell (2021).

Composition. Under the primary reference phylogeny, Heterodontosauridae comprises Abrictosaurus consors, Echinodon becklesii, Eocursor parvus, Fruitadens haagarorum, Heterodontosaurus tucki, Lycorhinus angustidens, Manidens condorensis, Pegomastax africana, and Tianyulong confuciusi.

Synonyms. No other taxon names are currently in use for the same or approximate clade.

Comments. We follow Sereno (2012) in recognizing Kuhn (1966), rather than Romer (1966), as the author establishing Heterodontosauridae. The name Heterodontosauridae has been (informally) defined before (Sereno, 1998; Sereno, 2005). These definitions were maximum-clade and used Heterodontosaurus as the internal specifier and Parasaurolophus (Sereno, 1998) or Parasaurolophus walkeri, Pachycephalosaurus wyomingensis, Triceratops horridus, and Ankylosaurus magniventris (Sereno, 2005) as the external specifiers. We apply the name Heterodontosauridae for the same known contents; adopting the mandatory Heterodontosaurus tucki as the internal specifier and representatives of all major ornithischian lineages, Ceratopsia (Triceratops horridus), Ornithopoda (Iguanodon bernissartensis), Pachycephalosauria (Pachycephalosaurus wyomingensis), and Thyreophora (Stegosaurus stenops), as the external specifiers. Note that the external specifiers Pachycephalosaurus wyomingensis, Stegosaurus stenops, and Triceratops horridus are not included in the primary reference phylogeny. P. wyomingensis and T. horridus belong to Marginocephalia that is indicated on Figure 4 of Madzia, Boyd & Mazuch (2018), while S. stenops is nested within Thyreophora (e.g., Maidment et al., 2020).

Huayangosauridae Dong, Tang & Zhou, 1982 (converted clade name)

Registration number: 623

Definition. The largest clade containing Huayangosaurus taibaii Dong, Tang & Zhou, 1982 but not Stegosaurus stenops Marsh, 1887. This is a maximum-clade definition. Abbreviated definition: max ∇ (Huayangosaurus taibaii Dong, Tang & Zhou, 1982 ~ Stegosaurus stenops Marsh, 1887).

Reference phylogeny. Figure 12 of Maidment et al. (2020) is treated here as the primary reference phylogeny. Additional reference phylogenies include Figure 11 of Maidment et al. (2008) and Figure 1 of Raven & Maidment (2017).

Composition. Under the primary reference phylogeny, Huayangosauridae comprises Chungkingosaurus jiangbeiensis and Huayangosaurus taibaii.

Synonyms. No other taxon names are currently in use for the same or approximate clade.

Comments. The name Huayangosauridae was first (informally) defined by Galton & Upchurch (2004: 358) who used the maximum-clade definition and selected Huayangosaurus as the internal specifier and Stegosaurus as the external specifier. We formalize this definition.

Hypsilophodontia Cooper, 1985 (converted clade name)

Registration number: 624

Definition. The smallest clade within Ornithopoda containing Hypsilophodon foxii Huxley, 1869 and Tenontosaurus tilletti Ostrom, 1970, provided that it does not include Iguanodon bernissartensis Boulenger in Beneden, 1881. This is a minimum-clade definition. Abbreviated definition: min ∇ ∈ Ornithopoda (Hypsilophodon foxii Huxley, 1869 & Tenontosaurus tilletti Ostrom, 1970 | ~ Iguanodon bernissartensis Boulenger in Beneden, 1881).

Reference phylogeny. Figure 50 of Norman (2015) is treated here as the primary reference phylogeny.

Composition. Under the primary reference phylogeny, Hypsilophodontia comprises Hypsilophodon foxii, Tenontosaurus spp., and members of the clade Rhabdodontidae. However, see ‘Comments’ below for discussion of potential alternative composition of Clypeodonta.

Synonyms. No other taxon names are currently in use for the same or approximate clade.

Comments. The name Hypsilophodontia was (informally) defined as pertaining to “Hypsilophodon foxii, Tenontosaurus tilletti, their most recent common ancestor, and all of its descendants” (Norman, 2015: 171). However, such definition does not reflect alternative topologies that do not support Hypsilophodontia as reconstructed by Norman (2015), making it applicable for markedly different contents (see, e.g., Madzia, Boyd & Mazuch, 2018: Fig. 4).

Here we define the name Hypsilophodontia using a similar minimum-clade definition as that of Norman (2015) but by including the part “within Ornithopoda” in the definition, and adding an external specifier, we restrict the use of Hypsilophodontia to the node only when H. foxii represents an ornithopod (see Article 11.14 of the ICPN) and when Hypsilophodon foxii and Tenontosaurus tilletti are more closely related to each other than either is to I. bernissartensis, following the original intent of Norman (2015). Note that the internal specifier Tenontosaurus tilletti is not indicated in the primary reference phylogeny. The taxon is the type species of Tenontosaurus Ostrom, 1970 and is comprised there within the ‘tenontosaurs’.

Hypsilophodontidae Dollo, 1882 (converted clade name)

Registration number: 625

Definition. The largest clade containing Hypsilophodon foxii Huxley, 1869 but not Iguanodon bernissartensis Boulenger in Beneden, 1881 and Rhabdodon priscus Matheron, 1869. This is a maximum-clade definition. Abbreviated definition: max ∇ (Hypsilophodon foxii Huxley, 1869 ~ Iguanodon bernissartensis Boulenger in Beneden, 1881 & Rhabdodon priscus Matheron, 1869).

Reference phylogeny. Figure 2 of Dieudonné et al. (2020) is treated here as the primary reference phylogeny.

Composition. Under the primary reference phylogeny, Hypsilophodontidae comprises Hypsilophodon foxii, Gasparinisaura cincosaltensis, and Parksosaurus warreni.

Synonyms. The name Parksosaurinae has been recently for the same contents (Yang et al., 2020), and attributed (apparently following the Principle of Coordination) to Buchholz (2002). No other taxon names are currently in use for the same or approximate clade.

Comments. Hypsilophodontidae was first (informally) defined by Sereno (1998: 61) who used the maximum-clade definition and Hypsilophodon foxii as the internal specifier and Parasaurolophus walkeri as the external specifier. Here we use the same type of definition but replace P. walkeri with I. bernissartensis. This taxon has always been considered outside Hypsilophodontidae. Additionally, we include Rhabdodon priscus as a second external specifier to prevent the inclusion of Rhabdodontidae within Hypsilophodontidae under the topology of Norman (2015: Fig. 50).

Iguanodontia Baur, 1891 (converted clade name)

Registration number: 626

Definition. The smallest clade containing Dryosaurus altus (Marsh, 1878), Iguanodon bernissartensis Boulenger in Beneden, 1881, Rhabdodon priscus Matheron, 1869, and Tenontosaurus tilletti Ostrom, 1970, provided that it does not include Hypsilophodon foxii Huxley, 1869. This is a minimum-clade definition. Abbreviated definition: min ∇ (Dryosaurus altus (Marsh, 1878) & Iguanodon bernissartensis Boulenger in Beneden, 1881 & Rhabdodon priscus Matheron, 1869 & Tenontosaurus tilletti Ostrom, 1970 | ~ Hypsilophodon foxii Huxley, 1869).

Reference phylogeny. Figure 12 of Madzia, Jagt & Mulder (2020) is treated here as the primary reference phylogeny. Additional reference phylogenies include Figure 16 of Han et al. (2018), Figure 20 of Verdú et al. (2018), Figure 25 of Herne et al. (2019), and Figure 9 of Verdú et al. (2020).

Composition. Under the primary reference phylogeny, Iguanodontia comprises members of the clade Rhabdodontomorpha, Tenontosaurus spp., and Dryomorpha.

Synonyms. No other taxon names are currently in use for the same or approximate clade. Clypeodonta, as reconstructed by Norman (2015) covers a similar taxic composition; though the topology of Norman (2015) differs from that of the primary phylogeny of Iguanodontia significantly.

Comments. The application of Iguanodontia has been described and discussed by Madzia, Boyd & Mazuch (2018: Appendix 1) and Madzia, Jagt & Mulder (2020: Table 1). We therefore refer to these studies for details. Our definition differs from that advocated by Madzia, Boyd & Mazuch (2018) and Madzia, Jagt & Mulder (2020) in that the name is newly applicable only if it is used for a clade that does not include Hypsilophodon foxii (e.g., it becomes inapplicable under the topology of Norman, 2015: Fig. 50).

Iguanodontidae Bonaparte, 1850 (converted clade name)

Registration number: 627

Definition. The largest clade containing Iguanodon bernissartensis Boulenger in Beneden, 1881 but not Hadrosaurus foulkii Leidy, 1858. This is a maximum-clade definition. Abbreviated definition: max ∇ (Iguanodon bernissartensis Boulenger in Beneden, 1881 ~ Hadrosaurus foulkii Leidy, 1858).

Reference phylogeny. Figure 13 of Madzia, Jagt & Mulder (2020) is treated here as the primary reference phylogeny. Additional reference phylogenies include Figure 3 of Madzia, Boyd & Mazuch (2018), Figure 20 of Verdú et al. (2018), Figure 32 of Tsogtbaatar et al. (2019), Figure 7 of Kobayashi et al. (2021), and Figure 11 of Santos-Cubedo et al. (2021).

Composition. Under the primary reference phylogeny, Iguanodontidae comprises Barilium dawsoni, Iguanodon bernissartensis, Iguanodon galvensis, and Lurdusaurus arenatus.

Synonyms. The name Iguanodontoidea Hay, 1902 is an approximate synonym of Iguanodontidae (see, e.g., Figure 20 of Verdú et al., 2018). Both these names have been used for various sets of taxa thought or reconstructed to be more closely related to Iguanodon bernissartensis than to hadrosaurids. Considering that significant differences exist between phylogeny reconstructions of Iguanodon-grade ornithopods (e.g., Madzia, Boyd & Mazuch, 2018; Verdú et al., 2018; Madzia, Jagt & Mulder, 2020; McDonald et al., 2021), it is difficult to link either of the names to a certain, stable composition. Here, we prefer to apply the name Iguanodontidae because it is more frequent in the literature and because it was coined 52 years before Iguanodontoidea. It is worth noting that the name Iguanodontoidea has been also used as an approximate synonym of Hadrosauriformes (see the name entry).

Comments. The name Iguanodontidae was first (informally) defined before (Sereno, 1998; Sereno, 2005; Santos-Cubedo et al., 2021). These definitions were maximum-clade and used Iguanodon bernissartensis as the internal specifier and Parasaurolophus walkeri (Sereno, 1998; Sereno, 2005) or Corythosaurus casuarius (Santos-Cubedo et al., 2021) as the external specifier. We apply a similar definition but replace P. walkeri/Corythosaurus casuarius with H. foulkii. Note that even though the study of Santos-Cubedo et al. (2021) appeared after the publication of Phylonyms (de Queiroz, Cantino & Gauthier, 2020), the work does not meet the general requirements for establishing Iguanodontidae as a phylogenetically defined clade name (see Articles 7 of the ICPN), nor it provides anything that would indicate such intention. Specifically, the name Iguanodontidae is not explicitly designated as a converted clade name, no bibliographic citations demonstrating prior application of the name to a taxon approximating the clade for which it is being established have been provided (including the authorship of the preexisting name), and no evidence is provided that the required information has been submitted to the registration database for phylogenetically defined names, the RegNum (registration number is missing). The study specifies the phylogenetic information, such as the placement of the clade on the ornithopod tree and the distribution of apomorphies supporting the existence of the clade, and presents the hypothesized composition of the clade. This information alone, however, would not be sufficient for the name Iguanodontidae to be established as a converted clade name, as required by the ICPN.

Jeholosauridae Han et al., 2012 (converted clade name)

Registration number: 628

Definition. The largest clade outside Hypsilophodontidae or Thescelosauridae containing Jeholosaurus shangyuanensis Xu, Wang & You, 2000 but not Hypsilophodon foxii Huxley, 1869, Iguanodon bernissartensis Boulenger in Beneden, 1881, Pachycephalosaurus wyomingensis (Gilmore, 1931), Thescelosaurus neglectus Gilmore, 1913, and Triceratops horridus Marsh, 1889. This is a maximum-clade definition. Abbreviated definition: max ∇ ∉ Hypsilophodontidae ∨ Thescelosauridae (Jeholosaurus shangyuanensis Xu, Wang & You 2000 ~ Hypsilophodon foxii Huxley, 1869 & Iguanodon bernissartensis Boulenger in Beneden, 1881 & Pachycephalosaurus wyomingensis (Gilmore, 1931) & Thescelosaurus neglectus Gilmore, 1913 & Triceratops horridus Marsh, 1889).

Reference phylogeny. Figure 25 of Herne et al. (2019) is treated here as the primary reference phylogeny. Additional reference phylogenies include Figure 16 of Han et al. (2018), Figure 4 of Madzia, Boyd & Mazuch (2018), and Figure 57 of Barta & Norell (2021).

Composition. Under the primary reference phylogeny, Jeholosauridae comprises Changchunsaurus parvus, Haya griva, and Jeholosaurus shangyuanensis. Under alternative hypotheses, however, Jeholosauridae includes Jeholosaurus shangyuanensis and Yueosaurus tiantaiensis (e.g., Madzia, Boyd & Mazuch, 2018: Fig. 4; Barta & Norell, 2021: Fig. 57).

Synonyms. The name Jeholosaurinae has been used recently for the same contents (Yang et al., 2020), and attributed (apparently following the Principle of Coordination) to Han et al. (2012). No other taxon names are currently in use for the same or approximate clade.

Comments. We use a maximum-clade definition similar to that of Han et al. (2012), which is the only definition (informally) used for Jeholosauridae. Our definition differs in that we replaced the original representative of Ceratopsia (Protoceratops andrewsi) with a taxon that is widely used in phylogenetic definitions of ornithischian clade names (Triceratops horridus). Additionally, our definition prevents the use of Jeholosauridae under the potential hypotheses in which Jeholosaurus is inferred as part of Hypsilophodontidae or Thescelosauridae. Note that the internal specifiers Pachycephalosaurus wyomingensis and Triceratops horridus are not included in the primary reference phylogeny. The former belongs to Pachycephalosauria (see, e.g., Dieudonné et al., 2020), while the latter is part of Ceratopsia (e.g., Morschhauser et al., 2019), both within Marginocephalia that is indicated on Figure 25 of Herne et al. (2019).

Kritosaurini Glut, 1997 (converted clade name)

Registration number: 629

Definition. The largest clade containing Kritosaurus navajovius Brown, 1910 but not Brachylophosaurus canadensis Sternberg, 1953, Edmontosaurus regalis Lambe, 1917, Hadrosaurus foulkii Leidy, 1858, and Saurolophus osborni Brown, 1912. This is a maximum-clade definition. Abbreviated definition: max ∇ (Kritosaurus navajovius Brown, 1910 ~ Brachylophosaurus canadensis Sternberg, 1953 & Edmontosaurus regalis Lambe, 1917 & Hadrosaurus foulkii Leidy, 1858 & Saurolophus osborni Brown, 1912).

Reference phylogeny. Figure 18 of Prieto-Márquez, Wagner & Lehman (2020) is treated here as the primary reference phylogeny. Additional reference phylogenies include Figure 5 of Kobayashi et al. (2019), Figure 11 of Prieto-Márquez et al. (2019), Figure 9 of Zhang et al. (2019), Figure 5 of Zhang et al. (2020), Figure 7 of Kobayashi et al. (2021), and Figure 10 of Longrich et al. (2021).

Composition. Under the primary reference phylogeny Kritosaurini comprises Gryposaurus spp., Kritosaurus spp., Rhinorex condrupus, Secernosaurus koerneri, and the specimen ‘Big Bend UTEP 37.7’.

Synonyms. No other taxon names are currently in use for the same or approximate clade.

Comments. The study of Lapparent & Lavocat (1955) has been cited to be the reference establishing the name Kritosaurini (e.g., Prieto-Márquez, 2014). However, Lapparent & Lavocat (1955) used ‘Kritosaurinés’ rather than ‘Kritosaurini’. The name Kritosaurini was then used by Brett-Surman (1989) and by Glut (1997). Since Brett-Surman (1989) is an unpublished doctoral dissertation, we consider Glut (1997) to be the earliest publication to spell the name Kritosaurini. The name was first (informally) defined by Prieto-Márquez (2014) who applied the minimum-clade definition and used Kritosaurus navajovius, Gryposaurus notabilis, and Naashoibitosaurus ostromi as the internal specifiers. We preserve the original intent of Prieto-Márquez (2014) but prefer to apply the maximum-clade definition. Kritosaurus navajovius is used as the internal specifier and Hadrosaurus foulkii, and representatives of Brachylophosaurini (Brachylophosaurus canadensis), Edmontosaurini (Edmontosaurus regalis), and Saurolophini (Saurolophus osborni), as the external specifiers.

Lambeosaurinae Parks, 1923 (converted clade name)

Registration number: 630

Definition. The largest clade containing Lambeosaurus lambei Parks, 1923 but not Hadrosaurus foulkii Leidy, 1858 and Saurolophus osborni Brown, 1912. This is a maximum-clade definition. Abbreviated definition: max ∇ (Lambeosaurus lambei Parks, 1923 ~ Hadrosaurus foulkii Leidy, 1858 & Saurolophus osborni Brown, 1912).

Reference phylogeny. Figure 18 of Prieto-Márquez, Wagner & Lehman (2020) is treated here as the primary reference phylogeny. Additional reference phylogenies include Figure 5 of Kobayashi et al. (2019), Figure 11 of Prieto-Márquez et al. (2019), Figure 9 of Zhang et al. (2019), Figure 5 of Zhang et al. (2020), Figure 7 of Kobayashi et al. (2021), and Figure 10 of Longrich et al. (2021).

Composition. Under the primary reference phylogeny, Lambeosaurinae comprises Aralosaurus tuberiferus, Canardia garonnensis, Jaxartosaurus aralensis, and members of the clades Corythosauria and Tsintaosaurini.

Synonyms. No other taxon names are currently in use for the same or approximate clade.

Comments. The name Lambeosaurinae has been (informally) defined before (Sereno, 1998; Sereno, 2005; Prieto-Márquez, 2010). These definitions were maximum-clade and used Parasaurolophus (Sereno, 1998) or Lambeosaurus lambei (Prieto-Márquez, 2010) as the internal specifiers and Saurolophus (Sereno, 1998) or Hadrosaurus foulkii, Saurolophus osborni, and Edmontosaurus regalis (Prieto-Márquez, 2010) as the external specifiers. Sereno (2005), apparently erroneously, defined Lambeosaurinae as pertaining to “(t)he most inclusive taxon containing Saurolophus osborni Brown, 1912 but not Parasaurolophus walkeri Parks, 1922 and including Lambeosaurus lambei Parks, 1923”. Our formal maximum-clade definition is similar to that of Prieto-Márquez (2010) though we have removed E. regalis from the external specifiers because the taxon is consistently inferred outside Lambeosaurinae (Kobayashi et al., 2019; Prieto-Márquez et al., 2019; Prieto-Márquez, Wagner & Lehman, 2020; Zhang et al., 2019; Zhang et al., 2020; Gates, Evans & Sertich, 2021; Kobayashi et al., 2021; Longrich et al., 2021; Ramírez-Velasco et al., 2021).

Lambeosaurini Sullivan et al., 2011 (converted clade name)

Registration number: 631

Definition. The largest clade containing Lambeosaurus lambei Parks, 1923 but not Aralosaurus tuberiferus Rozhdestvensky, 1968, Parasaurolophus walkeri Parks, 1922, and Tsintaosaurus spinorhinus Young, 1958. This is a maximum-clade definition. Abbreviated definition: max ∇ (Lambeosaurus lambei Parks, 1923 ~ Aralosaurus tuberiferus Rozhdestvensky, 1968 & Parasaurolophus walkeri Parks, 1922 & Tsintaosaurus spinorhinus Young, 1958).

Reference phylogeny. Figure 18 of Prieto-Márquez, Wagner & Lehman (2020) is treated here as the primary reference phylogeny. Additional reference phylogenies include Figure 5 of Kobayashi et al. (2019), Figure 11 of Prieto-Márquez et al. (2019), Figure 9 of Zhang et al. (2019), Figure 5 of Zhang et al. (2020), Figure 7 of Kobayashi et al. (2021), and Figure 10 of Longrich et al. (2021).

Composition. Under the primary reference phylogeny, Lambeosaurini comprises Amurosaurus riabinini, Arenysaurus ardevoli, Blasisaurus canudoi, Corythosaurus spp., Hypacrosaurus stebingeri, Hypacrosaurus altispinus, Lambeosaurus spp., Magnapaulia laticaudus, Olorotitan arharensis (misspelled as ‘ararhensis’ in the primary reference phylogeny), Sahaliyania elunchunorum, and Velafrons coahuilensis.

Synonyms. The name Corythosaurini Glut, 1997 is an approximate synonym of Lambeosaurini (e.g., Evans & Reisz, 2007; Gates et al., 2007; Pereda-Suberbiola et al., 2009). However, its use has been discouraged (Prieto-Márquez et al., 2013) and all recent phylogenetic studies preferred to use Lambeosaurini instead (e.g., Xing, Mallon & Currie, 2017; Kobayashi et al., 2019; Prieto-Márquez et al., 2019; Zhang et al., 2020; Kobayashi et al., 2021; Longrich et al., 2021; Ramírez-Velasco et al., 2021). No other taxon names are currently in use for the same or approximate clade.

Comments. Even though Sullivan et al. (2011) did not explicitly formulate the definition of their newly proposed name Lambeosaurini, they noted that their “definition of the Lambeosaurini would be equivalent to node 38 of Prieto-Márquez (2010: fig. 9)” (Sullivan et al., 2011: 417). The name Lambeosaurini was first (informally) defined by Prieto-Márquez et al. (2013) who applied the maximum-clade definition and used Lambeosaurus lambei as the internal specifier and Parasaurolophus walkeri, Tsintaosaurus spinorhinus, and Aralosaurus tuberiferus as the external specifier. Such defined, the use of Lambeosaurini adheres to the original intent of Sullivan et al. (2011). We formalize this definition.

Leptoceratopsidae Nopcsa, 1923 (converted clade name)

Registration number: 632

Definition. The largest clade containing Leptoceratops gracilis Brown, 1914b but not Protoceratops andrewsi Granger & Gregory, 1923 and Triceratops horridus Marsh, 1889. This is a maximum-clade definition. Abbreviated definition: max ∇ (Leptoceratops gracilis Brown, 1914b ~ Protoceratops andrewsi Granger & Gregory, 1923 & Triceratops horridus Marsh, 1889).

Reference phylogeny. Figure 10 of Morschhauser et al. (2019) is treated here as the primary reference phylogeny. Additional reference phylogenies include Figure S1 of Knapp et al. (2018), Figure 8A of Arbour & Evans (2019), Figure 3 of Yu et al. (2020), and Figure 4 of Yu et al. (2020).

Composition. Under the primary reference phylogeny, Leptoceratopsidae comprises Cerasinops hodgskissi, Gryphoceratops morrisoni, Helioceratops brachygnathus, Ischioceratops zhuchengensis, Koreaceratops hwaseongensis, Leptoceratops gracilis, Montanoceratops cerorhynchus, Prenoceratops pieganensis, Udanoceratops tchizhovi, Unescoceratops koppelhusae, and Zhuchengceratops inexpectus.

Synonyms. No other taxon names are currently in use for the same or approximate clade.

Comments. The name Leptoceratopsidae has been (informally) defined before by Makovicky (2001) who used Leptoceratops gracilis as the internal specifier and Triceratops horridus as the external specifier. Since Leptoceratopsidae has never been proposed an alternative use, we formalize a similar definition that differs only in adding Protoceratops andrewsi as a second external specifier.

Marginocephalia Sereno, 1986 (converted clade name)

Registration number: 633

Definition. The smallest clade containing Ceratops montanus Marsh, 1888, Pachycephalosaurus wyomingensis (Gilmore, 1931), and Triceratops horridus Marsh, 1889. This is a minimum-clade definition. Abbreviated definition: min ∇ (Ceratops montanus Marsh, 1888 & Pachycephalosaurus wyomingensis (Gilmore, 1931) & Triceratops horridus Marsh, 1889).

Reference phylogeny. Figure 16 of Han et al. (2018) is treated here as the primary reference phylogeny. Additional reference phylogenies include Figure 4 of Madzia, Boyd & Mazuch (2018), Figure 25 of Herne et al. (2019), Figure 1 of Dieudonné et al. (2020), Figure 12 of Yang et al. (2020), and Figure 57 of Barta & Norell (2021).

Composition. Under the primary reference phylogeny, Marginocephalia comprises members of the clades Ceratopsia and Pachycephalosauria.

Synonyms. No other taxon names are currently in use for the same or approximate clade.

Comments. The name Marginocephalia has been (informally) defined before (Currie & Padian, 1997; Sereno, 1998; Sereno, 2005; Madzia, Boyd & Mazuch, 2018; Herne et al., 2019). These definitions, except for that of Herne et al. (2019), were minimum-clade and used Ceratopsia and Pachycephalosauria (Currie & Padian, 1997) or Triceratops horridus and Pachycephalosaurus wyomingensis (Sereno, 1998; Sereno, 2005; Madzia, Boyd & Mazuch, 2018) as the internal specifiers. Madzia, Boyd & Mazuch (2018) further included Ceratops montanus as a third internal specifier, stating that “(t)he first definition of Marginocephalia was node-based and used ‘Ceratopsia’ and ‘Pachycephalosauria’ as the internal specifiers […]. To follow the definition, and adhere to the ICPN (Art. 11), we have to use name-bearing species or their type specimens as specifiers which makes the name to be anchored on the types of Ceratops montanus and Pachycephalosaurus wyomingensis. Even if C. montanus may be a nomen dubium, its type specimen is unequivocally nested deeply within Ceratopsia and thus its use does not change the extent of the name” (Madzia, Boyd & Mazuch, 2018: Appendix 1). In turn, Herne et al. (2019) preferred a maximum-clade definition with T. horridus and P. wyomingensis as the internal specifiers and Parasaurolophus walkeri as the external specifier, arguing that “(previous) definitions (were) not complementary with present definitions of Cerapoda and Ornithopoda within a node-stem triplet arrangement of clades” and that “re-definition of Marginocephalia as a stem now mirrors its sister stem clade, Ornithopoda, within a node-based Cerapoda. As a result, this stabilization of definition allows for the definitive assignment of all cerapodan OTUs either as ornithopods or marginocephalians” (Herne et al., 2019: Supplemental Text S1: 4). However, Marginocephalia has never formed such ‘triplet’. When its use in a ‘node-branch triplet’ is considered, it is more closely tied with Ceratopsia and Pachycephalosauria rather than with Cerapoda and Ornithopoda. Here, the internal specifiers in the definition of Marginocephalia are used from among the taxa representing the two major subclades – Ceratopsia (Ceratops montanus and Triceratops horridus) and Pachycephalosauria (Pachycephalosaurus wyomingensis). Note that none of the internal specifiers is included in the primary reference phylogeny. Ceratops montanus and Pachycephalosaurus wyomingensis are name-bearers of Ceratopsia and Pachycephalosauria, respectively, and are deeply nested within these clades (e.g., Mallon et al., 2016; Dieudonné et al., 2020). Triceratops horridus is a late-diverging member of Chasmosaurinae within Ceratopsia (e.g., Morschhauser et al., 2019; Fowler & Freedman Fowler, 2020).

Nasutoceratopsini Ryan et al., 2017 (converted clade name)

Registration number: 689

Definition. The largest clade containing Nasutoceratops titusi Sampson et al., 2013 but not Centrosaurus apertus Lambe, 1905. This is a maximum-clade definition. Abbreviated definition: max ∇ (Nasutoceratops titusi Sampson et al., 2013 ~ Centrosaurus apertus Lambe, 1905).

Reference phylogeny. Figure 9 of Chiba et al. (2018) is treated here as the primary reference phylogeny. Additional reference phylogenies include Figure 7 of Fiorillo & Tykoski (2012), Figure 10 of Ryan et al. (2017), and Figure 13 of Dalman et al. (2018).

Composition. Under the primary reference phylogeny, Nasutoceratopsini comprises Avaceratops lammersi, Nasutoceratops titusi, and the specimens CMN 8804, MOR 692, and the ‘Malta New Taxon’ (GPDM 63). Under an alternative hypothesis, however, Nasutoceratopsini includes only a single unequivocal member, Nasutoceratops titusi (Dalman et al., 2021: Fig. 23).

Synonyms. No other taxon names are currently in use for the same or approximate clade.

Comments. The name was first (informally) defined by Ryan et al. (2017) who applied the maximum-clade definition and used Nasutoceratops titusi as the internal specifier and Centrosaurus apertus as the external specifier. We formalize this definition.

Neoceratopsia Sereno, 1986 (converted clade name)

Registration number: 634

Definition. The largest clade containing Triceratops horridus Marsh, 1889 but not Chaoyangsaurus youngi Zhao, Cheng & Xu, 1999 and Psittacosaurus mongoliensis Osborn, 1923. This is a maximum-clade definition. Abbreviated definition: max ∇ (Triceratops horridus Marsh, 1889 ~ Chaoyangsaurus youngi Zhao, Cheng & Xu, 1999 & Psittacosaurus mongoliensis Osborn, 1923).

Reference phylogeny. Figure 10 of Morschhauser et al. (2019) is treated here as the primary reference phylogeny. Additional reference phylogenies include Figure 16 of Han et al. (2018), Figure S1 of Knapp et al. (2018), and Figure 4 of Yu et al. (2020).

Composition. Under the primary reference phylogeny, Neoceratopsia comprises Aquilops americanus, Archaeoceratops oshimai, Asiaceratops salsopaludalis, Auroraceratops rugosus, ZPAL MgD-I/156 (= Graciliceratops mongoliensis), Liaoceratops yanzigouensis, Mosaiceratops azumai, Stenopelix valdensis, Yamaceratops dorngobiensis, and members of the clade Euceratopsia.

Synonyms. No other taxon names are currently in use for the same or approximate clade.

Comments. The name Neoceratopsia has been (informally) defined before by Sereno (1998, 2005) who applied a maximum-clade definition and used Triceratops horridus as the internal specifier and Psittacosaurus mongoliensis as the external specifier. We further include a second external specifier, Chaoyangsaurus youngi, to ensure that Chaoyangsauridae, a clade usually reconstructed as some of the earliest-diverging ceratopsians (e.g., Han et al., 2018; Knapp et al., 2018; Yu et al., 2020), are maintained outside Neoceratopsia.

Neoiguanodontia Norman, 2014 (converted clade name)

Registration number: 635

Definition. The smallest clade containing Hypselospinus fittoni (Lydekker, 1889), Iguanodon bernissartensis Boulenger in Beneden, 1881, and Parasaurolophus walkeri Parks, 1922. This is a minimum-clade definition. Abbreviated definition: min ∇ (Hypselospinus fittoni (Lydekker, 1889) & Iguanodon bernissartensis Boulenger in Beneden, 1881 & Parasaurolophus walkeri Parks, 1922).

Reference phylogeny. Figure 2.26 of Norman (2014) is treated here as the primary reference phylogeny. Additional reference phylogenies include Figure 50 of Norman (2015), Figure 3 of Párraga & Prieto-Márquez (2019), and Figure 11 of McDonald et al. (2021).

Composition. Under the primary reference phylogeny, Neoiguanodontia comprises Hypselospinus fittoni and members of the clade Hadrosauriformes.

Synonyms. Neoiguanodontia is a potential heterodefinitional synonym of Hadrosauriformes. If Hypselospinus fittoni nests within the smallest clade containing Hadrosaurus foulkii and Iguanodon bernissartensis (e.g., Verdú et al., 2018; Santos-Cubedo et al., 2021: Fig. 11), the name Hadrosauriformes should have priority.

Comments. The application of Neoiguanodontia has been described and discussed by Madzia, Jagt & Mulder (2020: Table 1). We therefore refer to that study for details.

Neornithischia Cooper, 1985 (converted clade name)

Registration number: 636

Definition. The largest clade containing Iguanodon bernissartensis Boulenger in Beneden, 1881 and Triceratops horridus Marsh, 1889 but not Ankylosaurus magniventris Brown, 1908 and Stegosaurus stenops Marsh, 1887. This is a maximum-clade definition. Abbreviated definition: max ∇ (Iguanodon bernissartensis Boulenger in Beneden, 1881 & Triceratops horridus Marsh, 1889 ~ Ankylosaurus magniventris Brown, 1908 & Stegosaurus stenops Marsh, 1887).

Reference phylogeny. Figure 4 of Madzia, Boyd & Mazuch (2018) is treated here as the primary reference phylogeny. Additional reference phylogenies include Figure 16 of Han et al. (2018), Figure 25 of Herne et al. (2019), Figure 1 of Dieudonné et al. (2020), and Figure 57 of Barta & Norell (2021).

Composition. Under the primary reference phylogeny, Neornithischia comprises Agilisaurus louderbacki, Hexinlusaurus multidens, Hypsilophodon foxii, Kulindadromeus zabaikalicus, Leaellynasaura amicagraphica, Lesothosaurus diagnosticus, Othnielosaurus consors (= Nanosaurus agilis; see Carpenter & Galton, 2018), Yandusaurus hongheensis, and members of the clades Cerapoda, Jeholosauridae, and Thescelosauridae.

Synonyms. No other taxon names are currently in use for the same or approximate clade.

Comments. The name Neornithischia has been (informally) defined before (Sereno, 1998; Sereno, 2005; Butler, Upchurch & Norman, 2008; Herne et al., 2019). These definitions were maximum-clade and used Triceratops horridus (Sereno, 1998), Parasaurolophus walkeri (Butler, Upchurch & Norman, 2008) or both, T. horridus and P. walkeri (Sereno, 2005; Herne et al., 2019) as the internal specifiers, and Ankylosaurus magniventris (Sereno, 1998; Sereno, 2005; Herne et al., 2019) or A. magniventris and Stegosaurus stenops (Butler, Upchurch & Norman, 2008) as the external specifiers. In order to maintain the ‘traditional’ concept of Genasauria as a clade comprising Neornithischia and Thyreophora, the internal specifiers in the definition of Neornithischia are used from among the taxa representing the two major subclades – Ornithopoda (Iguanodon bernissartensis) and Marginocephalia (Triceratops horridus) – and the external specifiers are used from among the taxa representing the thyreophoran clades Ankylosauria (Ankylosaurus magniventris) and Stegosauria (Stegosaurus stenops). Note that the internal specifier Triceratops horridus and the external specifiers Ankylosaurus magniventris and Stegosaurus stenops are not included in the primary reference phylogeny. T. horridus belongs to Ceratopsia (see, e.g., Morschhauser et al., 2019), while Ankylosaurus magniventris and Stegosaurus stenops are deeply nested members of Thyreophora (e.g., Thompson et al., 2012; Maidment et al., 2020), a clade that is indicated on Figure 4 of Madzia, Boyd & Mazuch (2018).

Nodosauridae Marsh, 1890 (converted clade name)

Registration number: 637

Definition. The largest clade containing Nodosaurus textilis Marsh, 1889 but not Ankylosaurus magniventris Brown, 1908. This is a maximum-clade definition. Abbreviated definition: max ∇ (Nodosaurus textilis Marsh, 1889 ~ Ankylosaurus magniventris Brown, 1908).

Reference phylogeny. Figure 5 of Rivera-Sylva et al. (2018a) is treated here as the primary reference phylogeny. Additional reference phylogenies include Figure 3 of Thompson et al. (2012), Figure 11 of Arbour & Currie (2016), Figure 1 of Arbour, Zanno & Gates (2016), and Figure 3 of Brown et al. (2017).

Composition. Under the primary reference phylogeny, Nodosauridae comprises Dongyangopelta yangyanensis, Gastonia burgei, Gargoyleosaurus parkpinorum, and members of the clades Nodosaurinae and Polacanthinae.

Synonyms. No other taxon names are currently in use for the same or approximate clade.

Comments. The name Nodosauridae has been (informally) defined before by Sereno (1998, 2005) who used Panoplosaurus mirus (Sereno, 1998) or Panoplosaurus mirus and Nodosaurus textilis Sereno (2005) as the internal specifiers and Ankylosaurus magniventris as the external specifier. Considering that all phylogeny reconstructions that include P. mirus and N. textilis indicate that these taxa are more closely related to each other than either is to A. magniventris (or placed outside the Ankylosauridae + Nodosauridae node), we use a definition that incorporates Nodosaurus textilis as the sole internal specifier.

Nodosaurinae Abel, 1919 (converted clade name)

Registration number: 638

Definition. The largest clade containing Nodosaurus textilis Marsh, 1889, but not Hylaeosaurus armatus Mantell, 1833, Mymoorapelta maysi Kirkland & Carpenter, 1994, and Polacanthus foxii Owen in Anonymous, 1865. This is a maximum-clade definition. Abbreviated definition: max ∇ (Nodosaurus textilis Marsh, 1889 ~ Hylaeosaurus armatus Mantell, 1833 & Mymoorapelta maysi Kirkland & Carpenter, 1994 & Polacanthus foxii Owen in Anonymous, 1865).

Reference phylogeny. Figure 5 of Rivera-Sylva et al. (2018a) is treated here as the primary reference phylogeny. Additional reference phylogenies include Figure 3 of Thompson et al. (2012), Figure 11 of Arbour & Currie (2016), Figure 1 of Arbour, Zanno & Gates (2016), and Figure 3 of Brown et al. (2017).

Composition. Under the primary reference phylogeny, Nodosaurinae comprises Acantholipan gonzalezi, Ahshislepelta minor, Niobrarasaurus coleii, Nodosaurus textilis, Peloroplites cedrimontanus, Sauropelta edwardsi, Silvisaurus condrayi, Taohelong jinchengensis, Tatankacephalus cooneyorum, members of the clades Panoplosaurini and Struthiosaurini, and the specimen CPC 273.

Synonyms. No other taxon names are currently in use for the same or approximate clade.

Comments. The name Nodosaurinae has been (informally) defined before (Sereno, 1998; Sereno, 2005). Both these definitions were maximum-clade and used Panoplosaurus (Sereno, 1998) or Panoplosaurus mirus and Nodosaurus textilis (Sereno, 2005) as the internal specifiers and Sarcolestes and Hylaeosaurus (Sereno, 1998) or Polacanthus foxii, Hylaeosaurus armatus, and Mymoorapelta maysi as the external specifiers (Sereno, 2005). We formalize a definition similar to that of Sereno (2005) but use a single internal specifier.

Ornithischia Seeley, 1888 (converted clade name)

Registration number: 639

Definition. The largest clade containing Iguanodon bernissartensis Boulenger in Beneden, 1881 but not Allosaurus fragilis Marsh, 1877a and Camarasaurus supremus Cope, 1877. This is a maximum-clade definition. Abbreviated definition: max ∇ (Iguanodon bernissartensis Boulenger in Beneden, 1881 ~ Allosaurus fragilis Marsh, 1877a & Camarasaurus supremus Cope, 1877).

Reference phylogeny. Figure 4 of Madzia, Boyd & Mazuch (2018) is treated here as the primary reference phylogeny. Additional reference phylogenies include Figure 2 of Boyd (2015), Figure 1 of Baron, Norman & Barrett (2017a), Figure 1 of Baron, Norman & Barrett (2017b), and Figure 1 of Langer et al. (2017).

Composition. Under the primary reference phylogeny, Ornithischia comprises Pisanosaurus mertii and members of the clades Heterodontosauridae and Genasauria. Note, however, that the early evolution and basal branching of Ornithischia is currently unsettled. For example, P. mertii may represent either an early-diverging ornithischian (recently, e.g., Desojo et al., 2020) or a (non-dinosaur) silesaurid dinosauriform (Agnolín & Rozadilla, 2018, Baron, 2019). The same may be true for members of Silesauridae, a group often reconstructed as the sister taxon to Dinosauria (e.g., Nesbitt et al., 2010; Peecook et al., 2013; Ezcurra, 2016; Cau, 2018; Ezcurra et al., 2020), that have recently been inferred to represent early-diverging representatives of Ornithischia (Langer & Ferigolo, 2013; Cabreira et al., 2016; Pacheco et al., 2019; Müller & Garcia, 2020).

Synonyms. No other taxon names are currently in use for the same or approximate clade. Though not in use, the name Predentata Marsh, 1894 is being occasionally recalled as an approximate synonym (e.g., Butler, Upchurch & Norman, 2008; Langer & Ferigolo, 2013).

Comments. The name Ornithischia has been (informally) defined before (Padian & May, 1993; Sereno, 1998; Weishampel, 2004; Norman, Witmer & Weishampel, 2004; Sereno, 2005; Baron, Norman & Barrett, 2017a). These definitions were maximum-clade and used Triceratops horridus (Padian & May, 1993; Sereno, 1998; Weishampel, 2004; Sereno, 2005; Baron, Norman & Barrett, 2017a) or Iguanodon bernissartensis (Norman, Witmer & Weishampel, 2004) as the internal specifiers. In turn, “birds” (Padian & May, 1993), Neornithes (Sereno, 1998), Tyrannosaurus (Weishampel, 2004), Cetiosaurus (Norman, Witmer & Weishampel, 2004), Passer domesticus and Saltasaurus loricatus (Sereno, 2005), and Passer domesticus and Diplodocus carnegii (Baron, Norman & Barrett, 2017a) were used as the external specifiers. Although both, I. bernissartensis and T. horridus, are clearly ‘traditional’ members of Ornithischia, we have selected the former as the internal specifier and Allosaurus fragilis and Camarasaurus supremus as the external specifiers. These specifiers are preferred because (a) they represent deeply nested taxa within their respective clades (Ornithischia, Theropoda, and Sauropodomorpha), (b) they have been historically associated with these clades, thus being their ‘traditional’ members, and (c) their phylogenetic placements are stable across studies. Two external specifiers, instead of one, are used due to the alternative topologies of dinosaur relationships (see, e.g., Baron, Norman & Barrett, 2017a; Langer et al., 2017). Additionally, Iguanodon bernissartensis was used as the internal specifier in the formal definition of Dinosauria (Langer et al., 2020), considered therein as a ‘traditional’ representative of Ornithischia, and the external specifier in the formal definition of Sauropodomorpha (Fabbri et al., 2020), again considered therein as a ‘traditional’ representative of Ornithischia; A. fragilis was used as the internal specifier in the formal definitions of Theropoda (Naish et al., 2020) and Saurischia (Gauthier et al., 2020), and as the external specifier in the formal definition of Sauropodomorpha (Fabbri et al., 2020), considered in these contributions as a ‘traditional’ representative of Theropoda; and Camarasaurus supremus was used as the internal specifier in the formal definition of Saurischia (Gauthier et al., 2020) and considered therein as a ‘traditional’ representative of Sauropodomorpha.

Ornithopoda Marsh, 1881 (converted clade name)

Registration number: 640

Definition. The largest clade containing Iguanodon bernissartensis Boulenger in Beneden, 1881 but not Pachycephalosaurus wyomingensis (Gilmore, 1931) and Triceratops horridus Marsh, 1889. This is a maximum-clade definition. Abbreviated definition: max ∇ (Iguanodon bernissartensis Boulenger in Beneden, 1881 ~ Pachycephalosaurus wyomingensis (Gilmore, 1931) & Triceratops horridus Marsh, 1889).

Reference phylogeny. Figure 4 of Madzia, Boyd & Mazuch (2018) is treated here as the primary reference phylogeny. Additional reference phylogenies include Figure 16 of Han et al. (2018), Figure 25 of Herne et al. (2019), Figure 1 of Dieudonné et al. (2020), and Figure 57 of Barta & Norell (2021).

Composition. Under the primary reference phylogeny, Ornithopoda comprises Burianosaurus augustai, Gideonmantellia amosanjuanae, and members of the clades Elasmaria and Iguanodontia.

Synonyms. No other taxon names are currently in use for the same or approximate clade.

Comments. The name Ornithopoda has been (informally) defined before (Sereno, 1998; Norman et al., 2004; Sereno, 2005; Butler, Upchurch & Norman, 2008; Herne et al., 2019). Two of these definitions were minimum-clade (Sereno, 1998; Sereno, 2005) and used Parasaurolophus walkeri and Heterodontosaurus tucki as the internal specifiers. Sereno (2005) further restricted the name to a hypothesis in which P. walkeri and H. tucki were more closely related to each other than either was to Pachycephalosaurus wyomingensis, Triceratops horridus, and Ankylosaurus magniventris. In turn, Norman et al. (2004), Butler, Upchurch & Norman (2008), and Herne et al. (2019) defined Ornithopoda as pertaining to the largest clade containing Edmontosaurus regalis (in Norman et al., 2004) or P. walkeri (in Butler, Upchurch & Norman, 2008 and Herne et al., 2019) but not T. horridus. Herne et al. (2019) additionally included a second external specifier (P. wyomingensis). We selected a definition that follows Herne et al. (2019) in that it includes two external specifiers (T. horridus and P. wyomingensis, representatives of two clades closely related to ornithopods; i.e., Ceratopsia and Pachycephalosauria, respectively). However, we prefer to use Iguanodon bernissartensis as the internal specifier rather than P. walkeri, because the former is among the few taxa that have been considered a part of Ornithopoda when the name was being introduced in the literature (e.g., Marsh, 1882). The inclusion of a different internal specifier does not change the extent of Ornithopoda under any of the published phylogeny inferences. Note that the external specifiers Pachycephalosaurus wyomingensis and Triceratops horridus are not included in the primary reference phylogeny. The former belongs to Pachycephalosauria (see, e.g., Dieudonné et al., 2020), while the latter is part of Ceratopsia (e.g., Morschhauser et al., 2019), both within Marginocephalia that is indicated on Figure 4 of Madzia, Boyd & Mazuch (2018).

Orodrominae Brown et al., 2013 (converted clade name)

Registration number: 641

Definition. The largest clade within Hypsilophodontidae ∨ Thescelosauridae containing Orodromeus makelai Horner & Weishampel, 1988 but not Hypsilophodon foxii Huxley, 1869 and Thescelosaurus neglectus Gilmore, 1913. This is a maximum-clade definition. Abbreviated definition: max ∇ ∈ Hypsilophodontidae ∨ Thescelosauridae (Orodromeus makelai Horner & Weishampel, 1988 ~ Hypsilophodon foxii Huxley, 1869 & Thescelosaurus neglectus Gilmore, 1913).

Reference phylogeny. Figure 4 of Madzia, Boyd & Mazuch (2018) is treated here as the primary reference phylogeny. Additional reference phylogenies include Figure 25 of Herne et al. (2019) and Figure 57 of Barta & Norell (2021).

Composition. Under the primary reference phylogeny, Orodrominae comprises Albertadromeus syntarsus, Changchunsaurus parvus, Haya griva, Koreanosaurus boseongensis, Orodromeus makelai, Oryctodromeus cubicularis, Zephyrosaurus schaffi, and the ‘Kaiparowits orodromine’.

Synonyms. No other taxon names are currently in use for the same or approximate clade.

Comments. The name Orodrominae has been (informally) defined before (Brown et al., 2013; Boyd, 2015). Both these definitions were maximum-clade and used Orodromeus makelai as the internal specifier and Thescelosaurus neglectus (Brown et al., 2013) or Thescelosaurus neglectus and Parasaurolophus walkeri (Boyd, 2015) as the external specifiers. Considering the ‘traditional concept’ of Orodrominae, as a subclade of Thescelosauridae/‘hypsilophodonts’, and keeping in mind the unstable phylogenetic position of H. foxii (e.g., Madzia, Boyd & Mazuch, 2018), we apply Orodrominae only when it is inferred either within Thescelosauridae or Hypsilophodontidae (see Article 11.14 of the ICPN).

Pachycephalosauria Maryańska & Osmólska, 1974 (converted clade name)

Registration number: 642

Definition. The largest clade containing Pachycephalosaurus wyomingensis (Gilmore, 1931) but not Ceratops montanus Marsh, 1888 and Triceratops horridus Marsh, 1889. This is a maximum-clade definition. Abbreviated definition: max ∇ (Pachycephalosaurus wyomingensis (Gilmore, 1931) ~ Ceratops montanus Marsh, 1888 & Triceratops horridus Marsh, 1889).

Reference phylogeny. Figure 27 of Schott & Evans (2017) is treated here as the primary reference phylogeny. Additional reference phylogenies include Figure 5 of Evans et al. (2013), Figure 16 of Han et al. (2018), and Figure 1 of Dieudonné et al. (2020).

Composition. Under the primary reference phylogeny, Pachycephalosauria comprises Wannanosaurus yanshiensis and members of the clade Pachycephalosauridae.

Synonyms. No other taxon names are currently in use for the same or approximate clade.

Comments. The name Pachycephalosauria has been (informally) defined before (Sereno, 1998; Maryańska, Chapman & Weishampel, 2004; Sereno, 2005). These definitions were maximum-clade and used Pachycephalosaurus (Sereno, 1998) or Pachycephalosaurus wyomingensis (Maryańska, Chapman & Weishampel, 2004; Sereno, 2005) as the internal specifier and Triceratops (Sereno, 1998), Triceratops horridus (Maryańska, Chapman & Weishampel, 2004), or Triceratops horridus, Heterodontosaurus tucki, Hypsilophodon foxii, and Ankylosaurus magniventris (Sereno, 2005) as the external specifiers. Even though the position of Hypsilophodon foxii and Heterodontosaurus tucki is unstable across studies (e.g., see, e.g., Han et al., 2018; Madzia, Boyd & Mazuch, 2018; Herne et al., 2019; Dieudonné et al., 2020), and, for example, Heterodontosauridae were inferred to be more closely related to P. wyomingensis than to T. horridus (Dieudonné et al., 2020: Fig. 1), inclusion of these taxa among the external specifiers does not need to be necessary as it can be expected that Pachycephalosauria, as ‘traditionally’ defined, may cover taxa that are markedly different from the Late Cretaceous members of the clade. We use a definition similar to that of Maryańska, Chapman & Weishampel (2004) but include Ceratops montanus as a second external specifier. Note that none of the external specifiers is included in the primary reference phylogeny. Both, C. montanus and T. horridus, however, are members of Ceratopsidae within Ceratopsia (e.g., Mallon et al., 2016; Morschhauser et al., 2019).

Pachycephalosauridae Sternberg, 1945 (converted clade name)

Registration number: 643

Definition. The smallest clade containing Pachycephalosaurus wyomingensis (Gilmore, 1931) and Stegoceras validum Lambe, 1902, provided that it does not include Heterodontosaurus tucki Crompton & Charig, 1962. This is a minimum-clade definition. Abbreviated definition: min ∇ (Pachycephalosaurus wyomingensis (Gilmore, 1931) & Stegoceras validum Lambe, 1902 | ~ Heterodontosaurus tucki Crompton & Charig, 1962).

Reference phylogeny. Figure 27 of Schott & Evans (2017) is treated here as the primary reference phylogeny. Additional reference phylogenies include Figure 5 of Evans et al. (2013), Figure 3 of Williamson & Brusatte (2016), and Figure 14 of Woodruff et al. (2021).

Composition. Under the primary reference phylogeny, Pachycephalosauridae comprises Colepiocephale lambei, Hanssuesia sternbergi, Stegoceras spp., and members of the clade Pachycephalosaurinae.

Synonyms. No other taxon names are currently in use for the same or approximate clade.

Comments. The name Pachycephalosauridae has been (informally) defined before by Sereno (1998, 2005) who applied the minimum-clade definition and used Pachycephalosaurus wyomingensis and Stegoceras validum as the internal specifiers. This definition is followed here though we also include a qualifying clause that excludes H. tucki from Pachycephalosauridae. Even though no phylogenetic analysis has ever reconstructed H. tucki or any other ‘traditional’ heterodontosaurid to be within the smallest clade containing P. wyomingensis and S. validum, Heterodontosauridae were inferred to be early-diverging pachycephalosaurs (Dieudonné et al., 2020). The addition of a qualifying clause that excludes H. tucki from Pachycephalosauridae will therefore ensure that the name will never comprise Heterodontosauridae. Note that H. tucki is not included in the primary reference phylogeny. See Figure 1 of Dieudonné et al. (2020) for its potential placement with respect to pachycephalosaurids.

Pachycephalosaurinae Sereno, 1997 (converted clade name)

Registration number: 748

Definition. The largest clade containing Pachycephalosaurus wyomingensis (Gilmore, 1931) but not Stegoceras validum Lambe, 1902. This is a maximum-clade definition. Abbreviated definition: max ∇ (Pachycephalosaurus wyomingensis (Gilmore, 1931) ~ Stegoceras validum Lambe, 1902).

Reference phylogeny. Figure 27 of Schott & Evans (2017) is treated here as the primary reference phylogeny. Additional reference phylogenies include Figure 7 of Longrich, Sankey & Tanke (2010), Figure 5 of Evans et al. (2013), Figure 3 of Williamson & Brusatte (2016), and Figure 14 of Woodruff et al. (2021).

Composition. Under the primary reference phylogeny, Pachycephalosaurinae comprises Acrotholus audeti, Amtocephale gobiensis, Foraminacephale brevis, Goyocephale lattimorei, Homalocephale calathocercos, Prenocephale prenes, Sphaerotholus spp., Tylocephale gilmorei, and members of the clade Pachycephalosaurini.

Synonyms. No other taxon names are currently in use for the same or approximate clade.

Comments. The name Pachycephalosaurinae has been (informally) defined before (Sereno, 1998; Sullivan, 2003; Sereno, 2005). Both types of definitions, minimum-clade as well as maximum-clade, have been proposed for the name. Sereno (1998, 2005) preferred a maximum-clade definition and used Pachycephalosaurus (Sereno, 1998) or Pachycephalosaurus wyomingensis (Sereno, 2005) as the internal specifier and Stegoceras (Sereno, 1998) or Stegoceras validum (Sereno, 2005) as the external specifier, while Sullivan (2003) applied a minimum-clade definition, using Colepiocephale, Prenocephale, Tylocephale, Hanssuesia, and Pachycephalosaurus and Stygimoloch (Pachycephalosaurini sensu Sullivan, 2003), as the internal specifiers. We formalize the definition of Sereno (2005).

Pachycephalosaurini Sullivan, 2003 (converted clade name)

Registration number: 749

Definition. The largest clade containing Pachycephalosaurus wyomingensis (Gilmore, 1931) but not Prenocephale prenes Maryańska & Osmólska, 1974 and Sphaerotholus goodwini Williamson & Carr, 2003. This is a maximum-clade definition. Abbreviated definition: max ∇ (Pachycephalosaurus wyomingensis (Gilmore, 1931) ~ Prenocephale prenes Maryańska & Osmólska, 1974 & Sphaerotholus goodwini Williamson & Carr, 2003).

Reference phylogeny. Figure 27 of Schott & Evans (2017) is treated here as the primary reference phylogeny. Additional reference phylogenies include Figure 7 of Longrich, Sankey & Tanke (2010), Figure 5 of Evans et al. (2013), Figure 3 of Williamson & Brusatte (2016), and Figure 14 of Woodruff et al. (2021).

Composition. Under the primary reference phylogeny, Pachycephalosaurini comprises Alaskacephale gongloffi, Dracorex hogwartsia, Pachycephalosaurus wyomingensis, and Stygimoloch spinifer.

Synonyms. No other taxon names are currently in use for the same or approximate clade.

Comments. The name Pachycephalosaurini has been (informally) defined before by Sereno (2005) who applied a minimum-clade definition and used Pachycephalosaurus wyomingensis and Stygimoloch spinifer as the internal specifiers. Even though such definition is congruent with the original intent of Sullivan (2003) who established the taxon name to unite S. spinifer and P. wyomingensis, it has since been hypothesized that S. spinifer and P. wyomingensis may represent different growth stages of a single taxon (P. wyomingensis) rather than two distinct taxa (Horner & Goodwin, 2009). Nevertheless, the name Pachycephalosaurini may still be considered useful as recent studies indicate close relationships between P. wyomingensis and Alaskacephale gangloffi (Longrich, Sankey & Tanke, 2010; Evans et al., 2013; Williamson & Brusatte, 2016; Schott & Evans, 2017; Woodruff et al., 2021). Owing to the unsettled phylogenetic ties between the latest Cretaceous pachycephalosaurs, we prefer to establish a maximum-clade definition for Pachycephalosaurini to enable the name to be used for a wider range of late-diverging members of Pachycephalosauridae.

Pachyrhinosaurini Fiorillo & Tykoski, 2012 (converted clade name)

Registration number: 690

Definition. The largest clade containing Pachyrhinosaurus canadensis Sternberg, 1950 but not Centrosaurus apertus Lambe, 1905. This is a maximum-clade definition. Abbreviated definition: max ∇ (Pachyrhinosaurus canadensis Sternberg, 1950 ~ Centrosaurus apertus Lambe, 1905).

Reference phylogeny. Figure 9 of Chiba et al. (2018) is treated here as the primary reference phylogeny. Additional reference phylogenies include Figure 7 of Fiorillo & Tykoski (2012), Figure 10 of Ryan et al. (2017), Figure 13 of Dalman et al. (2018), Figure 10 of Wilson, Ryan & Evans (2020), and Figure 23 of Dalman et al. (2021).

Composition. Under the primary reference phylogeny, Pachyrhinosaurini comprises Einiosaurus procurvicornis and members of the clade Pachyrostra.

Synonyms. No other taxon names are currently in use for the same or approximate clade.

Comments. The name was first (informally) defined by Fiorillo & Tykoski (2012) who applied the maximum-clade definition and used Pachyrhinosaurus canadensis as the internal specifier and Centrosaurus apertus as the external specifier. We formalize this definition.

Pachyrostra Fiorillo & Tykoski, 2012 (converted clade name)

Registration number: 691

Definition. The smallest clade containing Achelousaurus horneri Sampson, 1995 and Pachyrhinosaurus canadensis Sternberg, 1950. This is a minimum-clade definition. Abbreviated definition: min ∇ (Achelousaurus horneri Sampson, 1995 & Pachyrhinosaurus canadensis Sternberg, 1950).

Reference phylogeny. Figure 9 of Chiba et al. (2018) is treated here as the primary reference phylogeny. Additional reference phylogenies include Figure 7 of Fiorillo & Tykoski (2012), Figure 10 of Ryan et al. (2017), Figure 13 of Dalman et al. (2018), Figure 10 of Wilson, Ryan & Evans (2020), and Figure 23 of Dalman et al. (2021).

Composition. Under the primary reference phylogeny, Pachyrostra comprises Achelousaurus horneri and Pachyrhinosaurus spp.

Synonyms. No other taxon names are currently in use for the same or approximate clade.

Comments. The name was first (informally) defined by Fiorillo & Tykoski (2012) who applied the minimum-clade definition and used Achelousaurus horneri and Pachyrhinosaurus canadensis as the internal specifiers. We formalize this definition.

Panoplosaurini (new clade name)

Registration number: 644

Definition. The largest clade containing Panoplosaurus mirus Lambe, 1919 but not Nodosaurus textilis Marsh, 1889 and Struthiosaurus austriacus Bunzel, 1871. This is a maximum-clade definition. Abbreviated definition: max ∇ (Panoplosaurus mirus Lambe, 1919 ~ Nodosaurus textilis Marsh, 1889 & Struthiosaurus austriacus Bunzel, 1871).

Etymology. Derived from the stem of Panoplosaurus Lambe, 1919, the name of an included taxon, which combines the Greek words pan (all), hoplon (type of shield), and sauros (lizard, reptile).

Reference phylogeny. Figure 5 of Rivera-Sylva et al. (2018a) is treated here as the primary reference phylogeny. Additional reference phylogenies include Figure 1 of Arbour, Zanno & Gates (2016), Figure 3 of Brown et al. (2017), and Figure 9 of Zheng et al. (2018).

Composition. Under the primary reference phylogeny, Panoplosaurini comprises Animantarx ramaljonesi, ‘Denversaurus’ schlessmani, Edmontonia longiceps, Edmontonia rugosidens, Panoplosaurus mirus, Texasetes pleurohalio, and the ‘Argentinian ankylosaur’.

Synonyms. The name Panoplosaurinae Nopcsa, 1929 has been recently suggested for the same clade (e.g., Rivera-Sylva et al., 2018a; see also ‘Comments’ below). Additionally, Bakker (1988) coined the name Edmontoniinae for Edmontonia rugosidens, Edmontonia longiceps, and Denversaurus schlessmani and Edmontoniidae to include Edmontoniinae and Panoplosaurinae; no phylogenetic definition was proposed for either and neither clade name has been widely used since.

Comments. The grouping, here covered under the name Panoplosaurini, has previously been suggested to be named Panoplosaurinae (Rivera-Sylva et al., 2018a). No (informal) phylogenetic definition for Panoplosaurinae has ever been published in the peer-reviewed literature, though Burns (2015) proposed “all Late Cretaceous nodosaurids more closely related to Panoplosaurus than to Pawpawsaurus” in his dissertation, and the name itself has not been widely used. Bakker (1988) provided a diagnosis of Panoplosaurinae, as nodosaurids with lumpy armor and expanded internarial bridges, which contained the two species of Panoplosaurus he recognized (Panoplosaurus mirus and Panoplosaurus sp. 1, represented by ROM 1215). Alpha taxonomic reviews of the Campanian-Maastrichtian North American nodosaurids generally recognize Panoplosaurus mirus, Edmontonia rugosidens, and Edmontonia longiceps as valid taxa (e.g., Carpenter, 2001) and these typically form a clade (e.g. Kirkland, 1998, Vickaryous, Maryanska & Weishampel, 2004; Thompson et al., 2012; Yang et al., 2013), sometimes with additional taxa such as Texasetes (Arbour, Zanno & Gates, 2016; Rivera-Sylva et al., 2018a) or Animantarx (Hill, Witmer & Norell, 2003). Rivera-Sylva et al. (2018a) noted that the grouping Animantarx, ‘Denversaurus’, Edmontonia, Panoplosaurus, Texasetes, and an unnamed Argentinian ankylosaur could bear the name Panoplosaurinae. In several recent analyses Edmontonia and Panoplosaurus are found as the sister clade to a clade containing Struthiosaurus (Arbour, Zanno & Gates, 2016; Brown et al., 2017; Rivera-Sylva et al., 2018a), here named Struthiosaurini (see the name entry). Owing to the fact that the ‘Panoplosaurus clade’ is nested within Nodosaurinae, we prefer to use a name that implies a lesser inclusiveness. The suffix -inae (as in Panoplosaurinae) is typically associated with the rank of ‘subfamily’ under the ICZN. Therefore, the use of the suffix ‘-inae’ for the ‘Panoplosaurus clade’, without discussing the phylogenetic context, may suggest that Panoplosaurinae represents a clade outside Nodosaurinae. When the widely used suffix -ini (typically associated with the rank of ‘tribe’) is applied, such confusion is eliminated.

Parasaurolophini Glut, 1997 (converted clade name)

Registration number: 645

Definition. The largest clade containing Parasaurolophus walkeri Parks, 1922 but not Aralosaurus tuberiferus Rozhdestvensky, 1968, Lambeosaurus lambei Parks, 1923 and Tsintaosaurus spinorhinus Young, 1958. This is a maximum-clade definition. Abbreviated definition: max ∇ (Parasaurolophus walkeri Parks, 1922 ~ Aralosaurus tuberiferus Rozhdestvensky, 1968 & Lambeosaurus lambei Parks, 1923 & Tsintaosaurus spinorhinus Young, 1958).

Reference phylogeny. Figure 18 of Prieto-Márquez, Wagner & Lehman (2020) is treated here as the primary reference phylogeny. Additional reference phylogenies include Figure 5 of Kobayashi et al. (2019), Figure 11 of Prieto-Márquez et al. (2019), Figure 9 of Zhang et al. (2019), Figure 5 of Zhang et al. (2020), Figure 7 of Kobayashi et al. (2021), and Figure 10 of Longrich et al. (2021).

Composition. Under the primary reference phylogeny, Parasaurolophini comprises Charonosaurus jiayinensis and Parasaurolophus spp.

Synonyms. No other taxon names are currently in use for the same or approximate clade.

Comments. The name was first (informally) defined by Prieto-Márquez et al. (2013) who applied the maximum-clade definition and used Parasaurolophus walkeri as the internal specifier and Lambeosaurus lambei, Tsintaosaurus spinorhinus, and Aralosaurus tuberiferus as the external specifiers. We formalize this definition.

Polacanthinae Lapparent & Lavocat, 1955 (converted clade name)

Registration number: 646

Definition. The largest clade within Ankylosauridae or Nodosauridae containing Polacanthus foxii Owen in Anonymous, 1865 but not Ankylosaurus magniventris Brown, 1908 and Nodosaurus textilis Marsh, 1889. This is a maximum-clade definition. Abbreviated definition: max ∇ ∈ Ankylosauridae ∨ Nodosauridae (Polacanthus foxii Owen in Anonymous, 1865 ~ Ankylosaurus magniventris Brown, 1908 & Nodosaurus textilis Marsh, 1889).

Reference phylogeny. Figure 9 of Yang et al. (2013) is treated here as the primary reference phylogeny. Additional reference phylogenies include Figure 3 of Kirkland (1998), Figure 2 of Thompson et al. (2012), Figure 1 of Arbour, Zanno & Gates (2016), Figure 5 of Rivera-Sylva et al. (2018a), and Figure 9 of Zheng et al. (2018).

Composition. Under the primary reference phylogeny, Polacanthinae comprises Polacanthus foxii and Taohelong jinchengensis.

Synonyms. Jaekel (1910) introduced the name Polacanthidae to include ankylosaurs that appeared intermediate between Ankylosauridae and Nodosauridae. Kirkland (1998) was the first to assess ‘polacanthids’ using cladistic methods and found them to be a clade of early-diverging ankylosaurids, and as such should preferably be called Polacanthinae rather than Polacanthidae, to eliminate the possible confusion that Ankylosauridae and Polacanthidae refer to mutually exclusive clades. Carpenter (2001) argued that Polacanthidae was instead valid and defined the name to cover all ankylosaurs closer to Gastonia than to Edmontonia or Euoplocephalus.

Comments. The name Polacanthinae was (informally) defined before by Yang et al. (2013), who used Polacanthus foxii as the internal specifier and Ankylosaurus magniventris and Panoplosaurus mirus as the external specifiers. Kirkland (1998) diagnosed Polacanthinae as comprising ankylosaurs with an ankylosaurid-like skulls, nearly straight and parallel tooth rows, long basipterygoid processes, well-developed acromion arising from dorsal margin of scapula, ventrally flexed ischia, coossified pelvic osteoderms forming pelvic shield, posteriorly grooved and elongate pectoral osteoderms, and caudal osteoderms large, elongate laterally directed, and with hollow bases. Kirkland (1998) initially found Polacanthinae at the base of Ankylosauridae including Gastonia, Polacanthus, and Mymoorapelta and also referred Hoplitosaurus and Hylaeosaurus to the clade. Arbour, Zanno & Gates (2016), Rivera-Sylva et al. (2018a), and Zheng et al. (2018) inferred what could be called Polacanthinae at the base of Nodosauridae, including Polacanthus foxii and Hoplitosaurus marshi. Polacanthinae is poorly supported in most phylogenetic analyses yet frequently referenced in the literature. Taxa typically referred to as ‘polacanthines’ most often form a grade of early-diverging nodosaurids (e.g., Thompson et al., 2012; Brown et al., 2017). Additional taxonomic and phylogenetic revisions are needed to provide an assessment of Polacanthinae. We define the name here to ensure that it is applicable either within Ankylosauridae or Nodosauridae. If the ‘Polacanthus clade’ is reconstructed outside the Ankylosauridae + Nodosauridae node, the name Polacanthinae becomes inapplicable and the preferred name for the grouping should probably be Polacanthidae (not defined here).

Protoceratopsidae Granger & Gregory, 1923 (converted clade name)

Registration number: 647

Definition. The largest clade containing Protoceratops andrewsi Granger & Gregory, 1923 but not Ceratops montanus Marsh, 1888, Leptoceratops gracilis Brown, 1914b, and Triceratops horridus Marsh, 1889. This is a maximum-clade definition. Abbreviated definition: max ∇ (Protoceratops andrewsi Granger & Gregory, 1923 ~ Ceratops montanus Marsh, 1888 & Leptoceratops gracilis Brown, 1914b & Triceratops horridus Marsh, 1889).

Reference phylogeny. Figure 10 of Morschhauser et al. (2019) is treated here as the primary reference phylogeny. Additional reference phylogenies include Figure S1 of Knapp et al. (2018), Figure 8A of Arbour & Evans (2019), Figure 3 of Yu et al. (2020), and Figure 4 of Yu et al. (2020).

Composition. Under the primary reference phylogeny, Protoceratopsidae comprises Bagaceratops rozhdestvenskyi, Magnirostris dodsoni (?= Bagaceratops rozhdestvenskyi; see Czepiński, 2020), and Protoceratops spp.

Synonyms. No other taxon names are currently in use for the same or approximate clade.

Comments. The name Protoceratopsidae has been (informally) defined before by Sereno (1998, 2005) who applied a maximum-clade definition and used Protoceratops andrewsi as the internal specifier and Triceratops horridus as the external specifier. We include two additional external specifiers Ceratops montanus and Leptoceratops gracilis. C. montanus was added because the name Protoceratopsidae has been traditionally applied to the sister taxon of Ceratopsoidea, and L. gracilis was included to ensure that Protoceratopsidae and Leptoceratopsidae remain mutually exclusive clades.

Rhabdodontidae Weishampel et al., 2003 (converted clade name)

Registration number: 648

Definition. The smallest clade containing Rhabdodon priscus Matheron, 1869 and Zalmoxes robustus (Nopcsa, 1900). This is a minimum-clade definition. Abbreviated definition: min ∇ (Rhabdodon priscus Matheron, 1869 & Zalmoxes robustus (Nopcsa, 1900)).

Reference phylogeny. Figure 4 of Madzia, Boyd & Mazuch (2018) is treated here as the primary reference phylogeny. Additional reference phylogenies include Figure 3 of Madzia, Boyd & Mazuch (2018), Figure 20 of Verdú et al. (2018), Figure 25 of Herne et al. (2019), Figure 3 of Párraga & Prieto-Márquez (2019), Figure 2 of Dieudonné et al. (2020), and Figure 9 of Verdú et al. (2020).

Composition. Under the primary reference phylogeny, Rhabdodontidae comprises Rhabdodon priscus, Zalmoxes robustus, Zalmoxes shqiperorum, Mochlodon suessi, and Mochlodon vorosi.

Synonyms. No other taxon names are currently in use for the same or approximate clade.

Comments. The name Rhabdodontidae was first (informally) defined by Weishampel et al. (2003: 69) who used the minimum-clade definition and selected Rhabdodon priscus and Zalmoxes robustus as the internal specifiers. Sereno (2005) later used a maximum-clade definition, using Rhabdodon priscus as the internal specifier and Parasaurolophus walkeri as the external specifier. We formalize the former, minimum-clade, definition. A definition similar in effect to that of Sereno (2005) is applied to Rhabdodontomorpha.

Rhabdodontomorpha Dieudonné et al., 2016 (converted clade name)

Registration number: 649

Definition. The largest clade containing Rhabdodon priscus Matheron, 1869 but not Hypsilophodon foxii Huxley, 1869 and Iguanodon bernissartensis Boulenger in Beneden, 1881. This is a maximum-clade definition. Abbreviated definition: max ∇ (Rhabdodon priscus Matheron, 1869 ~ Hypsilophodon foxii Huxley, 1869 & Iguanodon bernissartensis Boulenger in Beneden, 1881).

Reference phylogeny. Figure 2 of Dieudonné et al. (2020) is treated here as the primary reference phylogeny. Additional reference phylogenies include Figure 4 of Madzia, Boyd & Mazuch (2018), Figure 25 of Herne et al. (2019), and Figure 57 of Barta & Norell (2021).

Composition. Under the primary reference phylogeny, Rhabdodontomorpha comprises Muttaburrasaurus langdoni, Fostoria dhimbangunmal, the ‘Vegagete ornithopod’, and members of the clade Rhabdodontidae.

Synonyms. No other taxon names are currently in use for the same or approximate clade.

Comments. The application of Rhabdodontomorpha has been described, and (informally) proposed definitions have been discussed, by Madzia, Boyd & Mazuch (2018: Appendix 1) and Madzia, Jagt & Mulder (2020: Table 1). We therefore refer to these studies for details. Our formalized maximum-clade definition is similar to that of Madzia, Jagt & Mulder (2020) in that it uses Rhabdodon priscus as the internal specifier and Iguanodon bernissartensis as the external specifier. We have further added a second external specifier, Hypsilophodon foxii, to prevent its inclusion to Rhabdodontomorpha under phylogenies similar to that of Norman (2015: Fig. 50).

Saphornithischia (new clade name)

Registration number: 747

Definition. The smallest clade containing Heterodontosaurus tucki Crompton & Charig, 1962, Iguanodon bernissartensis Boulenger in Beneden, 1881, Stegosaurus stenops Marsh, 1887, and Triceratops horridus Marsh, 1889. This is a minimum-clade definition. Abbreviated definition: min ∇ (Heterodontosaurus tucki Crompton & Charig, 1962 & Iguanodon bernissartensis Boulenger in Beneden, 1881 & Stegosaurus stenops Marsh, 1887 & Triceratops horridus Marsh, 1889).

Etymology. Derived from the Greek safis (clear, definite) and formed to include all members of Ornithischia whose placement within the clade is well established.

Reference Phylogeny. Figure 4 of Madzia, Boyd & Mazuch (2018) is treated here as the primary reference phylogeny. Additional reference phylogenies include Figure 25 of Herne et al. (2019), Yang et al. (2020), and Figure 57 of Barta & Norell (2021).

Composition. Under the primary reference phylogeny, Saphornithischia comprises Pisanosaurus mertii and members of the clades Heterodontosauridae and Genasauria.

Synonyms. Under alternative topologies, where Heterodontosauridae is reconstructed within Neornithischia (e.g., Butler, 2005; Xu et al., 2006; Dieudonné et al., 2020), Saphornithischia would be a heterodefinitional synonym of Genasauria.

Comments. Given the repeated inference of heterodontosaurids outside Genasauria in multiple studies (e.g., Butler, Upchurch & Norman, 2008; Boyd, 2015; Han et al., 2018; Madzia, Boyd & Mazuch, 2018; Andrzejewski, Winkler & Jacobs, 2019; Herne et al., 2019; Yang et al., 2020) and the uncertainty surrounding the potential ornithischian affinities of Pisanosaurus mertii (Agnolín & Rozadilla, 2018, Baron, 2019; Desojo et al., 2020) and members of the Silesauridae (Langer & Ferigolo, 2013; Cabreira et al., 2016; Pacheco et al., 2019; Müller & Garcia, 2020), we provide a new clade name to cover taxa whose placement within Ornithischia is well-supported. Note that the internal specifiers Stegosaurus stenops and Triceratops horridus are not included in the primary reference phylogeny. The former belongs to Thyreophora (e.g., Maidment et al., 2020), while the latter is part of Marginocephalia (see, e.g., Morschhauser et al., 2019; Fowler & Freedman Fowler, 2020). Both these clades are indicated on Figure 4 of Madzia, Boyd & Mazuch (2018).

Saurolophinae Brown, 1914a (converted clade name)

Registration number: 650

Definition. The largest clade containing Saurolophus osborni Brown, 1912 but not Lambeosaurus lambei Parks, 1923, provided that it does not include Hadrosaurus foulkii Leidy, 1858. This is a maximum-clade definition. Abbreviated definition: max ∇ (Saurolophus osborni Brown, 1912 ~ Lambeosaurus lambei Parks, 1923 | ~ Hadrosaurus foulkii Leidy, 1858).

Reference phylogeny. Figure 18 of Prieto-Márquez, Wagner & Lehman (2020) is treated here as the primary reference phylogeny. Additional reference phylogenies include Figure 5 of Kobayashi et al. (2019), Figure 11 of Prieto-Márquez et al. (2019), Figure 9 of Zhang et al. (2019), Figure 5 of Zhang et al. (2020), Figure 7 of Kobayashi et al. (2021), and Figure 10 of Longrich et al. (2021).

Composition. Under the primary reference phylogeny, Saurolophinae comprises ?Gryposaurus alsatei, Naashoibitosaurus ostromi, members of the clades Brachylophosaurini, Edmontosaurini, Kritosaurini, and Saurolophini, and the specimen ‘PASAC 1 (‘Sabinosaur’)’.

Synonyms. Following the widespread application of the Principle of Coordination, under which Hadrosaurinae has to be attributed to Cope (1869), the name Hadrosaurinae is generally considered to have priority over Saurolophinae, even though the latter was coined 4 years earlier (Saurolophinae Brown, 1914a; Hadrosaurinae Lambe, 1918). In recent years, both Hadrosaurinae and Saurolophinae, have been used for the sister taxon of Lambeosaurinae. The selection of the proper name has traditionally depended on whether the clade includes Hadrosaurus foulkii or not (Fig. 5). In the cases in which H. foulkii falls within the smallest clade containing Saurolophus osborni and Lambeosaurus lambei, and within the ‘Saurolophus branch’, the name Hadrosaurinae is preferred (e.g., Cruzado-Caballero & Powell, 2017; Xing, Mallon & Currie, 2017; Kobayashi et al., 2019; Zhang et al., 2020). However, when H. foulkii falls outside the clade, the name Saurolophinae is used (e.g., Prieto-Márquez et al., 2019; Prieto-Márquez, Wagner & Lehman, 2020; Gates, Evans & Sertich, 2021; Kobayashi et al., 2021; McDonald et al., 2021; Ramírez-Velasco et al., 2021).

Figure 5 Specifier-based phylogeny of Hadrosauridae showing alternative placements of Hadrosaurus foulkii.

The silhouette of Lambeosaurinae was obtained from phylopic.org (Dmitry Bogdanov, CC BY 3.0). The silhouette of Hadrosaurinae/Saurolophinae was prepared by Victoria M. Arbour (CC BY 4.0).

Comments. The name Saurolophinae has been (informally) defined before by Prieto-Márquez (2010) who applied a maximum-clade definition and used Saurolophus osborni as the internal specifier and Lambeosaurus lambei and Hadrosaurus foulkii as the external specifiers. Here, we formalize a maximum-clade definition of Saurolophinae that applies the name to the sister clade of Lambeosaurinae only on the condition that it does not contain H. foulkii. In turn, the name Hadrosaurinae is defined to be used for the ‘Saurolophus branch’ when H. foulkii falls within the clade. Although our definition may be considered similar to that of Prieto-Márquez (2010) it differs substantially because under our definition, the name Saurolophinae may become inapplicable.

Saurolophini Glut, 1997 (converted clade name)

Registration number: 651

Definition. The largest clade containing Saurolophus osborni Brown, 1912 but not Brachylophosaurus canadensis Sternberg, 1953, Edmontosaurus regalis Lambe, 1917, Hadrosaurus foulkii Leidy, 1858, and Kritosaurus navajovius Brown, 1910. This is a maximum-clade definition. Abbreviated definition: max ∇ (Saurolophus osborni Brown, 1912 ~ Brachylophosaurus canadensis Sternberg, 1953 & Edmontosaurus regalis Lambe, 1917 & Hadrosaurus foulkii Leidy, 1858 & Kritosaurus navajovius Brown, 1910).

Reference phylogeny. Figure 18 of Prieto-Márquez, Wagner & Lehman (2020) is treated here as the primary reference phylogeny. Additional reference phylogenies include Figure 5 of Kobayashi et al. (2019), Figure 11 of Prieto-Márquez et al. (2019), Figure 9 of Zhang et al. (2019), Figure 5 of Zhang et al. (2020), Figure 7 of Kobayashi et al. (2021), and Figure 10 of Longrich et al. (2021).

Composition. Under the primary reference phylogeny, Saurolophini comprises Augustynolophus morrisi, Prosaurolophus maximus, and Saurolophus spp.

Synonyms. No other taxon names are currently in use for the same or approximate clade.

Comments. The name Saurolophini has been (informally) defined before (Sereno, 2005; Prieto-Márquez et al., 2014). Both these definitions were maximum-clade and used Saurolophus osborni as the internal specifier and Edmontosaurus regalis and Maiasaura peeblesorum (Sereno, 2005) or Brachylophosaurus canadensis, Edmontosaurus regalis, Kritosaurus navajovius, and Lambeosaurus lambei (Prieto-Márquez et al., 2014) as the external specifiers. Here we apply a definition similar to that of Prieto-Márquez et al. (2014) but remove L. lambei and instead add Hadrosaurus foulkii.

Shamosaurinae Tumanova, 1983 (converted clade name)

Registration number: 652

Definition. The largest clade containing Gobisaurus domoculus Vickaryous et al., 2001 and Shamosaurus scutatus Tumanova, 1983 but not Ankylosaurus magniventris Brown, 1908. This is a maximum-clade definition. Abbreviated definition: max ∇ (Gobisaurus domoculus Vickaryous et al., 2001 & Shamosaurus scutatus Tumanova, 1983 ~ Ankylosaurus magniventris Brown, 1908).

Reference phylogeny. Figure 11 of Arbour & Currie (2016) is treated here as the primary reference phylogeny. Additional reference phylogenies include Figure 1 of Arbour, Zanno & Gates (2016), Figure 8 of Arbour & Evans (2017), and Figure 5 of Rivera-Sylva et al. (2018a).

Composition. Under the primary reference phylogeny, Shamosaurinae comprises Gobisaurus domoculus and Shamosaurus scutatus.

Synonyms. No other taxon names are currently in use for the same or approximate clade.

Comments. Tumanova (1987) described Shamosaurinae based on a list of diagnostic features: shamosaurines were ankylosaurids with narrow anterior snouts, angle of the orbital plane with the skull axis less than 25°, anterior wall of the pterygoid inclined posteriorly, occipital condyle a wide oval, pterygoids fused with the basisphenoid, small interpterygoid fenestra, and orbits at the midlength of the skull. Shamosaurinae is not reconstructed in all recent phylogenetic analyses, as Shamosaurus and Gobisaurus are sometimes inferred as successive outgroups to Ankylosaurinae rather than as a clade (e.g., Thompson et al., 2012; Wiersma & Irmis, 2018). We provide a maximum-clade definition that makes Shamosaurinae applicable only under the topologies in which Shamosaurus and Gobisaurus are more closely related to each other than either is to Ankylosaurus.

Stegosauria Marsh, 1877b (converted clade name)

Registration number: 653

Definition. The largest clade containing Stegosaurus stenops Marsh, 1887 but not Ankylosaurus magniventris Brown, 1908. This is a maximum-clade definition. Abbreviated definition: max ∇ (Stegosaurus stenops Marsh, 1887 ~ Ankylosaurus magniventris Brown, 1908).

Reference phylogeny. Figure 12 of Maidment et al. (2020) is treated here as the primary reference phylogeny. Additional reference phylogenies include Figure 11 of Maidment et al. (2008), Figure 1 of Raven & Maidment (2017), and Figure 1 of Dieudonné et al. (2020).

Composition. Under the primary reference phylogeny, Stegosauria comprises Isaberrysaura mollensis, Gigantspinosaurus sichuanensis, and members of the clades Stegosauridae and Huayangosauridae.

Synonyms. No other taxon names are currently in use for the same or approximate clade.

Comments. The name Stegosauria has been (informally) defined before (Galton, 1997; Sereno, 1998; Galton & Upchurch, 2004; Sereno, 2005) using Stegosaurus (Galton, 1997; Sereno, 1998; Galton & Upchurch, 2004) or Stegosaurus stenops (Sereno, 2005) as the internal specifier and Ankylosaurus (Galton, 1997; Sereno, 1998), Ankylosauria (Galton & Upchurch, 2004), or Ankylosaurus magniventris (Sereno, 2005) as the external specifiers. Since Stegosauria has never been proposed an alternative use, we apply S. stenops as the internal specifier and A. magniventris as the external specifier. Note that A. magniventris is not included in the primary reference phylogeny. See Figure 3 of Thompson et al. (2012) for its placement with respect to Stegosauria.

Stegosauridae Marsh, 1880 (converted clade name)

Registration number: 654

Definition. The largest clade containing Stegosaurus stenops Marsh, 1887 but not Huayangosaurus taibaii Dong, Tang & Zhou, 1982. This is a maximum-clade definition. Abbreviated definition: max ∇ (Stegosaurus stenops Marsh, 1887 ~ Huayangosaurus taibaii Dong, Tang & Zhou, 1982).

Reference phylogeny. Figure 12 of Maidment et al. (2020) is treated here as the primary reference phylogeny. Additional reference phylogenies include Figure 11 of Maidment et al. (2008) and Figure 1 of Raven & Maidment (2017).

Composition. Under the primary reference phylogeny, Stegosauridae comprises Adratiklit boulahfa, Alcovasaurus longispinus, Dacentrurus armatus, Hesperosaurus mjosi, Jiangjunosaurus junggarensis, Kentrosaurus aethiopicus, Loricatosaurus priscus, Miragaia longicollum, Paranthodon africanus, Stegosaurus homheni, Stegosaurus stenops, and Tuojiangosaurus multispinus.

Synonyms. No other taxon names are currently in use for the same or approximate clade.

Comments. The name Stegosauridae was first (informally) defined by Sereno (1998, 2005) who used the maximum-clade definition and selected Stegosaurus stenops as the internal specifier and Huayangosaurus taibaii as the external specifier. We formalize this definition.

Struthiosaurini (new clade name)

Registration number: 655

Definition. The largest clade containing Struthiosaurus austriacus Bunzel, 1871 but not Nodosaurus textilis Marsh, 1889 and Panoplosaurus mirus Lambe, 1919. This is a maximum-clade definition. Abbreviated definition: max ∇ (Struthiosaurus austriacus Bunzel, 1871 ~ Nodosaurus textilis Marsh, 1889 & Panoplosaurus mirus Lambe, 1919).

Etymology. Derived from the stem of Struthiosaurus Bunzel, 1871, the name of an included taxon, which combines the Latin word struthio (ostrich) and Greek sauros (lizard, reptile).

Reference phylogeny. Figure 5 of Rivera-Sylva et al. (2018a) is treated here as the primary reference phylogeny. Additional reference phylogenies include Figure 1 of Arbour, Zanno & Gates (2016), Figure 3 of Brown et al. (2017), and Figure 9 of Zheng et al. (2018).

Composition. Under the primary reference phylogeny, Struthiosaurini comprises Europelta carbonensis, Hungarosaurus tormai, Pawpawsaurus campbelli, Stegopelta landerensis, and Struthiosaurus spp.

Synonyms. The name Struthiosaurinae Nopcsa, 1923 has been recently used for an approximate clade (Kirkland et al., 2013; Blows & Honeysett, 2014; Villanueva-Amadoz et al., 2015). No other taxon names are currently in use for the same or approximate clade.

Comments. A grouping similar to that covered here under the name Struthiosaurini has previously been named Struthiosaurinae (Kirkland et al., 2013). The name Struthiosaurinae was (informally) defined by Kirkland et al. (2013) who applied the maximum-clade definition and used Europelta as the internal specifier and Cedarpelta, Peloroplites, Sauropelta, and Edmontonia as the external specifiers. Struthiosaurinae was considered to represent the clade of Late Cretaceous European nodosaurids. However, Kirkland et al. (2013) did not include a character matrix or phylogenetic analysis in their study and have not yet published a follow-up paper with results indicating the extent of their Struthiosaurinae. They provided, however, a list of diagnostic characters. According to Kirkland et al. (2013), Struthiosaurinae includes nodosaurid ankylosaurs with narrow predentaries, nearly horizontal and unfused quadrates, quadrate condyles that are 3 times transversely wider than long, premaxillary teeth and dentary teeth that are near the predentary symphysis, dorsally arched sacra, an acromion process dorsal to the midpoint of the scapulocoracoid suture, straight ischia with a straight dorsal margin, long slender limbs, a sacral shield, and erect sacral osteoderms with flat bases. This suite of characters was considered to unite Anoplosaurus, Europelta, Hungarosaurus, and Struthiosaurus, but many of these characters have a broad distribution in Ankylosauria and Nodosauridae (Ősi, 2015). Arbour, Zanno & Gates (2016) reconstructed a clade containing Ahshislepelta, Europelta, Hungarosaurus, Niobrarasaurus, Nodosaurus, Pawpawsaurus, Stegopelta, Struthiosaurus, and the ‘Paw Paw juvenile’ as the sister clade to that containing Edmontonia, which would thus be considered Struthiosaurinae. Brown et al. (2017) added Borealopelta to the matrix of Arbour, Zanno & Gates (2016) and reconstructed a clade of Borealopelta, Europelta, Hungarosaurus, and Pawpawsaurus; Stegopelta and Struthiosaurus were outside of this clade and sister to Edmontonia, ‘Denversaurus’, and Panoplosaurus. As was the case with Panoplosaurinae, owing to the fact that the ‘Struthiosaurus clade’ is nested within Nodosaurinae, we prefer to use a name that implies a lesser inclusiveness (that is, -ini rather than -inae). The use of Struthiosaurinae, without discussing the phylogenetic context, may suggest that Struthiosaurinae and Nodosaurinae are mutually exclusive clades. When the suffix -ini is applied, such confusion is eliminated. Note that the recent use of Struthiosaurinae has been largely limited to mentions of Kirkland et al. (2013) application of the name (Blows & Honeysett, 2014; Villanueva-Amadoz et al., 2015).

Styracosterna Sereno, 1986 (converted clade name)

Registration number: 656

Definition. The largest clade containing Iguanodon bernissartensis Boulenger in Beneden, 1881 but not Camptosaurus dispar (Marsh, 1879). This is a maximum-clade definition. Abbreviated definition: max ∇ (Iguanodon bernissartensis Boulenger in Beneden, 1881 ~ Camptosaurus dispar (Marsh, 1879)).

Reference phylogeny. Figure 12 of Madzia, Jagt & Mulder (2020) is treated here as the primary reference phylogeny. Additional reference phylogenies include Figure 20 of Verdú et al. (2018), Figure 9 of Verdú et al. (2020), Figure 11 of McDonald et al. (2021), and Figure 11 of Santos-Cubedo et al. (2021).

Composition. Under the primary reference phylogeny, Styracosterna comprises Cedrorestes crichtoni, Cumnoria prestwichii, Dakotadon lakotaensis, Draconyx loureioi, Fukuisaurus tetoriensis, Hippodraco scutodens, Iguanacolossus fortis, Lanzhousaurus magnidens, Muttaburrasaurus langdoni, Osmakasaurus depressus, Owenodon hoggii, Planicoxa venenica, Theiophytalia kerri, Uteodon aphanoecetes, Yunganglong datongensis, and members of the clade Hadrosauriformes.

Synonyms. No other taxon names are currently in use for the same or approximate clade.

Comments. The name Styracosterna was first (informally) defined by Sereno (1998: 62) who used the maximum-clade definition and selected Parasaurolophus as the internal specifier and Camptosaurus as the external specifier. We prefer to use Iguanodon bernissartensis as the external specifier to maintain the ‘node-branch triplet’ (‘node-stem triplet’ of Sereno (1998: 52–54)) comprising Ankylopollexia, Camptosauridae, and Styracosterna (all formally defined in the present paper). The inclusion of a different external specifier does not change the extent of Styracosterna under any of the published phylogeny inferences.

Thescelosauridae Sternberg, 1937 (converted clade name)

Registration number: 657

Definition. The largest clade containing Thescelosaurus neglectus Gilmore, 1913 but not Iguanodon bernissartensis Boulenger in Beneden, 1881, provided that it does not include Hypsilophodon foxii Huxley, 1869. This is a maximum-clade definition. Abbreviated definition: max ∇ (Thescelosaurus neglectus Gilmore, 1913 ~ Iguanodon bernissartensis Boulenger in Beneden, 1881 | ~ Hypsilophodon foxii Huxley, 1869).

Reference phylogeny. Figure 4 of Madzia, Boyd & Mazuch (2018) is treated here as the primary reference phylogeny. Additional reference phylogenies include Figure 25 of Herne et al. (2019) and Figure 57 of Barta & Norell (2021).

Composition. Under the primary reference phylogeny, Thescelosauridae comprises members of the clades Thescelosaurinae and Orodrominae.

Synonyms. The name Parksosauridae Buchholz, 2002 has been used recently for the same contents (Boyd, 2015; Rivera-Sylva et al., 2018b). No other taxon names are currently in use for the same or approximate clade.

Comments. The name Thescelosauridae has been (informally) defined before (Brown et al., 2013; Madzia, Boyd & Mazuch, 2018). Both these definitions were minimum-clade and used Thescelosaurus neglectus and Orodromeus makelai as the internal specifiers. Madzia, Boyd & Mazuch (2018) further added one external specifier, Iguanodon bernissartensis, to ensure that the name is applicable with a similar circumscription (see Madzia, Boyd & Mazuch, 2018: Appendix 1 for details). We apply a complex maximum-clade definition to ensure that Thescelosauridae is not inferred within Hypsilophodontidae; for example under the potential topology in which Hypsilophodon is the sister taxon to a Thescelosaurinae + Orodrominae node. Even though no such phylogenetic hypothesis has been proposed, the placement of taxa ‘traditionally’ dubbed the ‘hypsilophodonts’ is highly pliable across studies (Han et al., 2018; Madzia, Boyd & Mazuch, 2018; Andrzejewski, Winkler & Jacobs, 2019; Herne et al., 2019; Dieudonné et al., 2020; Rotatori, Moreno-Azanza & Mateus, 2020; Yang et al., 2020) and often differs significantly even under different tree-search methods applied to a single dataset. Therefore, it can be expected that phylogeny inferences of the rootward neornithischian-ornithopod transitional segment of the ornithischian phylogenetic trees may result in such topology at some point. A maximum-clade definition with a single internal specifier (T. neglectus) was preferred to allow Thescelosauridae in use regardless of the relationship of T. neglectus to O. makelai.

Thescelosaurinae Sternberg, 1940 (converted clade name)

Registration number: 658

Definition. The largest clade within Hypsilophodontidae or Thescelosauridae containing Thescelosaurus neglectus Gilmore, 1913 but not Hypsilophodon foxii Huxley, 1869 and Orodromeus makelai Horner & Weishampel, 1988. This is a maximum-clade definition. Abbreviated definition: max ∇ ∈ Hypsilophodontidae ∨ Thescelosauridae (Thescelosaurus neglectus Gilmore, 1913 ~ Hypsilophodon foxii Huxley, 1869 & Orodromeus makelai Horner & Weishampel, 1988).

Reference phylogeny. Figure 4 of Madzia, Boyd & Mazuch (2018) is treated here as the primary reference phylogeny. Additional reference phylogenies include Figure 25 of Herne et al. (2019) and Figure 57 of Barta & Norell (2021).

Composition. Under the primary reference phylogeny, Thescelosaurinae comprises Notohypsilophodon comodorensis, Parksosaurus warreni, and Thescelosaurus spp.

Synonyms. No other taxon names are currently in use for the same or approximate clade.

Comments. The name Thescelosaurinae has been (informally) defined before (Brown & Druckenmiller, 2011; Boyd, 2015). Both these definitions were maximum-clade and used Thescelosaurus neglectus as the internal specifier and Orodromeus makelai and Hypsilophodon foxii (Brown & Druckenmiller, 2011) or Orodromeus makelai and Parasaurolophus walkeri (Boyd, 2015) as the external specifiers. Considering the ‘traditional concept’ of Thescelosaurinae, as a subclade of Thescelosauridae/‘hypsilophodonts’, and keeping in mind the unstable phylogenetic position of H. foxii (e.g., Madzia, Boyd & Mazuch, 2018), we apply Thescelosaurinae only when it is inferred either within Thescelosauridae or Hypsilophodontidae (see Article 11.14 of the ICPN).

Thyreophora Nopcsa, 1915 (converted clade name)

Registration number: 659

Definition. The largest clade containing Ankylosaurus magniventris Brown, 1908 and Stegosaurus stenops Marsh, 1887 but not Iguanodon bernissartensis Boulenger in Beneden, 1881 and Triceratops horridus Marsh, 1889. This is a maximum-clade definition. Abbreviated definition: max ∇ (Ankylosaurus magniventris Brown, 1908 & Stegosaurus stenops Marsh, 1887 ~ Iguanodon bernissartensis Boulenger in Beneden, 1881 & Triceratops horridus Marsh, 1889).

Reference phylogeny. Figure 16 of Han et al. (2018) is treated here as the primary reference phylogeny. Additional reference phylogenies include Figure 4 of Madzia, Boyd & Mazuch (2018), Figure 25 of Herne et al. (2019), Figure 1 of Dieudonné et al. (2020), Figure 12 of Yang et al. (2020), and Figure 57 of Barta & Norell (2021).

Composition. Under the primary reference phylogeny, Thyreophora comprises Scutellosaurus lawleri, Emausaurus ernsti, Scelidosaurus harrisonii, and members of the clade Eurypoda.

Synonyms. No other taxon names are currently in use for the same or approximate clade.

Comments. The name Thyreophora has been (informally) defined before (Sereno, 1998; Sereno, 2005; Norman, 2021). All these definitions were maximum-clade. The definitions of Sereno (1998, 2005) used Ankylosaurus (Sereno, 1998) or Ankylosaurus magniventris (Sereno, 2005) as the internal specifier, and Triceratops (Sereno, 1998) or Triceratops horridus, Parasaurolophus walkeri, and Pachycephalosaurus wyomingensis (Sereno, 2005) as the external specifiers. In turn, Norman (2021) defined Thyreophora using Euoplocephalus and Stegosaurus as the internal specifiers and Hypsilophodon as the external specifier. In order to maintain the ‘traditional’ concept of Genasauria as a clade comprising Neornithischia and Thyreophora, the internal specifiers in the definition of Thyreophora are used from among the taxa representing the two major subclades – Ankylosauria (Ankylosaurus magniventris) and Stegosauria (Stegosaurus stenops) – and the external specifiers are used from among the taxa representing the neornithischian clades Ornithopoda (Iguanodon bernissartensis) and Marginocephalia (Triceratops horridus). Note that the internal specifier Ankylosaurus magniventris and the external specifier Triceratops horridus are not included in the primary reference phylogeny. The former belongs to Ankylosauria within Thyreophora (see, e.g., Thompson et al., 2012), while the latter is part of Ceratopsia (e.g., Morschhauser et al., 2019).

Triceratopsini Longrich, 2011 (converted clade name)

Registration number: 692

Definition. The largest clade containing Triceratops horridus Marsh, 1889 but not Anchiceratops ornatus Brown, 1914c and Arrhinoceratops brachyops Parks, 1925. This is a maximum-clade definition. Abbreviated definition: max ∇ (Triceratops horridus Marsh, 1889 ~ Anchiceratops ornatus Brown, 1914c & Arrhinoceratops brachyops Parks, 1925).

Reference phylogeny. Figure 9a of Fowler & Freedman Fowler (2020) is treated here as the primary reference phylogeny. Additional reference phylogenies include Figure 11 of Longrich (2011), Figure 3 of Brown & Henderson (2015), Figure 14 of Mallon et al. (2016), and Figure 3 of Campbell et al. (2019).

Composition. Under the primary reference phylogeny, Triceratopsini comprises Eotriceratops xerinsularis, Nedoceratops hatcheri, Ojoceratops fowleri, Torosaurus spp., and Triceratops spp.

Synonyms. No other taxon names are currently in use for the same or approximate clade.

Comments. The name was first (informally) defined by Longrich (2011) who applied the maximum-clade definition and used Triceratops horridus as the internal specifier and Anchiceratops ornatus and Arrhinoceratops brachyops as the external specifiers. We formalize this definition.

Tsintaosaurini Prieto-Márquez et al., 2013 (converted clade name)

Registration number: 660

Definition. The largest clade containing Pararhabdodon isonensis Casanovas-Cladellas, Santafé-Llopis & Isidro-Llorens, 1993 and Tsintaosaurus spinorhinus Young, 1958 but not Aralosaurus tuberiferus Rozhdestvensky, 1968, Lambeosaurus lambei Parks, 1923 and Parasaurolophus walkeri Parks, 1922. This is a maximum-clade definition. Abbreviated definition: max ∇ (Pararhabdodon isonensis Casanovas-Cladellas, Santafé-Llopis & Isidro-Llorens, 1993 & Tsintaosaurus spinorhinus Young, 1958 ~ Aralosaurus tuberiferus Rozhdestvensky, 1968 & Lambeosaurus lambei Parks, 1923 & Parasaurolophus walkeri Parks, 1922).

Reference phylogeny. Figure 18 of Prieto-Márquez, Wagner & Lehman (2020) is treated here as the primary reference phylogeny. Additional reference phylogenies include Figure 20 of Xing, Mallon & Currie (2017), Figure 5 of Kobayashi et al. (2019), Figure 11 of Prieto-Márquez et al. (2019), Figure 5 of Zhang et al. (2020), Figure 7 of Kobayashi et al. (2021), and Figure 11 of McDonald et al. (2021).

Composition. Under the primary reference phylogeny, Tsintaosaurini comprises Pararhabdodon isonensis and Tsintaosaurus spinorhinus.

Synonyms. No other taxon names are currently in use for the same or approximate clade.

Comments. The name was first (informally) defined by Prieto-Márquez et al. (2013) who applied the minimum-clade definition and used Pararhabdodon isonensis and Tsintaosaurus spinorhinus as the internal specifiers. We preserve the original intent of Prieto-Márquez et al. (2013) but prefer to use the maximum-clade definition. Pararhabdodon isonensis and Tsintaosaurus spinorhinus are used as the internal specifiers and representatives of Aralosaurini (Aralosaurus tuberiferus), Lambeosaurini (Lambeosaurus lambei), and Parasaurolophini (Parasaurolophus walkeri), as the external specifiers. The name Tsintaosaurini is inapplicable under some recent phylogenies (Prieto-Márquez et al., 2019; Gates, Evans & Sertich, 2021; Longrich et al., 2021).

Discussion

Phylogeny reconstructions of some ornithischian clades currently face challenges that have an impact on the construction of the phylogenetic definitions of several taxon names. Below, we provide discussion of some topological conflicts.

The phylogeny of early-diverging ornithischians

The early evolution of Ornithischia and the phylogenetic relationships of taxa nested near the base of the clade are currently contentious, particularly with respect to the potential Triassic members of the clade. Ornithischians have been ‘traditionally’ represented by a single undisputed Triassic taxon, Pisanosaurus mertii Casamiquela, 1967. Recent reassessments of the type specimen of P. mertii showed, however, that the morphological features of the taxon are rather difficult to interpret and that it may represent either a non-dinosaur dinosauriform from the clade Silesauridae (Agnolín & Rozadilla, 2018, Baron, 2019) or an early-diverging ornithischian (Desojo et al., 2020).

Even if P. mertii turns out to be a silesaurid, however, it may still represent an early-diverging ornithischian dinosaur as a few studies have proposed that silesaurids, a group of Anisian–?Rhaetian (Middle and Late Triassic) dinosauriforms that are usually inferred to be the sister group to dinosaurs (e.g., Nesbitt et al., 2010; Peecook et al., 2013; Ezcurra, 2016; Cau, 2018; Ezcurra et al., 2020), may form an early clade of ornithischians (Langer & Ferigolo, 2013; Cabreira et al., 2016; Pacheco et al., 2019) or a paraphyletic assemblage of early-diverging ornithischians (Müller & Garcia, 2020). Such placement of the silesaurid taxa, especially the oldest known members referred to the group, would have considerable implications for the early evolution of dinosaurs as a whole because neither of the two other major dinosaur clades, Theropoda and Sauropodomorpha, are known from the Middle Triassic.

Pending additional studies more focused on the basal dinosauriform-dinosaur transition, we do not define neither Silesauridae Langer et al., 2010 nor the recently proposed name Sulcimentisauria Martz & Small, 2019. If formal definitions for the names are to be proposed in the future, the definitions should comply with all recently proposed phylogenies, including the possible paraphyletic ‘dissolution’ of Silesauridae (Müller & Garcia, 2020) that would make Sulcimentisauria, as (informally) defined by Martz & Small (2019), applicable to a clade containing the vast majority of ‘traditional’ silesaurids and all ‘core’ ornithischians. One option is to restrict the use of Sulcimentisauria for a clade only when inferred within Silesauridae (e.g., ‘max ∇ ∈ Silesauridae (Silesaurus opolensis Dzik, 2003 ~ Asilisaurus kongwe Nesbitt et al., 2010)’), as originally intended by Martz & Small (2019).

Recently, Baron & Barrett (2017, 2018) reconstructed the enigmatic dinosaur Chilesaurus diegosuarezi from the Tithonian (uppermost Jurassic) of Central Patagonian Cordillera in Chile to represent the earliest-diverging ornithischian, which was in striking contrast to the original inference of the taxon at the base of Tetanurae, within Theropoda (Novas et al., 2015). Although the proposed placement of Chilesaurus among early-diverging ornithischians does not have any impact on the use of particular clade names (except when the extent of Ornithischia is to be indicated on some recently inferred phylogenies; see Fig. 3), it is perhaps appropriate to express some skepticism towards this inference. As already noted by Müller et al. (2018), the originally proposed tetanurine affinities have not been tested by Baron & Barrett (2017, 2018), nor by Baron (2019), or Müller & Dias-da-Silva (2019), all of whom have also reconstructed C. diegosuarezi at the earliest-diverging position within Ornithischia. It is worth noting that Dieudonné et al. (2020) included C. diegosuarezi in their data matrix as well; however, despite being considered an ornithischian by the authors, its placement at the very base of their tree does not indicate ornithischian affinities for the taxon. Dieudonné et al. (2020) did not include any theropods and/or sauropodomorphs in their analysis and, as such, they did not explore the placement of C. diegosuarezi among dinosaurs. In turn, the studies of Baron & Barrett (2017, 2018), Baron (2019), and Müller & Dias-da-Silva (2019) have all been based on a dataset modified from the one first published by Baron, Norman & Barrett (2017a) that was constructed to test the relationships of rootward dinosaurs (ornithischians, theropods, and sauropodomorphs), especially Late Triassic and Early Jurassic forms (though some younger taxa of Ornithischia were included as well) and their closest pan-avian relatives. We are of the opinion that, at present and with the evidence provided, the placement of Chilesaurus, a latest Jurassic taxon with mosaic features, at the very base of a clade that originated in the Triassic or at the Triassic/Jurassic boundary interval, may be best interpreted as being indicative of inadequate/inappropriate data sampling. In other words, the dataset of Baron, Norman & Barrett (2017a) and its more recent versions are most likely unable to actually test the phylogenetic placement of Chilesaurus.

The phylogenetic placement of Heterodontosauridae

The members of Heterodontosauridae have long been treated as early-diverging ornithopods (e.g., Sereno, 1986, 1998, 1999). The last two decades have shown, however, that heterodontosaurids represent some of the more problematic ornithischian groups, with some studies inferring them as non-ornithopod neornithischians (Butler, 2005), as the sister group to Marginocephalia (Xu et al., 2006), near the base of Ornithischia (e.g., Butler, Upchurch & Norman, 2008; Boyd, 2015; Sereno, 2012; Dieudonné et al., 2016; Han et al., 2018; Madzia, Boyd & Mazuch, 2018; Andrzejewski, Winkler & Jacobs, 2019; Herne et al., 2019; Yang et al., 2020), and within Pachycephalosauria (Dieudonné et al., 2020). With respect to the recent reconstruction of heterodontosaurids as early-diverging pachycephalosaurs by Dieudonné et al. (2020), it is worth noting that Heterodontosauridae still form a clade (contra Dieudonné et al., 2020). Even though some taxa that are usually inferred as members of Heterodontosauridae (Echinodon becklesii and Tianyulong confuciusi) are placed more closely to pachycephalosaurids in Dieudonné et al. (2020: Fig. 1) than to Heterodontosaurus, making the ‘traditional’ composition of the group as inferred in other recent studies paraphyletic, Heterodontosauridae still comprises Abrictosaurus consors, Fruitadens haagarorum, Heterodontosaurus tucki, and Lycorhinus angustidens in that study. Similarly, under the topology of Xu et al. (2006), heterodontosaurids and marginocephalians were inferred as the sister taxa, forming a clade named Heterodontosauriformes. Such topology has not been supported in more recent studies (see studies cited above).

Regardless of which of the hypotheses will gain further support in subsequent studies, the definition of the name Heterodontosauridae needs to reflect each of them. Therefore, the applied phylogenetic definition of the name includes representatives of all major ornithischian lineages, Ceratopsia (Triceratops horridus), Ornithopoda (Iguanodon bernissartensis), Pachycephalosauria (Pachycephalosaurus wyomingensis), and Thyreophora (Stegosaurus stenops).

The early-diverging thyreophorans and ankylosaurs

The ‘armored’ dinosaurs, Thyreophora, comprise two major clades, Ankylosauria and Stegosauria, and other taxa that are more closely related to members of the two species-rich lineages than to ornithopods and marginocephalians. These include Emausaurus ernsti, Scelidosaurus harrisonii, and Scutellosaurus lawleri (Han et al., 2018; Herne et al., 2019; Madzia, Boyd & Mazuch, 2018; Dieudonné et al., 2020), and some other, more problematic taxa, such as the dubious ‘Tatisaurus oehleri’ (Norman, Butler & Maidment, 2007) and ‘Bienosaurus lufengensis’ (Raven et al., 2019). Lesothosaurus diagnosticus and Laquintasaura venezuelae have been inferred as early-diverging thyreophorans as well (see, e.g., Butler, Upchurch & Norman, 2008 for the placement of L. diagnosticus, and, e.g., Baron, Norman & Barrett, 2017c and Andrzejewski, Winkler & Jacobs, 2019 for the position of La. venezuelae). Other studies, however, place Le. diagnosticus as an early neornithischian (Madzia, Boyd & Mazuch, 2018; Herne et al., 2019) or an early-diverging ornithischian in general (Han et al., 2018; Andrzejewski, Winkler & Jacobs, 2019; Dieudonné et al., 2020; Yang et al., 2020), and La. venezuelae as an early-diverging ornithischian (Han et al., 2018; Dieudonné et al., 2020; Yang et al., 2020).

Following his thorough redescription of Scelidosaurus harrisonii (Norman, 2020a, 2020b, 2020c), Norman (2021) assessed the phylogenetic relationships of early-diverging thyreophorans and reconstructed E. ernsti, Sce. harrisonii, and Scu. lawleri as the earliest-diverging ankylosauromorphs (Ankylosauria sensu this study), restricting the name Ankylosauria to a smaller clade, approximately comprising ankylosaurids and nodosaurids (two definitions – one minimum-clade and one maximum-clade – were provided; both applying the name to the same known contents). Norman (2021: 70) further noted that the node comprising ankylosaurids and nodosaurids “has the potential to become the new taxon Euankylosauria but this additional clade name is neither essential nor particularly desirable”.

When applying a minimum-clade definition (e.g., ‘min ∇ (Ankylosaurus magniventris Brown, 1908 & Nodosaurus textilis Marsh, 1889)’), the name Euankylosauria may indeed be useful in the future, especially if further studies support the placement of some taxa, such as Mymoorapelta maysi and Kunbarrasaurus ieversi (as in Arbour & Currie, 2016), or E. ernsti, Sce. harrisonii, and Scu. lawleri (as in Norman, 2020c), as non-ankylosaurid/non-nodosaurid ankylosaurs. However, there is no need to replace Ankylosauria with Ankylosauromorpha as the name for the largest clade containing A. magniventris but not Stegosaurus stenops. The branch has long been named Ankylosauria and it has always been expected that it may contain taxa with characters that are absent in ‘traditional’ ankylosaurs (i.e., ankylosaurids and nodosaurids). We suggest that the name Ankylosauromorpha is abandoned.

Problematic clades within Ankylosauria

Comprehensive alpha taxonomic reviews and phylogenetic analyses of Ankylosauridae in recent years have clarified many of the interrelationships within this clade (e.g., Arbour & Currie, 2013; Arbour & Currie, 2016). However, similar reviews for Nodosauridae have not been undertaken in recent years, and phylogenetic resolution within Nodosauridae is often poor and inconsistent between different phylogenies (e.g., Thompson et al., 2012; Arbour, Zanno & Gates, 2016; Brown et al., 2017), in part because many recent ankylosaur phylogenetic analyses are modified from Arbour & Currie (2016) which was designed to test relationships within Ankylosauridae, not Nodosauridae. Additionally, many names for clades within Nodosauridae have been introduced by various authors based on proposed diagnostic characters rather than phylogenetic hypotheses, and have not been defined phylogenetically. In particular, the validity of Polacanthidae or Polacanthinae, Sauropeltinae, Struthiosaurinae, and Stegopeltinae, and the contents of Edmontiniinae or Panoplosaurinae, are unclear. In this manuscript we provide a formal definition of Polacanthinae, and discuss the use of Struthiosaurinae and Panoplosaurinae, as the names have been mentioned recently with some frequency and have had informal definitions proposed previously. Ford (2000) introduced the names Sauropeltinae and Stegopeltinae and provided diagnostic characters but did not test their contents phylogenetically; Sauropeltinae included Sauropelta edwardsorum and Silvisaurus condrayi and Stegopeltinae included Aletopelta coombsi, Glyptodontopelta mimus, and Stegopelta landerensis. Sauropelta and Silvisaurus do not form a clade in any recent analyses, nor do Stegopelta, Glyptodontopelta, and Aletopelta. As such, we do not provide formal definitions for Sauropeltinae or Stegopeltinae at this time.

The origin of Ornithopoda

The understanding of the origin and early evolution of Ornithopoda is tightly connected with the knowledge of the character distribution among rootward neornithischians. With that respect, the basal neornithischian-ornithopod transition is among the poorest known stages of the ornithischian evolutionary history, as recent phylogenetic studies that focused on that particular tree segment provided strikingly conflicting topologies (Boyd, 2015; Dieudonné et al., 2016; Han et al., 2018; Madzia, Boyd & Mazuch, 2018; Andrzejewski, Winkler & Jacobs, 2019; Herne et al., 2019; Dieudonné et al., 2020; Yang et al., 2020).

Substantial conflicts are apparent especially with regards to the phylogenetic placements of taxa ‘traditionally’ dubbed the ‘hypsilophodonts’ (compare, e.g., Boyd, 2015; Han et al., 2018; Herne et al., 2019), including Hypsilophodon foxii itself (e.g., Madzia, Boyd & Mazuch, 2018). Phylogeny reconstructions of ornithopods provide more stable results around the node marking the origin of Iguanodontia (e.g., Madzia, Boyd & Mazuch, 2018; Madzia, Jagt & Mulder, 2020), although alternative hypotheses of early iguanodontian phylogenetic relationships exist as well (e.g., Norman, 2015). The names of non-cerapod neornithischian and rootward ornithopod clades are defined here to reflect these uncertainties though we recognize that some potential topologies may still render issues. For example, if Hypsilophodon forms a clade with thescelosaurids but falls outside the Thescelosaurus + Orodromeus node, Hypsilophodontidae would cover Thescelosauridae if the latter name was defined using a minimum-clade definition (as in Brown et al., 2013 and Madzia, Boyd & Mazuch, 2018). We do not include T. neglectus as an external specifier in the definition of Hypsilophodontidae because under the scenario, in which H. foxii would be inferred within the Thescelosaurus + Orodromeus node, the names Thescelosauridae, Thescelosaurinae, and Orodrominae would be all inapplicable, while Hypsilophodontidae could effectively remain in use only for H. foxii. The definitions we propose ensure that if H. foxii is component of the Thescelosaurus + Orodromeus clade, Thescelosauridae becomes inapplicable, while Thescelosaurinae and Orodrominae still remain in use. The potential issue with Hypsilophodontidae covering Thescelosauridae under a topology in which Hypsilophodon is the sister taxon to the Thescelosaurus + Orodromeus node was solved by providing Thescelosauridae with a maximum-clade definition that makes it inapplicable under such scenario.

Hadrosaurid ingroup relationships

Hadrosaurids are some of the most intensively researched ornithischians, with thoroughly explored phylogenetic relationships. Recent studies almost uniformly infer seven major hadrosaurid clades: Brachylophosaurini, Edmontosaurini, Kritosaurini, Lambeosaurini, Parasaurolophini, Saurolophini, and Tsintaosaurini (e.g., Freedman Fowler & Horner, 2015; Prieto-Márquez, Erickson & Ebersole, 2016; Xing, Mallon & Currie, 2017; Kobayashi et al., 2019; Prieto-Márquez et al., 2019; Prieto-Márquez, Wagner & Lehman, 2020; Zhang et al., 2020; Kobayashi et al., 2021; Longrich et al., 2021; McDonald et al., 2021; Ramírez-Velasco et al., 2021). Longrich et al. (2021) recently introduced a new clade name, Arenysaurini, for a diverse grouping of mostly European lambeosaurines, resulting, at the same time, in Tsintaosaurini (as originally used and as defined here) becoming inapplicable. The study of Longrich et al. (2021) was first to infer such topology. Other phylogenetic studies placed Arenysaurus ardevoli either deeply within Lambeosaurini (e.g., Prieto-Márquez, Erickson & Ebersole, 2016; Prieto-Márquez et al., 2019; Zhang et al., 2019; Prieto-Márquez, Wagner & Lehman, 2020; Gates, Evans & Sertich, 2021; Ramírez-Velasco et al., 2021), within Parasaurolophini (Cruzado-Caballero et al., 2013), or as the sister taxon or close to the clade uniting Lambeosaurini and Parasaurolophini (e.g., Pereda-Suberbiola et al., 2009; Cruzado-Caballero, Pereda-Suberbiola & Ruiz-Omeñaca, 2010; Godefroit, Bolotsky & Bolotsky, 2012; Cruzado-Caballero & Powell, 2017; Xing, Mallon & Currie, 2017; Kobayashi et al., 2019; Zhang et al., 2020).

Owing to the fact that the consensus regarding the placement of Arenysaurus ardevoli among lambeosaurines has yet to be reached, and that other ‘arenysaurins’ of Longrich et al. (2021) are distributed across the lambeosaurine tree in other studies, we do not define Arenysaurini here. If future studies support the results of Longrich et al. (2021), Arenysaurini should probably be defined so that it becomes inapplicable if inferred within Lambeosaurini. The easiest way to do so would be to define Arenysaurini through a maximum-clade definition using Arenysaurus ardevoli and at least one other internal specifier that would make the name applicable only in the case Arenysaurus is inferred outside Lambeosaurini. The taxon Adynomosaurus arcanus is a possible candidate, if such a solution is preferred. In turn, Blasisaurus canudoi should be avoided as this taxon has been inferred as the sister taxon of A. ardevoli in some analyses (e.g., Cruzado-Caballero, Pereda-Suberbiola & Ruiz-Omeñaca, 2010; Cruzado-Caballero et al., 2013; Prieto-Márquez et al., 2019; Prieto-Márquez, Wagner & Lehman, 2020; Gates, Evans & Sertich, 2021). Another option is to apply a clause similar to that used in the definitions of Clypeodonta, Euornithopoda, Hypsilophodontia, Orodrominae, and Thescelosaurinae. That is, by using the set theory symbol ∉, meaning “not element of”, the name Arenysaurini could be applicable only under the condition that the clade for which the name was intended was reconstructed outside Lambeosaurini and Parasaurolophini. Such definition could be abbreviated as follows: max ∇ ∉ Lambeosaurini & Parasaurolophini (Arenysaurus ardevoli Pereda-Suberbiola et al., 2009 ~ Lambeosaurus lambei Parks, 1923 & Parasaurolophus walkeri Parks, 1922).

Conclusions

Ornithischian dinosaurs were a major clade of globally distributed Mesozoic archosaurs that achieved substantial taxic diversity and apparent morphological disparity, expressed especially through their cranial features and the body armor of some of their most distinctive members. Throughout their two-century-long research history, ornithischians have been thoroughly assessed both taxonomically and phylogenetically, which has led to the recognition of numerous clades.

Following the pivotal studies establishing the theoretical foundation of the phylogenetic nomenclature in the 1980s and early 1990s, many names for the ornithischian clades have been provided phylogenetic definitions, some of which have proven useful and have not been changed since their introduction.

However, following the 2020 establishment of the International Code of Phylogenetic Nomenclature (ICPN), or the PhyloCode, all of the definitions proposed before the implementation of the Code are treated as formally ineffective.

We have reconsidered the utility of previously proposed phylogenetic definitions of established ornithischian taxon names and provide definitions for 81 names of ornithischian clades, five of which are newly proposed here, as specified by the Articles of the ICPN, thus marking a key step towards a formal phylogenetic nomenclature of ornithischian dinosaurs.

We would like to express our gratitude to Academic Editor Fabien Knoll (ARAID-Fundación Conjunto Paleontológico de Teruel-Dinópolis, Teruel, Spain) for handling our manuscript, and Paul M. Barrett (Natural History Museum, London, England), Thomas R. Holtz (University of Maryland, MD, USA), and Max C. Langer (Universidade de São Paulo, São Paulo, Brazil) for their detailed reviews that improved the manuscript.

Institutional abbreviations

CMN Canadian Museum of Nature, Ottawa, Ontario, Canada

CPC Colección Paleontológica de Coahuila, Museo del Desierto, Saltillo, Mexico

GPDM Great Plains Dinosaur Museum, Malta, Montana, USA

MOR Museum of the Rockies, Bozeman, Montana, USA

PASAC Paleontological Association of Sabinas, Coahuila, Mexico

ROM Royal Ontario Museum, Toronto, Ontario, Canada

UTEP Centennial Museum and Chihuahuan Desert Gardens, University of Texas at El Paso, Texas, USA

ZPAL Institute of Paleobiology, Polish Academy of Sciences, Warsaw, Poland

Additional Information and Declarations

Competing Interests

Author Contributions

Data Availability

Andrew A. Farke is an Academic Editor and Section Editor for PeerJ.

Daniel Madzia conceived and designed the experiments, performed the experiments, analyzed the data, prepared figures and/or tables, authored or reviewed drafts of the paper, and approved the final draft.

Victoria M. Arbour performed the experiments, analyzed the data, authored or reviewed drafts of the paper, and approved the final draft.

Clint A. Boyd performed the experiments, analyzed the data, authored or reviewed drafts of the paper, and approved the final draft.

Andrew A. Farke performed the experiments, analyzed the data, authored or reviewed drafts of the paper, and approved the final draft.

Penélope Cruzado-Caballero performed the experiments, analyzed the data, authored or reviewed drafts of the paper, and approved the final draft.

David C. Evans performed the experiments, analyzed the data, authored or reviewed drafts of the paper, and approved the final draft.

The following information was supplied regarding data availability:

This study did not generate any raw data.

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
