# Peer review of "The phylogenetic nomenclature of ornithischian dinosaurs"

_PeerJ, doi:10.7717/peerj.12362_

## Round 0.1 · original submission · Minor Revisions

Please, together with your unmarked revised manuscript, provide a marked-up copy as well as a document explaining how you have addressed the issues raised by the reviewers.

·

Basic reporting

This style of paper (i.e., the revision of nomenclature to fit the requirements of the new PhyloCode) has already shown up for other taxa and is an expectation for our new taxonomic system. This is just a personal thought and not a critique of the paper or such, but I do feel mildly troubled that the coiners of the original terms are not invited on such papers. Obviously when Marsh and Seeley and Nopsca are not around to participate I have no qualms about this, but when those who have actually created these names in the context of the phylogenetic system of nomenclature are still extant, I find it a tiny bit disconcerting they aren’t in on these revisions. Of course, this would result in extremely long authorship (perhaps even as long a list as on a typical modern genome or physics paper), and would be unwieldy both in organizing the manuscript and in citing the new name. Still, it just strikes me as a bit odd. That said, I salute the authors here for taking up the task, and they have done an excellent job in doing so.

I have little to comment on in terms of the body of the paper: the definitions and anchor taxon choices seem very reasonable, and the authors show every evidence of respecting the original spirit of the terminology (e.g., the distinction between a Norman-esque Hypsilophodontia and a more standard Hypsilophodontidae.) Some specific cases are discussed below in “General Comments.”

Experimental design

The paper is nomenclatural, not experimental. No hypotheses were tested as such.

That said, I very much appreciate Figure 2 and the demonstration of how the system operates under two mutually exclusive topologies. However, these are only slightly different situations from each other, and I believe I would be extremely illustrative to show how the nomenclature operates under some radically different topologies. In particular, I think using Yang et al. 2020 (reference below: this paper restores a rather refreshingly old-fashioned pre-2010s broadly inclusive Ornithopoda), Madzia et al. 2018 (with its ultra-reduced Ornithopoda), and Dieudonné et al. 2020 (“everything you know about neornithischians is wrong; heterodontosaurs are basal pachychephalosaurs”) could be very instructive to how the nomenclature works under alternative hypotheses.

Yang, Y., W. Wu, P.-E. Dieudonné & P. Godefroit. 2020. A new basal ornithopod dinosaur from the Lower Cretaceous of China. PeerJ 8: e9832. Doi: 10.7717/peerJ.9832

Validity of the findings

Again, this paper is nomenclatural, so it doesn't have "findings" as such.

Additional comments

COMMENTS
Line 410 Lucas (in early editions of his textbook Dinosaurs: The Textbook) used “Pachyrhinosaurinae” instead of “Centrosaurinae”, during the interval when it was thought that “Centrosaurus” was an invalid junior homonym of “Kentrosaurus”. However, the only case I can find where Pachyrhinosaurinae was used in a technical paper was in a comment by Lucas et al. (2016, p. 202) where they claim that this should be the name to use, but employ Centrosaurinae instead.

Lucas, S.G., R.M. Sullivan, A.J. Lichtig, S.G. Dalman & S.E. Jasinksi. 2016. Late Cretaceous dinosaur biogeography and endemism in the Western Interior Basin, North America: A critical re-evaluation. New Mexico Museum of Natural History and Science Bulletin 71: 195-213.
https://www.researchgate.net/profile/Asher-Lichtig-2/publication/303936024_Late_Cretaceous_dinosaur_biogeography_and_endemism_in_the_Western_Interior_basin_North_America_A_critical_re-evaluation/links/575ef2b708aec91374b42caa/Late-Cretaceous-dinosaur-biogeography-and-endemism-in-the-Western-Interior-basin-North-America-A-critical-re-evaluation.pdf

line 565 ff A rare case when I disagree with the authors definitional decision. While it is true that Norman clearly conceived of Clypeodonta as a clade within Ornithopoda, a Hypsilophodon+Iguanodon clade will always exist in any phylogeny and seems worthy as a named entity even in cases where it contains Cerapoda rather than being contained within it. (Such cases exist on the saurischian-side of things: bird phylogenetics can never settle if Ornithurae (the Hesperornis+Aves clade) contains Carinatae (the Ichthyornis+Aves clade) or is instead contained by it.) Thus, I would think they could remove the ∈ Ornithopoda specifier so that the name might serve in contexts such as Madzia et al. 2018.

Line 718 Euceratopsia has been missing from my life and I wasn’t even aware of that fact; thank you for this very useful clade name!

Line 1601 There is no necessity for this suggested inclusion, but given that a masterpiece of Parasaurolophini-literature has just been published (Ramírez-Velasco et al. in press), it might be useful to have this listed among the reference phylogenies:
Ramírez-Velasco, A.A., F.J. Aguilar, R. Hernández-Rivera, J.L.G. Maussán, M.L. Rodríguez & J. Alvarado-Ortega. In press. Tlatolophus galorum, gen. et sp. nov., a parasaurolophini dinosaur from the upper Campanian of the Cerro del Peublo Formation, Coahuila, northern Mexico. Cretaceous Research in press. Doi: 10.1016/j.cretres.2021.104884

Lines 1650-1652 Wise move on “Polacanthidae”.

Line 1906 ff In Dieudonné et al. 2020 and Yang et al. 2020 Thescelosaurus spp. are on a branch by themselves, exclusive of Orodomeus, Hypsilophodon, or Parksosaurus. In those two phylogenies that clade is simply “Thescelosaurus” but one can envision that additionally Jurassic and pre-Maastrichtian taxa closer to Thescelosaurus than to all other currently known dinosaurs (required by these topologies) will be discovered. Do you really want to deprive them of being in “Thescelosauridae” simply because Orodromeus is not with them? Do you really want that on your conscience? (In other words, I wonder if a definition that doesn’t necessarily include Orodromeus might work? At least consider this.)
For what it is worth, I am all for the definition of Thescelosaurinae provided.

Line 1975 Please correct the typo here: “Thyrophora”.

Line 2047 I wonder, relative to all this business at the base of Ornithischia, if it might not be prudent to include a clade name for Pisanosaurus + Genasauria, provided that Pisanosaurus is closer to Genasauria than to Silesaurus, Sauropodomorpha, or Theropoda. Marsh’s 1894 Predentata might serve.
Marsh, O.C. 1894. The typical Ornithopoda of the American Jurassic. American Journal of Science, Series 3 48: 85-90.

Line 2059 Actually, Dieudonné et al.’s 2020 phylogeny reduces the total “gappiness” of the ornithischian record. The addition of extensions of Ornithopoda from the Middle to Early Jurassic and of Ceratopsia from the earliest Late to Early Jurassic is a smaller amount of time than the earliest Oxfordian-to-Turonian gap in Pachycephlosauria required in the standard topology.

Line 2104 ff A very important point about “Ankylosauromorpha”.

Line 2138 In this discussion, Yang et al. 2020 should be added (and in particular as it has the most inclusive recent incarnation of Ornithopoda).

Line 2169 In a similar context, Ramírez-Velasco et al. in press might be added to this discussion of hadrosaurid ingroup relationships.

·

Basic reporting

The MS by Madzia et all provides a timely set of phylogenetic definitions for ornithischian clades, following the publication of the PhyloCode and Phylonyms. My understanding is that most of the definitions are adequate, balancing the historical understanding and current use of the names. My main concerns about some of the definitions are highlighted below and, along with other minor (typos and style) suggestions, in the annotated PDF.

Many of the “primary reference phylogenies” do not have all the specifiers figured in the tree. Even if this is obvious in most of the cases (e.g., Ankylosaurus magniventris is within Ankylosauria in Fig. 12 of Maidment et al. 2020), in other cases this imprecision may render the definition not promptly understandable, especially for the non-specialist. I strongly recommend that, for every specifier not strictly included in the “primary reference phylogeny”, its position should be indicated in the tree as, e.g., “Ankylosaurus magniventris is nested within Ankylosauria in Fig. 12 of Maidment et al. (2020)”. Alternatively, another “primary reference phylogeny” may be chosen.
Below is an exhaustive list of such cases, but please make sure I did not miss any of them …
For Ankylosauria, Stegosaurus stenops is not indicated in Fig. 11 of Arbour and Currie (2016).
For Ankylosauridae, Nodosaurus textilis is not indicated in Fig. 11 of Arbour and Currie (2016).
For Cerapoda, Triceratops horridus and Pachycephalosaurus wyomingensis are not indicated in Fig. 4 of Madzia et al. (2018).
For Ceratopsia, Ceratops montanus and Pachycephalosaurus wyomingensis are not indicated in Fig. 10 of Morschhauser et al. (2019).
For Euiguanodontia, Dryosaurus altus and Camptosaurus dispar are not indicated in Fig. 13 of Coria and Salgado (1996).
For Eurypoda, Stegosaurus stenops is not indicated in Fig. 3 of Thompson et al. (2012).
For Genasauria, Triceratops horridus and Ankylosaurus magniventris are not indicated in Fig. 16 of Han et al. (2018).
For Heterodontosauridae, Triceratops horridus, Stegosaurus stenops, and Pachycephalosaurus wyomingensis are not indicated in Fig. 4 of Madzia et al. (2018).
For Hypsilophodontia, Tenontosaurus tilletti is not indicated in Fig. 50 of Norman (2015).
For Jeholosauridae, Triceratops horridus and Pachycephalosaurus wyomingensis are not indicated in Fig. 25 of Herne et al. (2019).
For Marginocephalia, Ceratops montanus, Triceratops horridus and Pachycephalosaurus wyomingensis are not indicated in Fig. 16 of Han et al. (2018).
For Neornithischia, Triceratops horridus, Ankylosaurus magniventris and Stegosaurus stenops are not indicated in Fig. 4 of Madzia et al. (2018).
For Ornithopoda, Triceratops horridus and Pachycephalosaurus wyomingensis are not indicated in Fig. 4 of Madzia et al. (2018).
For Pachycephalosauria, Triceratops horridus and Ceratops montanus are not indicated in Fig. 27 of Schott and Evans (2017).
For Pachycephalosauridae, Heterodontosaurus tucki is not indicated in Fig. 27 of Schott and Evans (2017).
For Stegosauria, Ankylosaurus magniventris is not indicated in Fig. 12 of Maidment et al. (2020).
For Thyreophora, Triceratops horridus and Ankylosaurus magniventris are not indicated in Fig. 16 of Han et al. (2018).

The composition of some of the defined clades is in fact uncertain based on some of the “primary reference phylogenies”, because the pointed nodes are represented by polytomies. In such cases, it would be adequate to either change the “primary reference phylogenies” or discuss the taxa that may alternatively belong or not into the named clade.
Below is an exhaustive list of such cases, but please make sure I did not miss any of them …
For Elasmaria, it is not clear in Fig. 31 of Rozadilla et al. (2019) if Anabisetia saldiviai, Atlascopcosaurus loadsi, Fulgurotherium austral, Gasparinisaura cincosaltensis, Kangnasaurus coetzeei, Morrosaurus antarcticus, Notohypsilophodon comodorensis, Quantassaurus intrepidus and Trinisaura santamartaensis belong into the clade.
For Genasauria, it is not clear in Fig. 16 of Han et al. (2018) if Lesothosaurus diagnosticus belongs into the clade.
For Hadrosauridae, it is not clear in Fig. 18 of Prieto-Márquez et al. (2020) if Eotrachodon orientalis belongs into the clade.
For Hadrosauriformes, it is not clear in Fig. 12 of Madzia et al. (2020) if Siamodon ninmgami, Hypselospinus fittoni, Barilium dawsoni and Lurdusaurus arenatus belong into the clade.
For Hadrosauroidea, it is not clear in Fig. 12 of Madzia et al. (2020) if Siamodon ninmgami, Hypselospinus fittoni, Barilium dawsoni and Lurdusaurus arenatus belong into the clade.
For Hadrosauromorpha, it is not clear in Fig. 12 of Madzia et al. (2020) if Zouyunlong huangi and Sirindhorna khoratensis belong into the clade.
For Heterodontosauridae, it is not clear in Fig. 4 of Madzia et al. (2018) if Pisanosaurus mertii belongs into the clade.
For Iguanodontidae, it is not clear in Fig. 12 of Madzia et al. (2020) if Siamodon ninmgami, Hypselospinus fittoni, Barilium dawsoni and Lurdusaurus arenatus belong into the clade.

Experimental design

No comment.

Validity of the findings

See basic reporting

Additional comments

No comment

·

Basic reporting

This contribution is clearly written and the standard of written English is excellent throughout. It is extensively referenced though it might be useful to add a brief overview of changes and conflicts in ornithischian classification through time in the Introduction to set the scene for some of the subsequent discussion, although I wouldn't regard this as mandatory (and some of this information is mentioned at specific points in the MS where most directly relevant). There are no raw data to share as this is essentially a meta-analysis, pulling together published information in a novel way. Similarly, there is no hypothesis-testing here: the authors are instead providing a framework for future work, so the MS is fully self-contained in this respect. The paper is well referenced though there are a few minor issues that need to be addressed (missing references, a couple of superfluous references, one or two incorrect taxonomic references; these are noted on the annotated .pdf).

Experimental design

The paper is written in the context of the recently published PhyloCode and offers a novel perspective on the phylogenetic definition of ornithischian dinosaur clades, building upon previous work in the area. They follow the articles of the PhyloCode and use topologies taken from a range of published phylogenetic analyses to formalize and update a large number of clade definitions. The authors have clearly gone to some lengths to provide rigorous definitions and have attempted to future-proof some of these against possible future changes in tree topology, reflecting current uncertainties.

Validity of the findings

I read this paper through the lens of someone who is unconvinced of the need for the PhyloCode and who sees a number of disadvantages in this approach. That said, I cannot fault the authors' approach to the topic, which is internally consistent and fills a gap left in the current PhyloCode compendium, which omits ornithischian dinosaurs. The taxonomic work underpinning this article is solid and most of the things I find unpalatable about this article relate to the rules of PhyloCode itself, rather than the authors' conclusions or approach. As it stands, the MS is essentially a catalogue of definitions, with some commentary on several current problems in ornithischian phylogeny, which is entirely appropriate. I make a number of minor suggestions on the attached .pdf, but would note the following suggestions that the authors might like to consider in terms of a more general commentary on the application of PhyloCode through the example of ornithischians:
1. One of the aims of taxonomy is stability, and given that many parts of the ornithischian tree are currently highly labile I think setting these definitions in stone may be too soon (the lack of consensus over some major nodes in the tree means that there are at least three competing broad phylogenies available, as well as more minor re-arrangements within more deeply nested clades). The authors do discuss this, and have clearly thought about these issues, but I wonder if they could make more discussion of these potential conflicts to show how definitions (or names at least) for some taxa could change with changes in tree topology. Adding a figure to show an example or two of these changes could be useful for readers.
2. The PhyloCode has some areas that conflict with ICZN rules (e.g. over authorship of family-group names and the Principle of Coordination). This means that we can expect some conflicts when comparing this new scheme to the older literature. Also, as the ICZN isn't going anywhere, and as there's no compulsion for any individual researcher or journal to adopt PhyloCode (whereas ICZN rules are currently embedded in many journal policies), we could be in for a messy time taxonomically. For example, it would be entirely possible for different groups of researchers to name or resurrect a taxon under ICZN rules that would have different authorship under PhyloCode rules. This doesn't seem like a recipe for taxonomic stability.
3. It would be useful to know how the authors selected their reference phylogenies, as there's an ever increasing field to choose from. Are there consistent standards or is the choice subjective? At the moment, the authors provide no obvious rationale for choosing one reference phylogeny over another. Are the authors selecting them on the basis of taxonomic scope, numbers of characters, a particular type of analysis (e.g. MP vs ML), a particular type of standard tree topology (e.g. strict consensus) or some other criterion? As these trees are forming the backbone for a rigid set of definitions, which are meant to be in place for sometime, it would be good to know what the criteria for these choices are.
4. It seems bizarre to me that PhyloCode allows clade specifiers that are taxonomically indeterminate at the genus and species level, just because their name (historically) was the basis for a taxon name etymologically. Taxonomies should be based on clearly definable entities not ones that are essentially acting as wildcards within a tree (especially within a set of rules that is meant to be based explicitly on tree-based thinking).

Additional comments

A few minor comments:
1. I think your definition of 'formal' is equal to pre-PhyloCode. Most of us would probably consider any published definition as 'formal', especially those that did mention what type of clade definition and how it was specified. You might consider tweaking this a little as it slightly belittles the work of others who spent considerable time thinking about these issues before the publication of PhyloCode.
2. You could consider adding something on how any of your definitions of very early nodes might change if the enigmatic dinosaur Chilesaurus is an early member of Ornithischia, as suggested by a couple of papers (Baron & Barrett, 2018; Müller & Dias-da-Silva, 2019; Dieudonné et al., 2020).
3. A couple of taxon authorships are incorrect (Polacanthus, Dryosauridae; see .pdf).
4. One or two suggested minor tweaks to clade discussions (Iguanodontoidea, Ankylosauromorpha; see .pdf).
5. Minor referencing issues mentioned in .pdf.

---

## Round 0.2 · accepted · Accept

I confirm that your manuscript has been accepted for publication.

Please correct "OLFAL" on line 2619 (it should be O) and the volume and page numbers on line 2628 (it should be 47: 270 if I'm not mistaken).

·

Basic reporting

The manuscript is clearly written, and is extensively referenced (indeed, the few mis-referenced taxon names in the first version are now corrected.)

The flow of the paper works well.

This will be an important reference for future workers.

Experimental design

Not an experimental paper as such, so n/a.

Validity of the findings

Again, this paper doesn't have "findings" as such.

Additional comments

The previous manuscript was already in quite a fine shape, and post-review it is even more improved.

I look forward to its publication.